# Variability of the thermohaline structure and transport of Atlantic water in the Arctic Ocean based on NABOS hydrography data

Nataliya Zhurbas[1] and Natalia Kuzmina[1]

[1]Shirshov Institute of Oceanology, Russian Academy of Sciences, 36 Nakhimovsky Prospekt , 117997 Moscow, Russia

*Correspondence to:* Nataliya Zhurbas (nvzhurbas@gmail.com)

**Abstract.** Conductivity-temperature-depth (CTD) transects across continental slope of the Eurasian Basin and the St. Anna Trough performed during NABOS (Nansen and Amundsen Basins Observing System) project in 2002–2015 and a transect from the Polarstern-1996 expedition are used to describe the temperature and salinity characteristics and volume flow rates of the current carrying the Atlantic Water (AW) in the Arctic Ocean. The variability of the AW on its pathway along the slope of Eurasian Basin is investigated. A dynamic Fram Strait branch of the Atlantic Water (FSBW) is identified on all transects, including two transects in the Makarov Basin (along 159°E), while the cold waters on the eastern transects along 126°E, 142°E and 159°E , which can be associated with the influence of the Barents Sea branch of the Atlantic water (BSBW), were observed in the depth range below 800 m and had a negligible effect on the spatial structure of isopycnic surfaces. The geostrophic volume transport of AW decreases farther away from the areas of the AW inflow to the Eurasian Basin, decreasing by one order of magnitude in the Makarov Basin at 159°E, implying that the major part of the AW entering the Arctic Ocean circulates cyclonically within the Nansen and Amundsen Basins. There is an absolute maximum of $\theta_{max}$ (AW core temperature) in 2006–2008 time series and a maximum in 2013, but only at 103°E. Salinity $S(\theta_{max})$ (AW core salinity) time series display an increase of the AW salinity in 2006–2008 and 2013 (at 103°E) that can be referred to as a AW salinization in the early 2000s. The maxima of $\theta_{max}$ and $S(\theta_{max})$ in 2006 and 2013 are accompanied by the volume transport maxima. The time average geostrophic volume transports of AW are 0.5 Sv in the longitude range 31–92°E, 0.8 Sv in the St. Anna Trough and 1.1 Sv in the longitude range 94–107°E.

## 1 Introduction

Atlantic water (AW) enters the Eurasian Basin in two branches (see, e.g., Aagaard, 1981; Rudels et al.,1994; Schauer et al., 1997; Rudels et al., 1999; Schauer et al., 2002a, b; Rudels et al., 2006; Berzczynska-Möller et al., 2012; Rudels et al., 2015; Rudels, 2015; Dmitrenko et al.,

2015; Pnyushkov et al., 2015, 2018b): one branch originates from the Greenland and Norwegian seas and flows to the basin through Fram Strait (the Fram Strait branch of the Atlantic Water, hereinafter the FSBW), and the other reaches the deep part of the Arctic Ocean near St. Anna Trough after passing through the Barents Sea (the Barents Sea branch of the Atlantic water, hereinafter the BSBW). After entering the Eurasian Basin the FSBW moves eastward with a subsurface boundary current and has a core of higher temperature and salinity than the BSBW. In the longitude range of 80–90°E, it encounters and partially mixes with the BSBW, which is strongly cooled due to mixing with shallow waters of the Arctic shelf seas and atmospheric impact (Schauer et al., 1997; 2002a, b). Further, the water masses resulting from the interaction of the two branches spread cyclonically in the Eurasian Basin.

Within the NABOS (Nansen and Amundsen Basins Observing System) project (Polyakov et al., 2007) a unique volume of CTD data was collected: more than 30 sections were made in various regions of the Arctic Basin in the summer/fall 2002-2015. A number of sections in different years were made in the same regions of the Basin, which allows studying the interannual variability of the water masses thermohaline structure and the geostrophic volume flow rate in these areas.

The main goal of this work is to investigate the spatial and temporal variability of the AW geostrophic volume flow rate during its propagation along the continental slope of the Eurasian Basin. We further discuss the thermohaline structure and transformation of the FSBW and BSBW. The estimates of the AW transport are sensitive to the temperature and salinity ranges used for the identification of this water (Pnyushkov et al., 2018b) and mixing of FSBW, BSBW and surrounding waters may change the AW geostrophic volume flow rate.

**2 Material and Methods**

We used data from  the CTD transects across the slope of the Eurasian Basin in the longitude range of 31–159°E measured in the years 2002–2015 within the framework of NABOS project (in total 39 transects). The data are freely available at the site http://nabos.iarc.uaf.edu. In addition, a CTD transect across the entire Eurasian Basin and over the Lomonosov Ridge starting at 92°E at the slope from R/V *Polarstern* in 1996 (hereafter PS96) was also included. The locations of the CTD transects are shown in Fig. 1. Most of the transects are aligned cross-slope and grouped at longitudes of 31, 60, 90, 92, 94, 96, 98, 103, 126, 142, and 159°E. Four of the 40 transects crossed zonally the St. Anna Trough (at the latitude of 81, 81.33, 81.42, and 82°N) through which the BSBW enters the Eurasian Basin. Most of the CTD casts covered the upper layer from the sea surface to either 1000 m depth or to the bottom (if the total depth was

shallower). Approximately every third or fourth cast was down to the sea bottom even if the sea depth exceeded 1000 m.

To estimate the volume transport of the Atlantic Water, we applied standard dynamical method. The no-motion level (the depth of zero velocity) was determined from the following consideration. If the baroclinic current occupies the upper layer or/and some intermediate layer, the no-motion level can be chosen in a calm deep layer (where the horizontal density gradient is relatively small). On the contrary, in case of a near-bottom gravity flow, the no motion level can be reasonably chosen well above the near-bottom flow. We adopted for the level of no-motion either 1000 m depth or the sea bottom depth if the latter was smaller than 1000 m for the FSBW, and approximately 50 m, where density contours were more or less flat, for the observations of BSBW in the St. Anna Trough (see also below).

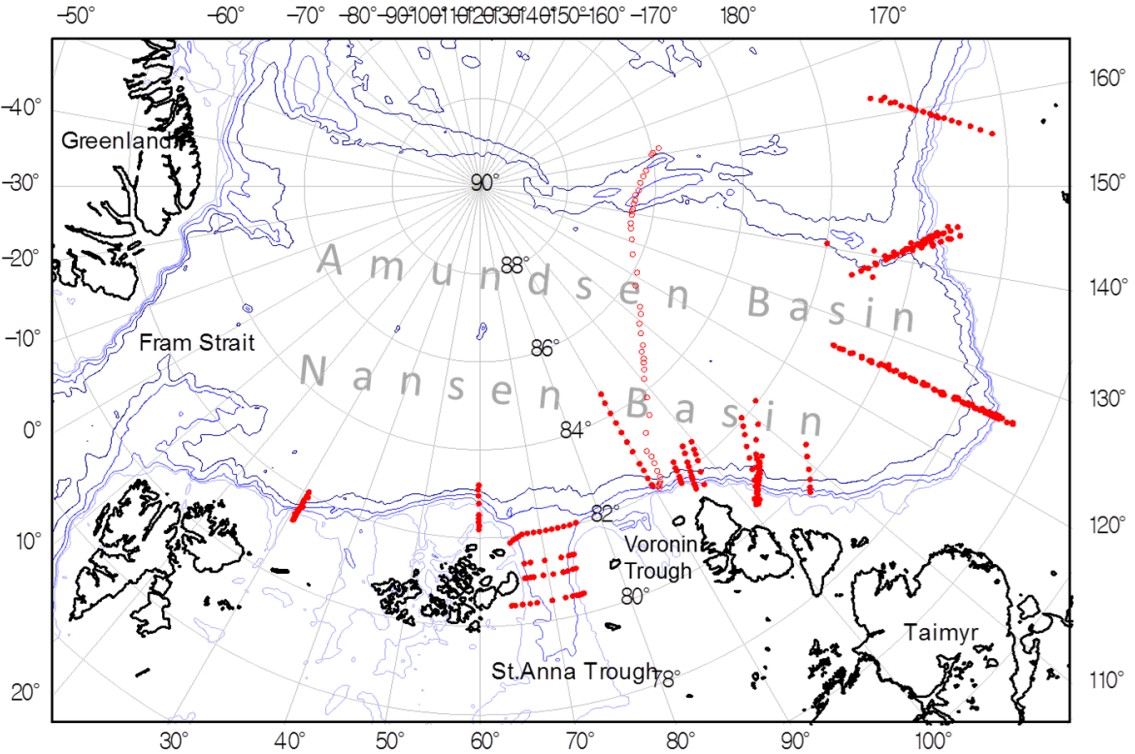

Fig. 1. Bathymetric map of the Eurasian Basin with 300, 500, 1000, and 2000 m contours shown. The red filled and blank circles are the locations of CTD stations on the NABOS and PS96 transects, respectively.

Since the FSBW brings saline and warm water to the Eurasian Basin, the geostrophic transport was found by integration over the depth range with positive temperature, $\theta > 0$ °C, and relatively high salinity, $S > 34.5$ (the salinity is given in the practical salinity scale), that is, near-surface layers with warm and fresh water (which cannot be attributed to AW) were excluded. For the observations of BSBW in the St. Anna Trough the geostrophic transport was calculated by

integration over a depth range with temperature below 0 °C and salinity above 34.5. If both AW branches were present on the transect, the integration was performed over the entire depth range but excluding the cold near-surface layer ($\theta < 0$ °C) and the warm ($\theta > 0$ °C) and relatively fresh ($S < 34.5$) near-surface layer. The zero velocity depth in this case was chosen after inspection of the observed pattern of density contours, i.e. suggesting either the near-surface flow pattern or the near-bottom flow pattern (see Section 3). The details and limitation of the geostrophic velocity calculations are discussed in Zhurbas (2019).

## 3. Results

### 3.1 Variability of the thermohaline pattern on the AW pathway along the slope of Eurasian Basin

#### 3.1.1 CTD transects analysis

The transformation of thermohaline signatures (i.e. patterns of salinity $S$, potential temperature $\theta$, and potential density anomaly $\sigma_\theta$, versus cross-slope distance and depth) of the AW flow on its pathway along the slope of the Eurasian Basin are presented in Fig.2. The $\sigma_\theta$ contours on transects at 31°E diverge towards the continental slope margin (to the south), shallowing above the warm/saline core of the AW and sloping down beneath it associated with a eastward subsurface flow. Such distribution of isopycnic surfaces was observed on all NABOS transects taken across available continental slope at 31°E. According to Fig. 2 the warm/saline core of the Fram Strait Branch of the AW with the maximum temperature $\theta max$ of 4.88°C at the depth $Z_{\theta max}=102$ m and the maximum salinity $Smax$ of 35.11 at the depth $Z_{Smax}=176$ m is found on the slope at about 1000 m isobath.

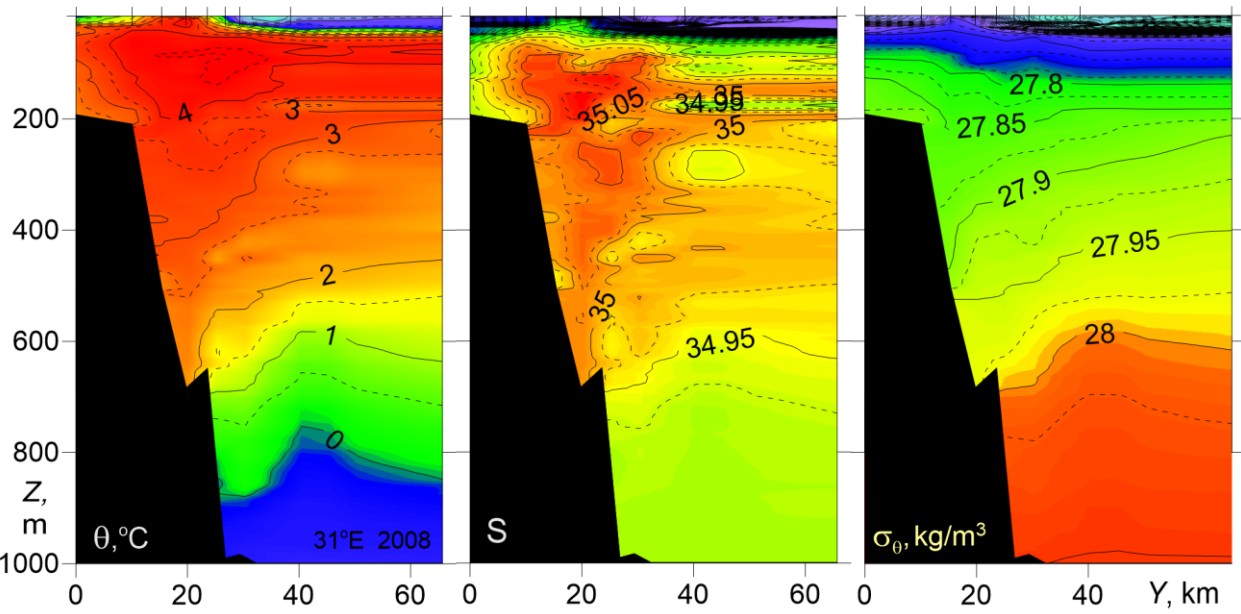

Fig. 2. Temperature $\theta$, salinity $S$, and potential density anomaly $\sigma_\theta$ versus cross-slope distance and depth for the NABOS-2008 transect across the Eurasian Basin slope at 31°E.

Figure 3 presents temperature, salinity, and potential density for two zonal transects across the St. Anna Trough at latitudes of 81 and 82°N. A stable pool of cold ($\theta < 0$°C) and dense ($\sigma_\theta > 28$ kg/m$^3$) water in the bottom layer is seen adjacent to the eastern slope of the trough. The transfer of the densest water pool to the eastern slope corresponds to a geostrophically balanced near-bottom gravity flow to the North. This near-bottom gravity current carries also waters of
Atlantic origin, which are strongly cooled due to mixing with shelf waters in the Barents and Kara seas. Above the near-bottom gravity flow of the BSBW one can observe two-core structure of warm FSBW with temperature up to 2.5 °C that enters the St. Anna Trough from the north-west at the western side of the trough and leaves it for the north-east at the eastern side of the trough. At 82°N, the BSBW overflows a ridge-like elevation east of the St. Anna Trough (top
panels in Fig. 3). Studies of the currents and hydrography in the St. Anna Trough can be found in (Schauer et al., 2002a, b; Rudels et al., 2015; Dmitrenko et al., 2015).

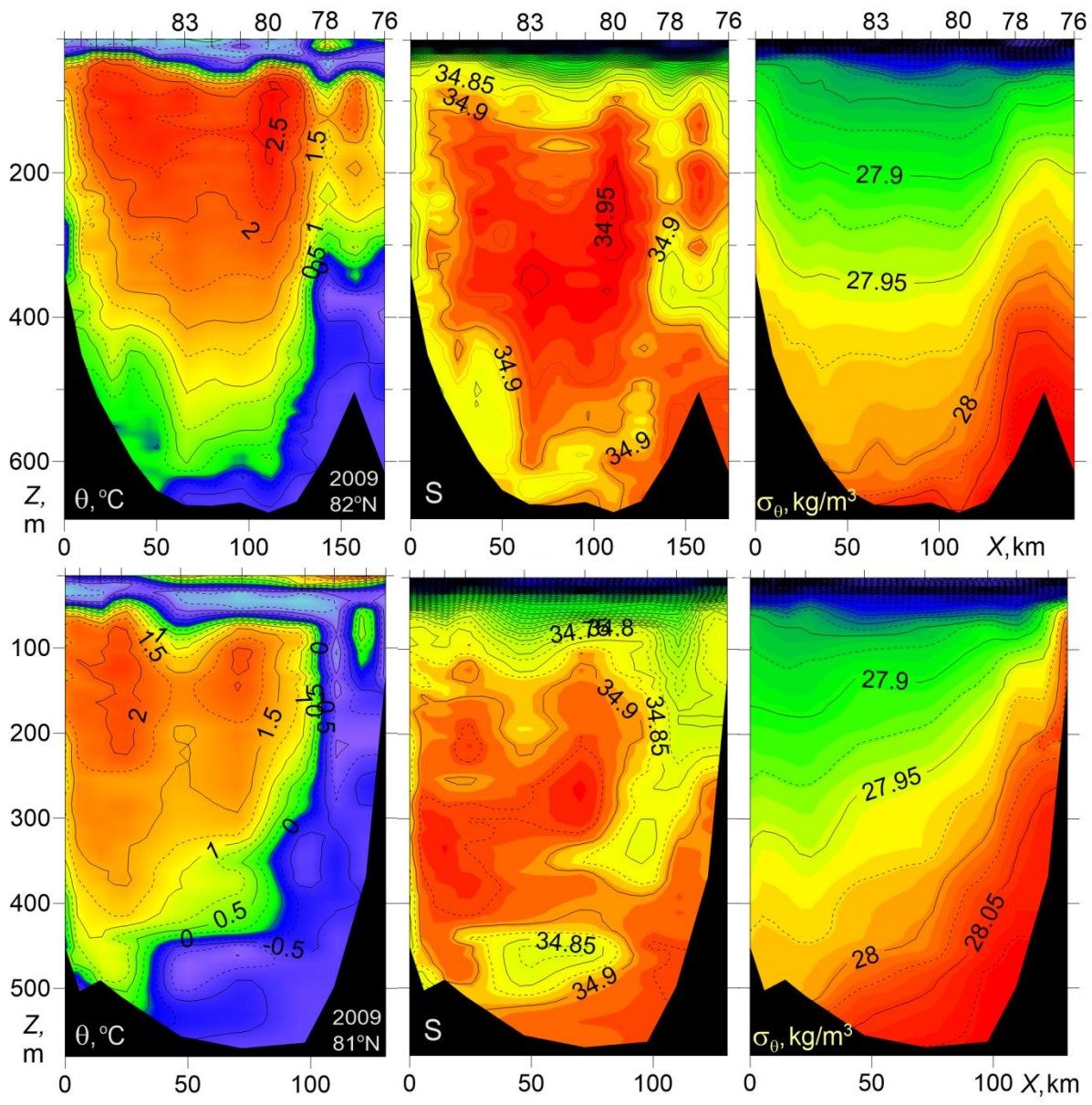

Fig. 3. Temperature $\theta$, salinity $S$, and potential density anomaly $\sigma_\theta$ versus distance and depth for zonal transects across the St. Anna Trough at latitudes of 81°N (bottom, NABOS-2009), and 82°N (top, NABOS-2009). The $X$-axis is directed to the east.

In order to understand the effect of the FSBW and the BSBW transformation on geostrophic volume flow rate, it is necessary to identify water masses of different origin. For that purpose the following criterion is often used (Walsh et al., 2007; Pfirman et al., 1994): the water masses of the FSBW are characterized by $\theta > 0$ °C, and the BSBW can be identified by $-2$ °C $<$ $\theta <0$ °C, $34.75 < S < 34.95$ and $27.8$ kg/m$^3 < \sigma_\theta < 28.0$ kg/m$^3$. Other approaches to define BSBW are given in Schauer et al. (1997; 2002a, b) and Dmitrenko et al. (2015). According to Schauer et al. (1997; 2002a, b) the BSBW includes all waters that enter the Nansen Basin from the St. Anna and Voronin troughs. The temperature of these waters, however, can reach ~1 ºC.

The justification for this approach was based on $\theta$-$S$ analysis of the waters of the north-eastern part of the Barents Sea and the St. Anna and Voronin troughs. According to Dmitrenko et al. (2015), the BSBW consists of two water masses, and the temperature of the warmer water mass can only slightly exceed 0 °C (for more details see section 3.1.2). Here we will rely on the definitions of the FSBW and BSBW proposed by Dmitrenko et al. (2015).

In Fig. 4 the CTD transect at 92°E carried out in the *Polarstern*-1996 expedition just east of the entrance point of the BSBW to the Eurasian Basin from the St. Anna Trough and Voronin Trough is presented. It can be assumed that a part of the BSBW extends deep into the Basin, mixing with the FSBW, while another part of the BSBW flows eastward along the slope according to the general cyclonic circulation observed in the Eurasian Basin. On the presented transect the BSBW is observed in the depth range below 600 m as a narrow, about 10 km wide strip of cold water near the slope (see also Subsection 3.1.2) adjacent to a 300 km wide zone occupied by the warm FSBW. The potential density distribution of FSBW on this transect is similar to transects at 31°E. Namely, despite of the masking effect of vertical undulations of $\sigma_\theta$ contours caused by internal waves and mesoscale eddies (one of subsurface, intra-pycnocline eddies is probably identified at the distance of $Y$=510 km), isopycnals tend to shoal/deepen above/below the FSBW core towards the continental slope margin (to the south) which, in terms of geostrophic balance implies the eastward flow of FSBW. The FSBW core on the 92°E transect is found at 40 km distance from the slope, with the maximum temperature $\theta max$=2.79°C at $Z_{\theta max}$=271 m and salinity $Smax$=34.97 at $Z_{Smax}$=329 m. Therefore, the FSBW on its pathway along the slope of the Eurasian Basin from 31°E to 92°E has cooled, desalinated, sank and become denser by about 2 °C, 0.1, 150 m, and 0.1 kg/m$^3$, respectively. Another distinct feature in the PS96 transect is a layer with increased temperature between 180 and 300 m depth at $Y$=600– 750 km in the vicinity of the Lomonosov Ridge, which can be attributed to the geostrophically-balanced FSBW return flow cyclonically circulating around the Eurasian Basin (Rudels et al., 1994; Swift et al., 1997).

According to Schauer et al. (2002b) who studied the PS-96 section, the horizontal and vertical scales of the BSBW were taken at 30 km and 800 m, respectively. This differs from our interpretation based on the definition of BSBW with temperature less than 0 °C.

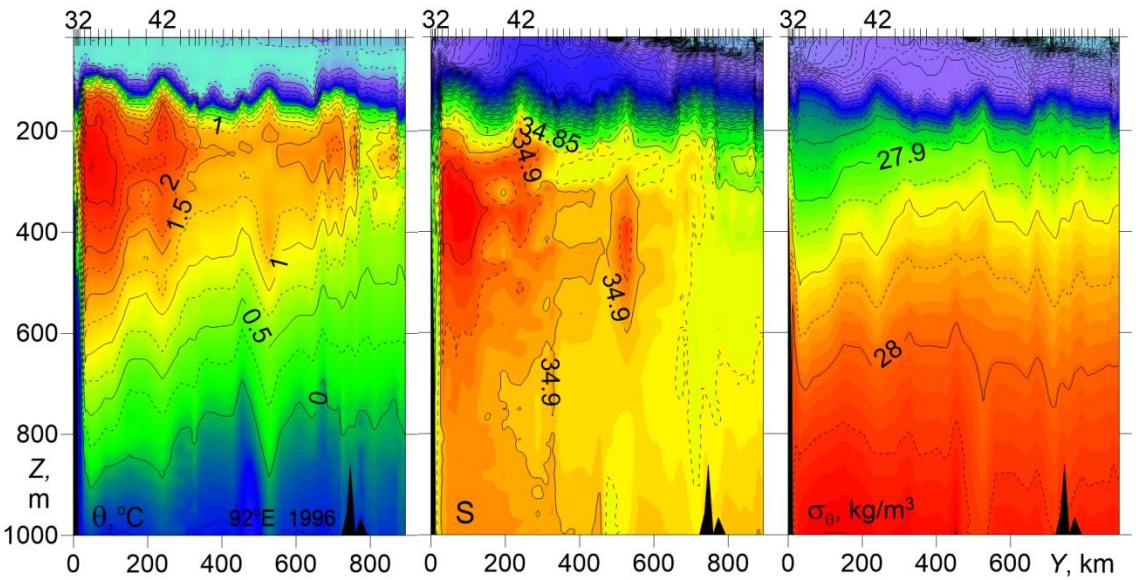

Fig. 4. Temperature $\theta$, salinity $S$, and potential density anomaly $\sigma_\theta$ versus distance and depth for cross-shelf transects at 92°E (PS-1996).

Further east, in the longitude range of 94–107 °E (NABOS-09), the denser part of BSBW under the FSBW is characterized by an eastward geostrophic current with isopycnals sloping towards the North in a 150 km wide zone adjacent to the slope (see Fig. 5, top panel). Less saline water at the slope is the less dense BSBW that has entered the Nansen Basin when the slope narrows north of Severnaya Zemlya (Schauer et al., 1997).

The vertical location of the FSBW layer is similar to the 92°E in the section PS-96 but the maximum temperature has further decreased: in the transect in Fig. 5, the top panel, $\theta max$=1.98 °C at $Z_{\theta max}$=245 m and $Smax$=34.95 at $Z_{Smax}$=365 m. The bottom panel of Fig. 5 presents the transect at 142°E (NABOS-09) which is located on the Lomonosov Ridge, between the Amundsen and Makarov Basins. The comparison of the two transects obtained in the same year shows that the vertical scale of the warm FSBW water ($\theta$>1.5 °C) has significantly decreased. Nevertheless, the FSBW waters are also observed at this longitude and affect the slopes of isopycnic surfaces in a layer up to 300 m. The cold waters with $\theta$<0 °C, which can be associated with the BSBW, are observed only at two stations in the depth range close to 1000 m, and are absent at depths above 950 m. The isopycnic surfaces in the bottom panel of Fig. 5 are relatively flat, indicating weak geostrophic flow (see Section 3.2). The "absolutely stable" thermohaline stratification below the temperature maximum with temperature decreasing and salinity increasing with depth (Fig. 5, bottom panel) is common to the Upper Polar Deep Water (UPDW) layer (Rudels et al., 1999).

In Fig. 6 three transects are presented, at 126°E and 142°E (NABOS-2005) and in the Makarov Basin at 159° E (NABOS-2007). On the transect along 126°E large slopes of isopycnic

surfaces are observed, which corresponds to a fairly strong geostrophic flow (see Section 3.2), confined to the depth range of 200−400 m, that is, to the area occupied by the FSBW. At the 142°E transect on the Lomonosov Ridge, and at the 159°E transect in the Makarov Basin, the FSBW can be still identified as a warm layer between 200 and 400 m, where the maximum temperature is reduced to 1.49 °C and 1.42 °C, respectively (Fig. 6). The 142°E transect implies some eastward geostrophic transport, whereas at the 159° E transect, and in the area of cold waters (below 800 m) in the sections shown in Fig. 6, the baroclinic flow is weak or absent.

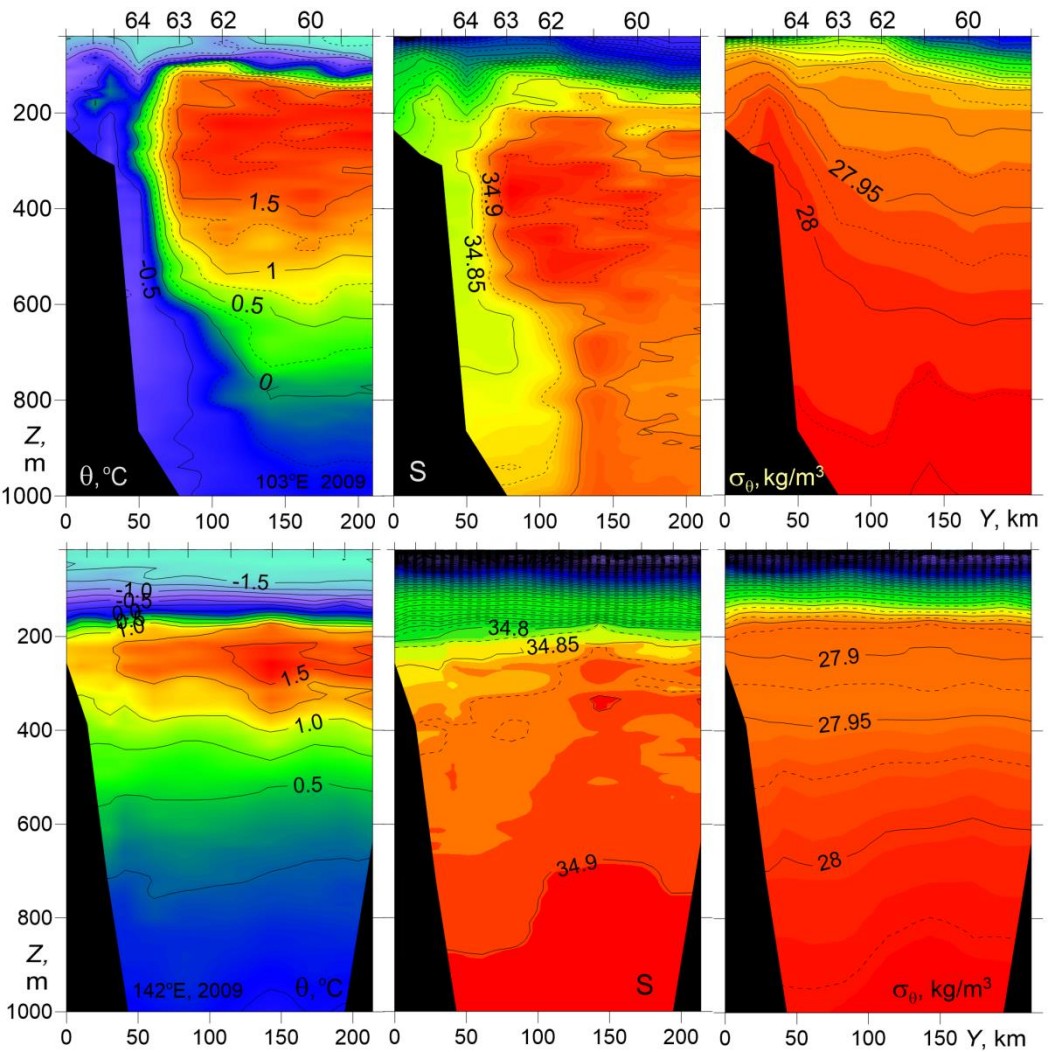

Fig. 5. Temperature $\theta$, salinity $S$, and potential density anomaly $\sigma_\theta$ versus distance and depth for cross-shelf transects at 103°E (upper) and 142°E (lower) (NABOS-09).

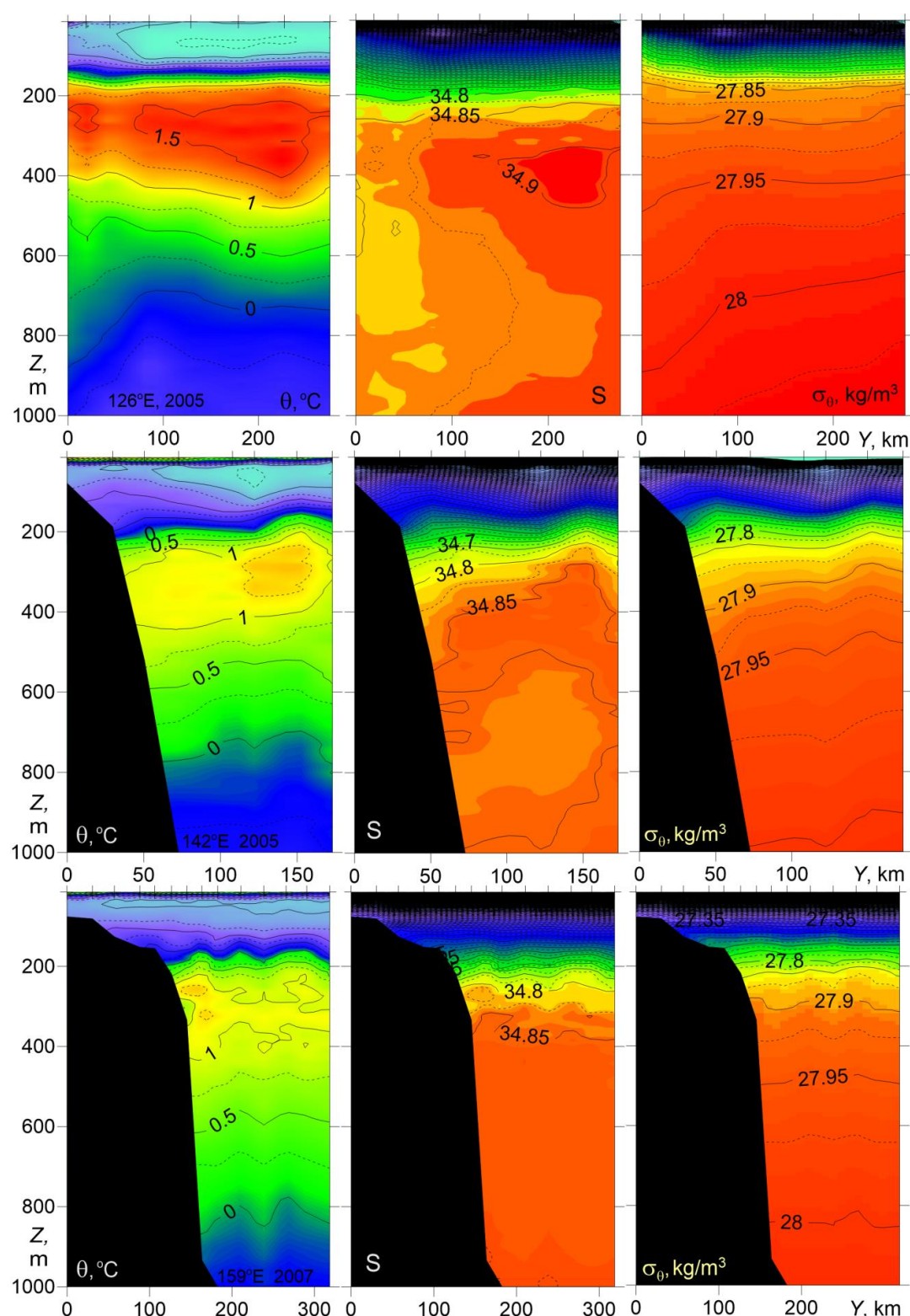

Fig. 6. Temperature $\theta$, salinity $S$, and potential density anomaly $\sigma_\theta$ versus distance and depth for cross-shelf transects at 126°E, 142°E (top and middle, NABOS-2005) and 159°E (bottom, NABOS-2007).

In summary, a combined FSBW-BSBW structure with isopycnals sloping down to the north (from the slope), is typical for the longitude range 94–107°E. In the transects along 126°E, 142°E, and 159°E, sloping isopycnals were observed generally in the depth range of 200–400 m,

that is in the area occupied by the FSBW. As the FSBW moved along the continental slope of the Eurasian Basin, its core temperature decreased, but could be identified at all transects, including the two transects in the Makarov Basin (159°E). The cold waters in the transects along 126°E, 142°E and 159°E, which can be associated with the BSBW, had a minimum temperature above −0.5 °C, were located below 800 m, and had relatively flat isopycnic surfaces.

### 3.1.2 $\theta$-$S$ analysis

The difficulty in identifying the BSBW in the eastern part of the Nansen Basin is related to the overlapping ranges of temperature and salinity inherent to the BSBW and the UPDW: − 0.5 °C<θ<0 °C, and the salinity is close to 34.9 (Rudels et al., 1994; Walsh et al., 2007). It is also important to note that the BSBW in the St. Anna Trough mixes with the FSBW. Therefore, not only the cold Atlantic Waters, which are transported by the bottom gravity current, but also mixed warmer waters can enter the Nansen Basin through the trough (see Fig. 3). A detailed $\theta$-$S$ analysis of different CTD sections can provide useful information on the transport and transformation of FSBW and BSBW. A distinct $\theta$-$S$ signature indicates that the water mass has entered the area of observation. The absence of a signature on the theta-S space indicates either the water mass did not enter the area of observation or was transformed after mixing with other waters.

The differences in the behavior of the $\theta$-$S$ values are observed in the upper and deep layers of the Eurasian Basin and the St. Anna Trough (Fig.7). On the other hand, one cannot miss a similarity in the shape of the $\theta$-$S$ curves in the salinity range of 34.5−35.0. The similarity is obviously caused by the presence of FSBW. Fig. 7 demonstrates the transformation of the FSBW and BSBW moving along the continental slope of the Eurasian Basin. More detailed information on the BSBW transformation can be extracted from $\theta$-$S$ diagrams presented in Fig. 8.

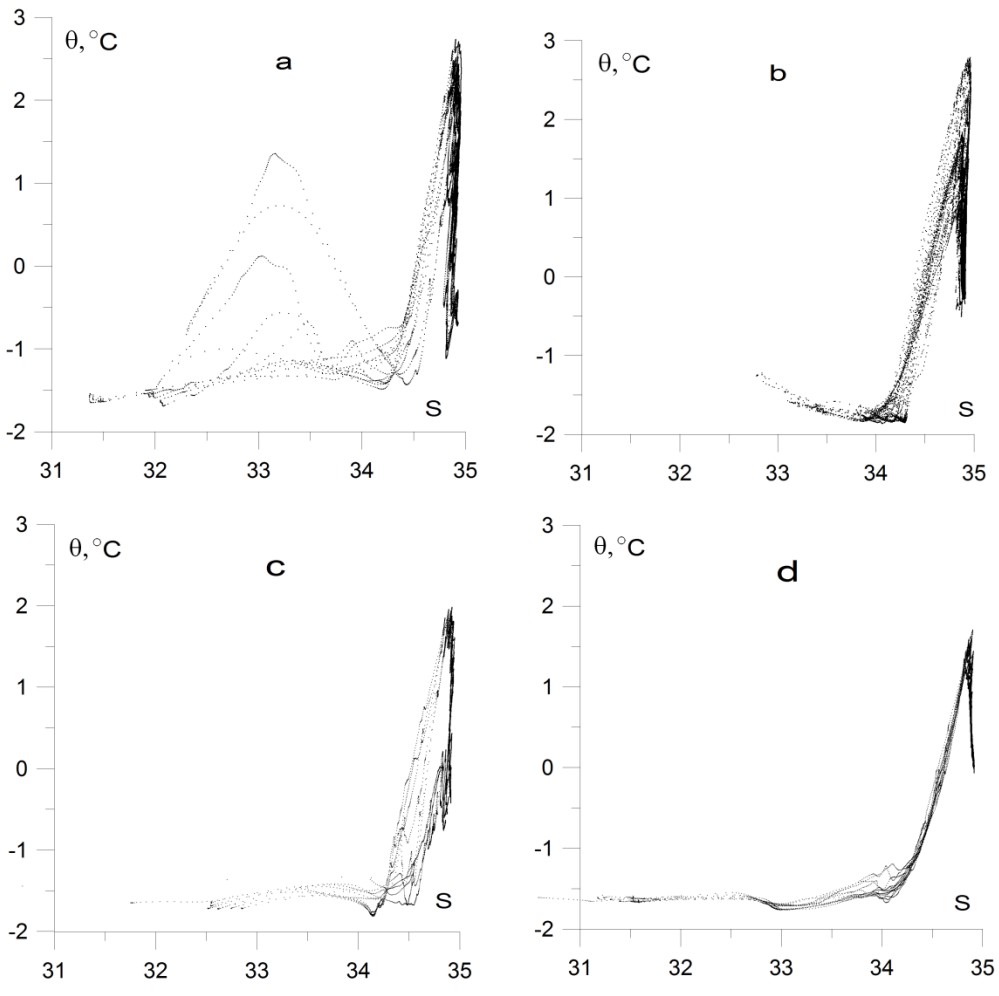

230

Fig. 7. *θ-S* diagrams based on the CTD profiling in (a) the St. Anna Trough (NABOS-09, 82° N), (b) the PS-96 section at 92°E, and the NABOS-09 sections at 103°E (c) and 142°E (d). For convenience of presentation, the points of the *θ-S* curves with salinity below 30 were excluded.

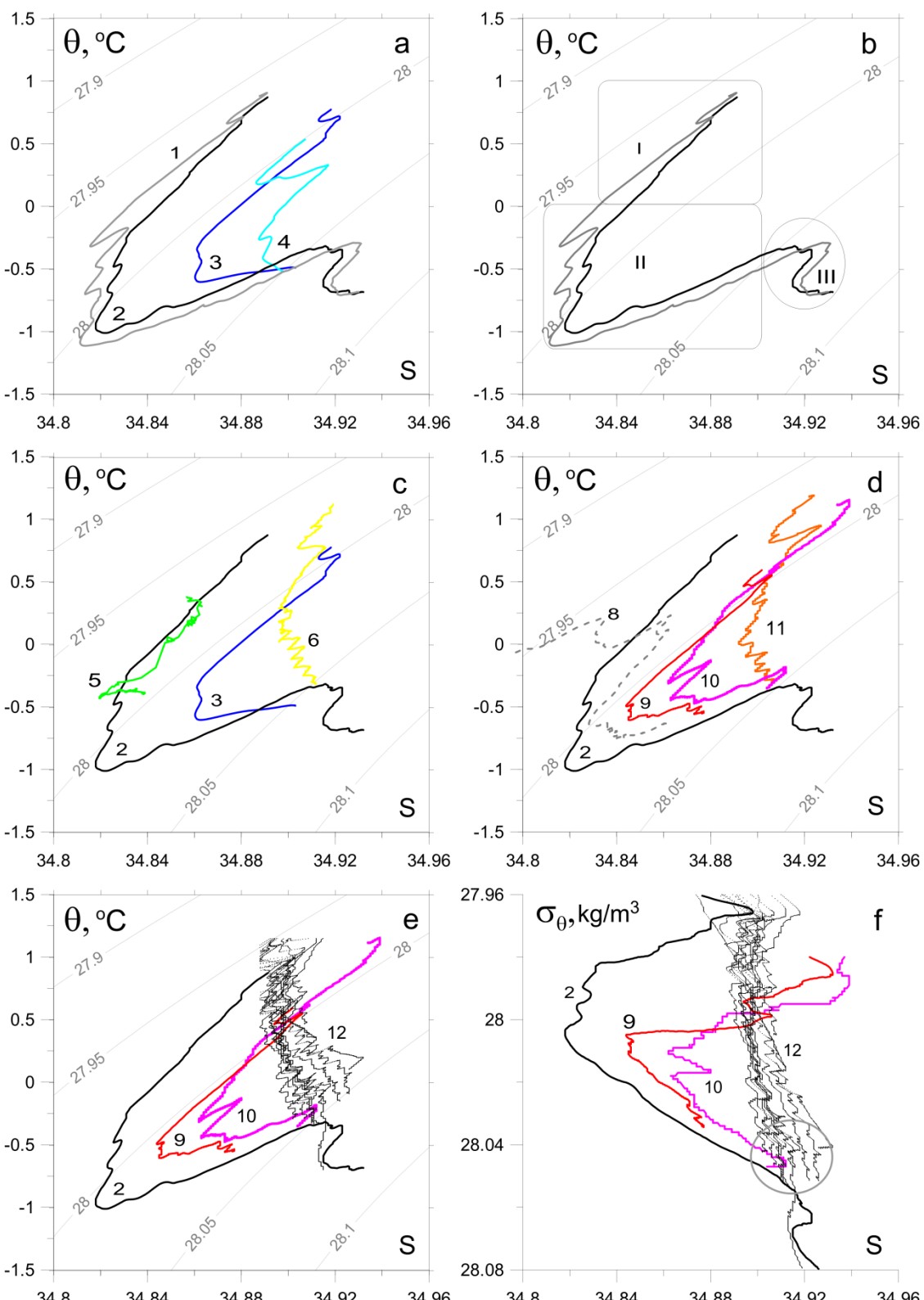

Fig. 8. Thermohaline values of the BSBW and FSBW: a) based upon the CTD profiles, obtained in the St. Anna Trough (NABOS-09, section 82°N), curves 1−4 correspond to the stations (st.) 76, 78, 83 and 80, respectively; b) the same as "a" but only curves 1 and 2 are presented; regions I, II, III illustrate three different water masses in accordance with (Dmitrenko et al., 2015); for explanation see the text; c) based upon the section of PS-96, curves 5 and 6 corresponding to st. 32 and 42, respectively (depth range 600−1000 m), curves 2 and 3 are shown for the reference;

d) for CTD profiles at the 103°E section, NABOS-09, curve 8 (st. 64), curve 9 (st. 63), curve 10 (st. 62), curve 11 (st. 60), and curve 2 for the reference (see Fig. 5 for the location of the stations); e) based upon the CTD profiles in the depth range 500−1200 m measured at the 126°E (section of NABOS-09), curves 12; curves 2, 9 and 10 are shown for the reference; f) the same as "e" but presented in coordinates $\sigma_\Theta$, $S$.

The $\theta$-$S$ curves marked as 1 and 2 in Fig.8a correspond to stations 76 and 78, respectively, which were located at the eastern slope of the St. Anna Trough just in the near-bottom gravity current carrying the BSBW, while the curves marked as 3 and 4 correspond to stations 83 and 80 located near the mid-point (thalweg) of the trough in the western periphery of the gravity current (the location of the stations is shown in Fig. 3). To visualize the BSBW transformation better, the points of $\theta$-$S$ curves in the temperature and salinity ranges of $\theta > 1.2$ °C and $S < 34.76$, respectively, were omitted. Similar $\theta$-$S$ curves in the St. Anna Trough were observed within NABOS Program in other years (NABOS-13, NABOS-15).

The curves 1 and 2 in Fig. 8a have similar knee-like shape (Dmitrenko et al., 2015) formed by (i) the upper warm and saline water layer of the FSBW ($\theta \gg 0$ °C), (ii) the intermediate colder and fresher water layer of BSBW ($\theta < 0$ °C) underlying the FSBW, and (iii) the denser, warmer and saltier "true" mode of the BSBW ($\theta \approx 0$ °C), see Fig. 8b: FSBW (region I), BSBW (region II), "true" mode BSBW (region III). The BSBW differs from the "true" mode BSBW, and is more diluted with the colder and fresher Barents Sea water (for more details see Dmitrenko et al., 2015). We will be interested in the transformation of the main part of the knee (region II), namely the transformation of BSBW.

In Fig. 8c the comparison of typical $\theta$-$S$ curves related to the St. Anna Trough (they are also shown in the other panels of Fig. 8 for reference) with that of the 92°E section of PS-96 is given: the curves 5 and 6 correspond to st. 32 and st. 42 (depth range 600−1000 m) of the PS-96 section, respectively. St. 32 was located next to the slope, while st. 42 was located about 250 km apart from the slope. The coincidence of curve 5 with a part of curve 2 implies a BSBW flow along the slope of Nansen Basin (see Fig. 4 and its legend 1). Curve 6 corresponds to the UPDW. The $\theta$-$S$ diagrams for CTD profiles at the section 103°E are presented by curves 8-11 (see Fig. 5 for the locations of stations). Curves 8, 9, and 10 are similar to curve 2, and indicate the BSBW. Curve 11, similar to curve 6 in Fig. 8c, corresponds to the UPDW. However, the BSBW is not observed at 126°E: see Fig. 8e, where a collection of $\theta$-$S$ curves (collectively referred as 12) presents all CTD profiles in the depth range 500−1800 m measured at 126°E of NABOS-09. Also we do not observe the BSBW further to the east on the 142°E section of NABOS-09 (not shown) or in the Makarov Basin.

The BSBW at 103°E and 126°E is also characterized by a knee-shape in $\sigma_\Theta$, $S$ coordinates (Fig. 8f, numbers correspond to those in other panels) . However the knee-shape diagram is not observed along 126°E (curves 12) in these coordinates. The dense and cold deep waters in the section 126°E have $\sigma_\Theta$, $\theta$, $S$ values typical for the "true" BSBW mode (Dmitrenko et al., 2015). Nevertheless, these waters (see $\sigma_\Theta$, $S$ values inside the circle; Fig. 8f) also correspond to the

UPDW characteristics hence cannot be distinguished as the "true" BSBW mode. To evaluate the transformation of the "true" mode of BSBW an additional analysis is required, which is beyond the scope of this paper.

The BSBW which is characterized by the knee-shape diagram in coordinates $\theta$-$S$ and $\sigma_\Theta$-$S$, is not visible at 126°E (Fig. 8). This is consistent with the conclusion formulated in Subsection

3.1.1 that by 126°E the BSBW is not accompanied by any noticeable tilt of isopycnals. Moreover, given the characteristic feature of the $\theta$-$S$ structure of BSBW in the St. Anna Trough (curves 1−4 in Fig. 8a) was observed in other years, we carried out a similar analysis using all available CTD data and found that the BSBW is not distinct at this longitude (see Fig.9). The only exception was 2002, when the BSBW was still observed at 126ºE. It suggests that the

BSBW and FSBW begin to mix intensively immediately after 103ºE. On the other hand, the FSBW is well identified at 126ºE and further along the slope of the Eurasian Basin (and even in the Makarov Basin), while we cannot say the same about the BSBW. Thus, one may assume that east of 126ºE the geostrophic volume flow rate of the AW is mainly provided by the FSBW.

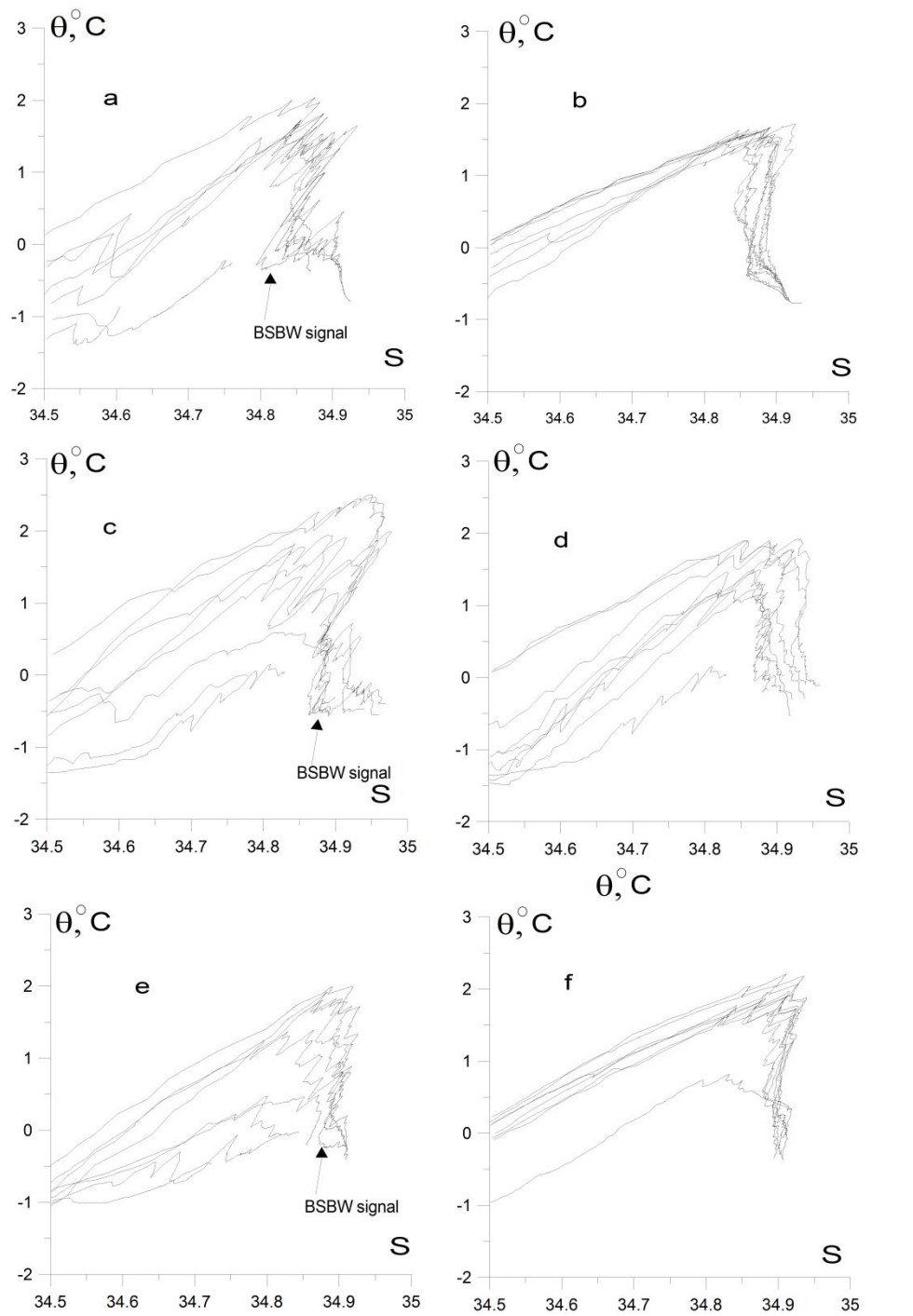

Fig. 9. *θ-S* diagrams based on the CTD profiling : NABOS-05: (a) and (b), 103°E (a), 126°E (b); NABOS-06: (c) and (d), 103°E (c), 126°E (d); NABOS-08: (e) and (f), 103°E (e), 126°E (f).

## 3.2 Characteristics of the Atlantic Water flow and geostrophic estimates of the volume flow rate

The estimates of the geostrophic volume flow rate and the hydrological parameters describing the AW flow in the Eurasian and Makarov Basins, are presented in Table 1. The geostrophic estimates of the near-bottom volume flow rate of the BSBW in zonal transects across the St. Anna Trough are presented in Table 2. The only exception is the transect at 82°N,

where the near-bottom gravity current with a considerable eastward component due to overflow across a sufficiently deep ridge (approx. 500 m deep) east of the St. Anna Trough (Fig. 3, top panels) makes the estimate of AW transport northward questionable. Note also that to the west of the St. Anna Trough our estimates refer to the FSBW; to east of this region BSBW enters the Eurasian Basin and our estimates should be attributed to the joint contribution of the two branches (FSBW and BSBW).

The hydrological parameters shown in Table 1 can be interpreted as follows. The maximum water temperature of the AW may exceed 5 °C in cases when the AW inflow to the Eurasian Basin consists of especially warm water masses. Typical changes in the temperature and salinity maxima of the AW moving along the slope over a distance of about 1000 km are approximately 1–2 °C and 0.1, respectively. These changes lead to a slight increase in potential density and therefore a deviation of the AW from the isopycnic distribution can be expected. These changes are most likely associated with the exchange of heat, salt, and mass with the surrounding waters through intrusive layering and double diffusion (see, e.g., Kuzmina et al., 2011; Polyakov et al., 2012; Kuzmina et al., 2018) and sea ice melting and cooling (Rudels, 1998). The intrusions, in particular, can also contribute to the reduction of the AW heat and salt content and the volume flow rate. The differences in the AW heat and salt content and the volume flow rate can be clearly seen from the PS-96 section when comparing data from stations near the continental slope of the Eurasian Basin at 92°E and from the vicinity of the Lomonosov Ridge at 140°E.

It is worth noting that the maximum value of the AW temperature ($\theta_{max}$) in this data set is always observed in the upper layer of the Eurasian Basin at depths below the pycnocline but not exceeding 350 m, while the maximum salinity ($S_{max}$) at sections in the eastern part of the Basin can be observed at depths greater than 1000 m.

$X_{\theta max}$ in Table 1 is the distance of the AW core (which can be associated with $\theta_{max}$) from the slope/shelf boundary. The highest value and the maximum variation of this parameter is observed near 126°E and 142°E, where a two-core structure of AW is often observed (Pnyushkov et al., 2015).

The noticeable increase of $\theta_{max}$ in 2006 at 31°E and 103°E and the intensive warming of the AW were first reported in (Polyakov et. al., 2011). The present results show that the increase of the temperature of the AW in 2006 was also accompanied by an increase of volume transport (see Table 1, the section along 103°E and reasonings below). This can be caused not only by the warming of the AW, but also by an increased inflow of the AW to the Eurasian Basin.

The geostrophic transport in the range of 31–159°E is characterized by a high variability (Table 1). This may be due to a) a section orientation oblique to the current; b) the difference in the horizontal scales of the sections; c) uncertainty in the choice of the reference level for geostrophic calculations; d) meandering of the flow; e) the effect of synoptic quasi-geostrophic eddies on the flow volume rate. In order to find statistically consistent estimates of the variability of geostrophic volume flow rate along the slope of the basin based on a limited data set, the following was done. The volume flow rates obtained for all sections within the range 31º−92 ºE for different years were used to calculate the mean volume flow rate (region I; the number of volume flow rate values averaged is $N = 6$). Similarly, the average volume flow rate was calculated for the region 94º−107ºE (region II; $N = 9$). The remaining average estimates of geostrophic volume flow rate were calculated for sections 126ºE (region III; $N = 9$), 142ºE (region IV; $N = 10$) and 159ºE (region V; $N = 2$). Then the 95% and 80% confidence intervals were determined using the Student t-distribution. All estimates of average volume flow rates and confidence intervals are presented in Tables 1 and 2.

On the average, the volume flow rate increases from region I to region II, then decreases to region III and region IV, followed by a sharp decrease in region V. However, only the difference between the volume flow rate in region II and the values in regions IV and V are significant at 95% confidence. Transport values bounded by the confidence intervals for regions II, IV and V are (0.46; 1.72), (0.12; 0.44) and (-0.37; 0.43), respectively. These intervals indicate that the mean volume flow rate in region II exceeds the value of the same parameter in regions IV and V with a high probability of 95%. The 80% confidence intervals overlap only for regions III and IV, (0.25; 0.53) and (0.18; 0.38), respectively. In this regard, the change in the volume flow rate along the slope is significant with a probability of 80%, except for changes in volume flow rate from region III to region IV.

The above values of the mean volume flow rate and confidence intervals also suggest that the increase in volume flow rate in 2006 is significant, and not caused by the "noise" in the data. Indeed, the volume flow rates in regions II, III, and IV in 2006 exceeded the upper limits of the corresponding 95% confidence intervals. From statistical point of view such a significant increase in volume flow rates at the same time in three regions is a very rare event that can hardly be explained by random "noise" in the data caused, for example, by the influence of synoptic eddies.

Let us turn our attention to the following features of the volume flow rate estimates: high volume flow rate estimates at 96°E, 103°E, 107°E, a negative volume flow rate estimate at 126°E in 2013 and low volume flow rate estimates at 31°E, 98°E in 2009 (Table 1). Indeed, the

AW volume flow rate in the BSBW area of entry into the Eurasian Basin in 2013 was almost equal to the maximum volume flow rate in 2006 (103°E) and was quite high up to the longitude 107°E. This phenomenon as well as the intense warming in 2006 can be associated with the recent changing conditions in the Arctic. We hypothesize that the negative volume flow rate at 126°E was because of the influence of local return flows which can be observed near the slope (Pnyushkov et al., 2015). Low FSBW volume flow rate estimates in 2009 are probably associated with a strong deviation of the flow from the slope, which may underestimate the AW volume transport due to the small length of the transects to the north (see also Section 4).

The mean value of the FSBW volume flow rate in region I is $V_{mean} = 0.5$ Sv. This estimate of volume flow rate is about half the estimate of the BSBW mean volume flow rate, $V_{mean} = 0.79$ Sv ($N = 3$, Table 2). (The difference is significant at 80% confidence interval). The BSBW mean volume flow rate exceeding nearly twice the FSBW mean volume flow rate results in a dominance of the BSBW pattern of potential density contours in the longitude range of 94–107°E (region II), where both branches of the AW are present. Moreover, the sum of the mean values of the FSBW and the BSBW volume flow rate geostrophic estimates $V_{mean} = 0.5 + 0.79 = 1.29$ Sv, corresponds well with the combined FSBW and BSBW flow within the region II: $V_{mean} = 1.09$ Sv. Thus, the increase in geostrophic transport in region II is mainly due to the influence of the BSBW. The decrease in geostrophic volume flow rate in region III can also be associated primarily with the BSBW, namely, with the decrease in the BSBW transport in the 126ºE section and further along the slope (see sect. 3.1.1 and 3.1.2).

Finally, at the 159°E section in the Makarov Basin, the geostrophic estimate of the along-slope volume flow rate of mixed waters of the FSBW and the BSBW has further greatly reduced down to $V_{mean} = 0.03$ Sv ($N = 2$), which is of more than one order of magnitude smaller than that in the Nansen and Amundsen Basins. Despite the low statistical significance of the latter estimate (due to small value of $N = 2$) one may conclude that the major part of the AW entering the Arctic Ocean circulates cyclonically within the Nansen and Amundsen Basins, and only its small part flows to the Makarov Basin (Rudels et al., 2015; Rudels, 2015). However, additional studies are required to confirm this result.

Table 1. Characteristics of the Atlantic Water flow in the course of its propagation along continental slope of the Eurasian Basin of the Arctic Ocean. *Dist* is the along-slope distance from the Fram Strait; $\theta_{max}$ is the maximum temperature; $\sigma_\theta$ ($Z_{\theta max}$), $S(Z_{\theta max})$, $Z_{\theta max}$, and $X_{\theta max}$ are the values of potential density, salinity, depth, and lateral displacement from the slope for the point $\theta_{max}$; $S_{max}$ and $Z_{Smax}$ are the maximum salinity and depth of $S_{max}$ ; $V$ is the geostrophic estimate of

the volume flow rate. The mean values and 95% / 80% confidence intervals of the volume rate, $V_{mean}$, calculated separately for CTD transects at 31−92°E, 94−107°E, 126°E, 142°E and 159°E, are also shown. The last row in the Table presents the characteristics of the return flow of the AW by the Lomonosov Rigde at the longitude 140°E and latitude 86.5°N (PS96, see Fig. 1). Year is given in the first column (e.g. NABOS06 corresponds to 2006).

| Exp | Lon [°E] | Dist [km] | $\theta max$ [°C] | $\sigma_\Theta(Z_{\theta max})$ [kg/m³] | $S(Z_{\theta max})$ | $Z_{\theta max}$ [m] | $X_{\theta max}$ [km] | $S_{max}$ | $Z_{Smax}$ [m] | V [Sv] |
|---|---|---|---|---|---|---|---|---|---|---|
| NABOS06 | 31 | 404 | 5.670 | 27.579 | 34.980 | 42 | -11 | 35.099 | 72 | 0.57 |
| NABOS08 | 31 | 404 | 4.883 | 27.771 | 35.103 | 101 | 0 | 35.105 | 176 | 0.80 |
| NABOS09 | 31 | 404 | 3.691 | 27.818 | 34.999 | 89 | 0 | 35.002 | 91 | 0.10 |
| NABOS09 | 60 | 856 | 2.503 | 27.891 | 34.951 | 175 | 10 | 34.981 | 363 | 0.47 |
| NABOS13 | 90 | 1290 | 2.600 | 27.903 | 34.975 | 250 | 41 | 34.996 | 333 | 0.46 |
| PS96 | 92 | 1322 | 2.786 | 27.875 | 34.960 | 271 | 33 | 34.968 | 329 | 0.58 |
| | | | | | | | | $V_{mean}$ = 0.50 ±0.24 / ±0.14 Sv | | |
| NABOS15 | 94 | 1355 | 2.445 | 27.946 | 35.012 | 331 | 33 | 35.015 | 365 | 0.47 |
| NABOS13 | 96 | 1388 | 2.548 | 27.902 | 34.969 | 207 | 70 | 34.978 | 264 | 2.06 |
| NABOS09 | 98 | 1421 | 2.300 | 27.906 | 34.948 | 220 | 79 | 34.971 | 345 | 0.09 |
| NABOS05 | 103 | 1561 | 2.029 | 27.870 | 34.876 | 179 | 39 | 34.934 | 309 | 0.32 |
| NABOS06 | 103 | 1561 | 2.528 | 27.888 | 34.950 | 220 | 50 | 34,978 | 260 | 2.23 |
| NABOS08 | 103 | 1561 | 1.980 | 27.886 | 34.891 | 201 | 60 | 34.929 | 325 | 0.42 |
| NABOS09 | 103 | 1561 | 1.984 | 27.913 | 34.925 | 244 | 50 | 34.951 | 365 | 0.87 |
| NABOS13 | 103 | 1561 | 2.278 | 27.904 | 34.942 | 215 | 80 | 34.956 | 419 | 1.59 |
| NABOS13 | 107 | 1695 | 1.903 | 27.937 | 34.945 | 359 | 120 | 34.948 | 404 | 1.77 |
| | | | | | | | | $V_{mean}$ = 1.09 ±0.63 / ±0.38 Sv | | |
| NABOS02 | 126 | 2104 | 1.406 | 27.938 | 34.902 | 324 | 243 | 34.932 | 2061 | 0.05 |
| NABOS03 | 126 | 2102 | 1.341 | 27.941 | 34.899 | 336 | 342 | 34.921 | 1886 | 0.41 |
| NABOS04 | 126 | 2102 | 1.770 | 27.906 | 34.896 | 271 | 87 | 34.925 | 2431 | 0.61 |
| NABOS05 | 126 | 2102 | 1.695 | 27.936 | 34.926 | 359 | 227 | 34.935 | 2841 | 0.75 |
| NABOS06 | 126 | 2102 | 1.905 | 27.923 | 34.930 | 284 | 193 | 34.960 | 968 | 0.77 |
| NABOS07 | 126 | 2102 | 2.085 | 27.907 | 34.928 | 266 | 242 | 34.942 | 340 | 0.60 |
| NABOS08 | 126 | 2102 | 2.195 | 27.885 | 34.911 | 206 | 235 | 34.939 | 365 | 0.31 |
| NABOS09 | 126 | 2102 | 1.907 | 27.909 | 34.913 | 316 | 33 | 34.932 | 1018 | 0.40 |
| NABOS13 | 126 | 2102 | 1.946 | 27.937 | 34.949 | 346 | 228 | 34.951 | 428 | -0.21 |
| NABOS15 | 126 | 2102 | 1.653 | 27.918 | 34.898 | 246 | 400 | 34.942 | 3816 | 0.22 |
| | | | | | | | | $V_{mean}$ = 0.39 ±0.22 / ±0.14 Sv | | |
| NABOS03 | 142 | 2456 | 1.089 | 27.912 | 34.841 | 269 | 41 | 34.862 | 1000 | 0.06 |
| NABOS04 | 142 | 2456 | 1.401 | 27.909 | 34.865 | 281 | 0 | 34.907 | 1608 | 0.21 |
| NABOS05 | 142 | 2456 | 1.492 | 27.906 | 34.870 | 284 | 100 | 34.906 | 1550 | 0.26 |
| NABOS06 | 142 | 2456 | 1.981 | 27.874 | 34.876 | 234 | 111 | 34.960 | 1016 | 0.60 |
| NABOS07 | 142 | 2456 | 1.855 | 27.879 | 34.870 | 231 | 0 | 34.920 | 2064 | 0.09 |
| NABOS08 | 142 | 2456 | 1.599 | 27.915 | 34.890 | 260 | 200 | 34.908 | 347 | 0.23 |
| NABOS09 | 142 | 2456 | 1.704 | 27.915 | 34.900 | 253 | 101 | 34.917 | 1082 | 0.22 |
| NABOS13 | 142 | 2456 | 1.475 | 27.940 | 34.909 | 331 | 115 | 34.926 | 1150 | 0.18 |
| NABOS15 | 142 | 2456 | 1.353 | 27.936 | 34.892 | 326 | 106 | 34.913 | 1372 | 0.63 |
| | | | | | | | | $V_{mean}$ = 0.28 ±0.16 / ±0.10 Sv | | |
| NABOS07 | 159 | 2783 | 1.424 | 27.887 | 34.839 | 255 | 0 | 34.880 | 1075 | -0.01 |
| NABOS08 | 159 | 2783 | 1.383 | 27.893 | 34.843 | 245 | 0 | 34.889 | 1266 | 0.06 |
| | | | | | | | | $V_{mean}$ = 0.03 ±0.40 / ±0.10 Sv | | |
| PS96back | 140E 86.5N | 3178 | 1.812 | 27.890 | 34.880 | 219 | ≈ 700 | 34.902 | 472 | -0.09 |


Table 2. Geostrophic estimates of the volume flow rate for near-bottom gravity flow of the Barents Sea Branch of Atlantic Water (BSBW) on zonal transects across the St. Anna Trough. The uncertainty estimates are 95% and 80% confidence intervals.

| Exp | NABOS09 | NABOS13 | NABOS15 | |
|---|---|---|---|---|
| Lat [°N] | 81.00 | 81.33 | 81.41 | $V_{mean}$ |
| V [Sv] | 0.89 | 0.73 | 0.76 | 0.79 ±0.22 / ±0.10 |


### 3.3 Interannual variability of the AW temperature-salinity values and the volume flow rate

Within the NABOS project, the cross-slope CTD transects at 103°E, 126°E, and 142°E were repeatedly performed for a number of annual campaigns (Table 1): 2005, 2006, 2008 and 2013 (103°E), 2002–2009, 2013 and 2015 (126°E), 2003–2009, 2013, and 2015 (142°E). We use

the repeated transects to describe the inter-annual variability of the AW.

Time series of the AW temperature maximum, $\theta_{max}$ , and the related values of salinity $S(\theta_{max})$ and potential density anomaly $\sigma_{\theta}(\theta_{max})$ (Fig. 10) show that the period of 2006 to 2008 was characterized by an increased temperature of the AW in the eastern part of the Eurasian Basin, an increased salinity and density reduction. The temperature excess during this period was as large

as 0.6–1.0 °C relative to 2002–2003 and 0.3–0.6 °C relative to 2013–2015. $S(\theta_{max})$ displayed in 2006 local maxima at the transects 126°E and 142°E, and the absolute maximum at the transect 103°E; the salinity excess for the maxima largely decreased with the longitude from approximately 0.06 at 103°E to less than 0.01 at 142°E. $\theta_{max}$ had a maximum in 2013 but only at 103°E (see Table 1 and Fig.10). The time series of $S(\theta_{max})$ display a trend of increase of AW

salinity over  time, that can be referred to as a AW salinization in early 2000s. The salinity of AW at 142°E increases almost monotonously in the period from 2003 to 2013. The mechanism behind this salinity evolution is not clear. It is also worth noting that the maxima of $\theta_{max}$ and $S(\theta_{max})$ in 2006 and 2013 (at 103°E) were accompanied by maxima in transport.

### 4 Discussion

Here we discuss the following issues: a) differences in the identification of the BSBW; b) a comparison of the geostrophic volume flow rate estimates with other studies; c) the weakening of the BSBW signal at 126 ºE and further east.

a) Advection and interaction of waters with different $\theta$-$S$ characteristics in the Arctic Basin, as well as the impact of climate change that has been observed over the past decade

(Polyakov et al., 2017) complicate an accurate identification of water masses. However, a robust approach proposed in Dmitrenko et al. (2015), is effective for distinguishing the water masses of

the FSBW and BSBW branches. As an exception, this approach fails when the FSBW temperature is below 0 °C (see Fig. 2 in Dmitrenko et al., 2015), and/or the BSBW temperature is close to 1 °C (see Fig. 6 in Schauer et al., 2002a). If such cases are rare, then either of the two approaches can be used to identify the BSBW and FSBW. Indeed, the identification of the BSBW on the PS-96 section in our case (we used the approach proposed by Dmitrenko et al., 2015; see paragraph 3.1.1) does not differ much from that proposed by Schauer et al. (2002b). However, these discrepancies can lead to almost an order of magnitude difference in estimates of the volume flow rate of the BSBW only due to the differences in the BSBW cross-sectional area.

b) Based on the velocity measurements with moored instruments (1997−2010) in the area of the West Spitsbergen Current (WSC) near Fram Strait (zonal transect at ~78º50ʹ N), approximately 3 Sv of the AW flows into the Nansen Basin (Beszczynska-Möller et. al., 2012). The long-term mean volume transport confined to the WSC core branch (or Svalbard branch in accordance with Schauer et al., 2004) included 1.3±0.1 Sv of the AW warmer than 2ºC. The offshore WSC branch (or Yermak branch) carried on average 1.7±0.1 Sv of the AW. The variability range of the AW geostrophic transport of the Svalbard branch for meridional sections from 1997, 2001, and 2003 (summer/fall) was between 0.06 Sv and 0.7 Sv (Marnela et al., 2013). In Kolås and Fer (2018) observations of the oceanic current and thermohaline field (in summer 2015) in the three sections were used to characterize the evolution of the WSC along 170 km downstream distance. Absolute geostrophic transports of AW ranged from 0.6 Sv to 1.3 Sv in the Svalbard branch. In accordance with earlier studies of the currents in Fram Strait, recirculation of the AW can be significant, and the volume flow rate of the AW entering the Arctic Ocean ranges from 0.6 Sv to 1.5 Sv (Rudels, 1987; Aagaard and Carmack, 1989).

Our estimate of the mean volume flow rate $V_{mean}$ in region I (31º−92 ºE) is in the range of the above estimates. However, the upper confidence limit of our estimate does not reach 1 Sv. Moreover, we used T> 0ºC to identify the AW while in Beszczynska-Möller et al. (2012) the volume flow rates of the AW entering the Eurasian Basin through Fram Strait were determined for waters with T > 2 °C. Comparatively smaller transport in region I may be because the sections along 31°E (see Fig. 1) are less than 100 km wide and do not cover the full extent of the FSBW (Fig. 2, upper panel). Given the sensitivity to the definition of AW and the resulting cross-sectional area (see point "a" above), the volume transport may be underestimated. It is possible that the formation and passage of synoptic eddies leads to variability in transport rates. According to Perez-Hernandez et al. (2017) north of Svalbard (between 21 and 33°E) in September, 2013, a large difference was found in the estimates of geostrophic volume flow rate (from 0.53 Sv to 3.39 Sv) due to the passage of eddies and meandering of the current. Våge et al.

(2016) based on geostophic velocities at two CTD sections across the boundary current near 30º E (September, 2012) evaluated a net AW volume flow rate of 1.6±0.3 Sv. They found evidence of a large eddy affecting the mean volume transport calculations.The barotropic velocity component, which is not taken into account in our estimates, can also contribute to larger transports. However, in conditions with high ice concentration in the Eurasian Basin, we might expect a reduced barotropic contribution from the sea level changes induced by wind forcing. In cruise reports, the NABOS CTD sections were characterized by ice concentrations of 50−100% (see https://uaf-iarc.org/nabos-cruises/). Exceptions occurred in the near-slope areas of the Laptev Sea, that is, in the sections along ~ 126ºE, where the ice concentration varied from 0 to 100%, having a maximum value in the northern part of the sections. In such areas, the contribution of the barotropic component to the flow velocity can be large. For example, using long-term measurements (1995 − 1996) from a mooring in the near-slope area of the Laptev Sea, Woodgate et al. (2001) showed that the contribution of the barotropic component to the velocity of the Arctic Ocean boundary current (AOBC) was equal to the contribution of the first three baroclinic modes. Assuming an average velocity based on the measurements in the upper 1200 m layer of 4.5 cm/s and a width of 50 to 84 km the volume flow rate was estimated at $5 \pm 1$ Sv. This is larger than our average estimate of the AW volume flow rate along 126 ºE (0.39±0.22, Table 1) by an order of magnitude. Such a difference can be explained not only by the absence of a barotropic contribution in our case, but also by the fact that we took into account the volume transport of AW only (i.e. the cold, low-salinity surface layer was excluded) and considered certain season (August and September). Indeed, according to long-term measurements at 6 moorings on a section along 126 ºE, the AOBC volume flow rate varied from 0.3 Sv to 9 Sv (Pnyushkov et al., 2018 b). Such a wide range in volume flow rate estimates is probably due to a combined effect of seasonal variability and mesoscale eddies (Pnyushkov et al., 2018 a).

The fact that seasonal variations can in some cases significantly affect the AW volume flow rates (see also the discussion in Pnyushkov et al., 2018 b) is confirmed by a number of observations (Schauer et al., 2002a; Beszczynska-Möller et al., 2012; Pnyushkov et al., 2018 b). For example, the volume flow rate of the AW in the northwestern part of the Barents Sea was 0.6 Sv (Schauer et al., 2002a). This agrees well with our estimate of the AW transport in the St. Anna Trough, $0.79 \pm 0.22$ Sv (Table 2). However, the analysis of current velocity measurements in the winter season at the same section in the northwestern part of the Barents Sea gave a completely different estimate of ~ 2.6 Sv (Schauer et al., 2002a).

c) According to Dmitrenko et al. (2009), the BSBW can be satisfactorily identified at 142°E. However, a "pattern" in the $\theta$-$S$ diagram far from the place of the BSBW entry into the Eurasian Basin can be regarded as the BSBW signal, if it maintains the similarity with the

"pattern" of the BSBW at the exit from the St. Anna Trough, that is, with the so-called "knee" (Dmitrenko et al., 2015). Our analysis showed that the "knee" is regularly observed at 103°E, while at 126°E it is absent, weak or distorted. This may be expected since the flow velocity is small, and the BSBW covers a distance from 103°E to 126°E for 1−2 years. However, despite of such a long travel time, Fram Strait branch is well identified not only at 126°E, but also further along the slope. This suggests stronger transformation and mixing of, primarily, the BSBW. The BSBW transformation can be due to various reasons, including mixing with the FSBW caused by thermohaline intrusive layering at absolutely stable stratification (Merryfield, 2002; Kuzmina et al., 2014; Kuzmina, 2016), the influence of the slope topography, the impact of local counterflows near the slope (see, for example, Pnyushkov et al., 2015), lateral convection (Ivanov and Shapiro, 2005; Ivanov and Golovin, 2007; Walsh et al., 2007), the impact of the Arctic Shelf Break Water (Aksenov et al., 2011; Ivanov and Aksenov, 2013) and  mixing due to eddies (Schauer et al., 2002; Dmitrenko et al., 2008; Aagaard et al., 2008; Pnyushkov et al., 2018a ). The understanding of the processes of transformation and mixing of the BSBW and FSBW is necessary to verify an important concept proposed by Rudels, et al. (2015) that the BSBW supplies the major part of the AW to the Amundsen, Makarov and Canadian Basins, while the FSBW remains almost fully in the Nansen Basin.

## 5 Summary

The $\theta$-$S$ properties and the volume flow rate estimates of the current carrying the AW in the Eurasian Basin and St. Anna Trough were obtained based on the analysis of CTD data collected within the NABOS program in 2002–2015; additionally CTD transect PS-96 was considered.

FSBW was present at all transects, including the two transects in the Makarov Basin (159°E), while the cold waters at the transects along longitudes 126°E, 142°E and 159°E, which can be associated with the influence of the BSBW, were observed in the depth range below 800 m and had little effect on the spatial structure of isopycnic surfaces and horizontal gradient of density. It is shown using $\theta$-$S$ analysis that the BSBW signal, which is characterized by the knee-shape feature in coordinates $\theta, S$ and $\sigma_\theta, S$ (see Fig.8), is either strongly weakened or not visible at the longitude 126°E (excluding the observations in 2002 at 126 °E), while the FSBW signal is well identified at 126ºE and further along the slope of the Eurasian Basin. Based on the revealed features of the temperature, salinity and density fields, it is suggested that east of 126ºE the geostrophic volume transport of AW is mainly provided by the FSBW.

The geostrophic volume flow rate of AW increases (with 80% confidence) from the region of 31ºE−92ºE (0.5 ± 0.14 Sv) to the region of 94ºE−107ºE (1.09 ± 0.38 Sv), then decreases to the region of 126ºE (0.39 ± 0.14 Sv) and becomes small (0.03 ± 0.1 Sv) in the Makarov Basin (159ºE).

The temporal variability of hydrological parameters and of the AW volume flow rate is summarized as follows. The time series of $\theta_{max}$ had an absolute maximum in 2006–2008 that can be interpreted as a result of heat pulse in the early 2000s (Polyakov et al., 2011). In accordance with our analysis the time series of $\theta_{max}$ had a maximum in 2013 but only at the longitude 103°E (Table 1 and Fig.10). The time series of $S(\theta_{max})$ display a trend of increase of AW salinity over time, that can be referred to as a AW salinization in early 2000s. Moreover the salinity increases almost monotonously in the period from 2003 to 2013 at 142ºE. It is important to underline also that the maxima of $\theta_{max}$ and $S(\theta_{max})$ in 2006 and 2013 (103°E) are accompanied by the volume flow rate highs. A significant increase in geostrophic volume flow rate identified in 2006 is shown to be caused by climate impact.

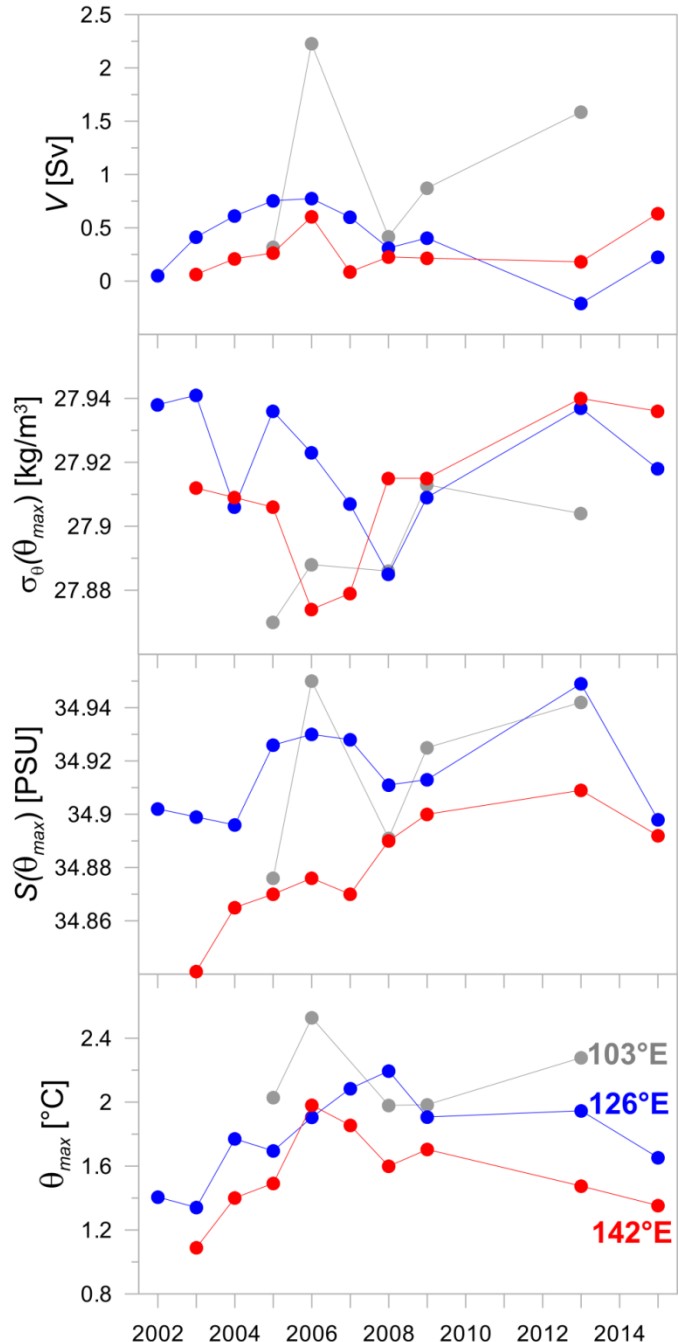

Fig. 10. Interannual variability of the maximum temperature $\theta_{max}$ and the related values of salinity $S(\theta_{max})$, potential density anomaly $\sigma_\theta(\theta_{max})$ and volume flow rate $V$ on the cross-slope transects at 103°E, 126°E and 142°E.

*Acknowledgments*. This research, including the approach development, data processing and interpretation, performed by Nataliya Zhurbas, was funded by Russian Science Foundation, project no. 17-77-10080. Natalia Kuzmina ($\theta$-$S$ analysis, statistical analysis, participation in discussion) was supported by the state assignment of the Shirshov Institute of Oceanology RAS (theme no. 0149-2019-0003).

The authors are very grateful to the NABOS group for providing the opportunity to use the CTD-data.

The authors are very grateful to the editor for evaluating the article and help in the work on the text and anonymous reviewers for useful comments.

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

# Variability of the thermohaline structure and transport of Atlantic water in the Arctic Ocean based on NABOS ~~CTD~~ hydrography data

Nataliya Zhurbas[1] and Natalia Kuzmina[1]

[1]Shirshov Institute of Oceanology, Russian Academy of Sciences, 36 Nakhimovsky Prospekt , 117997 Moscow, Russia

*Correspondence to:* Nataliya Zhurbas (nvzhurbas@gmail.com)

**Abstract.** ~~CTD~~ Conductivity-temperature-depth (CTD) transects across continental slope of the Eurasian Basin and the St. Anna Trough performed during NABOS (Nansen and Amundsen Basins Observing System) project in 2002–2015 and a transect from the Polarstern-1996 expedition are used to describe ~~$\theta$-S~~ the temperature and salinity characteristics and volume flow rates of the current carrying the Atlantic Water (AW) in the Arctic Ocean. ~~The CTD dataset includes 33 sections in the Eurasian Basin, 4 transects in the St. Anna Trough and 2 transects in the Makarov Basin; additionally a CTD transect of the Polarstern-1996 expedition (PS-96) is used.~~ The variability of ~~thermohaline pattern on~~ the AW on its pathway along the slope of Eurasian Basin is investigated. ~~The~~ A dynamic Fram Strait branch of the Atlantic Water (FSBW) is identified on all transects, including two transects in the Makarov Basin (along 159°E), while the cold waters on the eastern transects along 126°E, 142°E and 159°E , which can be associated with the influence of the Barents Sea branch of the Atlantic water (BSBW), ~~on the transects along 126°E, 142°E and 159°E,~~ were observed in the depth range below 800 m and had a negligible effect on the spatial structure of isopycnic surfaces. ~~An interpretation of the spatial and temporal variability of hydrological parameters characterizing the flow of the AW in the Eurasian Basin is presented.~~ The geostrophic volume transport of AW decreases farther away from the areas of the AW inflow to the Eurasian Basin, decreasing by one order of magnitude in the Makarov Basin at 159°E, implying that the major part of the AW entering the Arctic Ocean circulates cyclonically within the Nansen and Amundsen Basins. There is an absolute maximum of $\theta_{max}$ (AW core temperature) in 2006–2008 time series and a maximum in 2013, but only at 103°E. Salinity $S(\theta_{max})$ (AW core salinity) time series display an increase of the AW salinity in 2006–2008 and 2013 (at 103°E) that can be referred to as a AW salinization in the early 2000s. The maxima of $\theta_{max}$ and $S(\theta_{max})$ in 2006 and 2013 are accompanied by the volume transport maxima. The time average geostrophic volume transports of AW, ~~$V_{mean}$ , are $V_{mean} = 0.5 Sv$~~ are 0.5 Sv in the longitude range 31–92°E, ~~$V_{mean} = 0.8$ Sv~~ 0.8 Sv in the St. Anna Trough and ~~$V_{mean} = 1.1$ Sv~~ 1.1 Sv in the longitude range 94–107°E.

## 1 Introduction

Atlantic water (AW) enters the Eurasian Basin ~~by~~ in two branches (see, e.g., Aagaard, 1981; Rudels et al.,1994; Schauer et al., 1997; Rudels et al., 1999; Schauer et al., 2002a, b; Rudels et al., 2006; Berzczynska-Möller et al., 2012; Rudels et al., 2015; Rudels, 2015; Dmitrenko et al., 2015; Pnyushkov et al., 2015, 2018b): one ~~of them~~ branch originates from the Greenland and Norwegian seas and flows to the basin through ~~the~~ Fram Strait (the Fram Strait branch of the Atlantic Water, hereinafter the FSBW), and the other reaches the deep part of the Arctic Ocean near St. Anna Trough after passing through the Barents Sea (the Barents Sea branch of the Atlantic water, hereinafter the BSBW). After entering the Eurasian Basin the FSBW moves eastward with ~~the~~ a subsurface boundary current and has a core of higher temperature and salinity than the BSBW. In the longitude range of 80–90°E, it encounters and partially mixes with the BSBW, which is strongly cooled due to mixing with shallow waters of the Arctic shelf seas and atmospheric impact (Schauer et al., 1997; 2002a, b). Further, the water masses resulting from the interaction of the two branches ~~which transport the AW continue~~ spread~~ing~~ cyclonically in the Eurasian Basin.

~~To study the characteristics of the FSBW and BSBW flow in the Eurasian Basin, it is useful to estimate, first of all, its volume flow rate in different parts of the basin. Generally the estimates of the AW volume flow rate have been based on direct current observations (Fahrbach et al., 2001; Berzczynska-Möller et al., 2012; Rudels et al., 2014; Pnyushkov et al., 2015). However, it is useful also to consider the AW geostrophic volume flow rate calculated on the basis of CTD data. Such estimates, obtained for different regions of the Arctic Ocean, were given in a number of papers (e.g. Marnela et al., 2013; Våge et al., 2016; Pérez-Hernández et al., 2017; Kolås and Fer, 2018). For completeness, it is of interest to carry out estimates of the AW geostrophic volume flow rate along continental slope of the Eurasion Basin based on a large volume of empirical data.~~

Within the NABOS (Nansen and Amundsen Basins Observing System) project (Polyakov et al., 2007) a unique volume of CTD data was collected: more than 30 sections were made in various regions of the Arctic Basin in the summer/fall 2002-2015. A number of sections in different years were made in the same regions of the Basin, which allows studying the interannual variability of the water masses thermohaline structure and the geostrophic volume flow rate in these areas.

The main goal of this work is to investigate the spatial and temporal variability of the AW geostrophic volume flow rate during its propagation along the continental slope of the Eurasian Basin. ~~Another important aspect of our analysis is the investigation of~~ We further discuss the

thermohaline structure and transformation of the FSBW and BSBW. ~~Such analysis is essential for two reasons: a) t~~ The estimates of the AW transport are sensitive to the temperature and salinity ranges used for the identification of this water (Pnyushkov et al., 2018b)~~; b) it is reasonable to assume that~~ , and mixing of FSBW, BSBW and surrounding waters may change the AW geostrophic volume flow rate.

## 2 Material and Methods

We used data ~~of~~ from the CTD ~~profiling on~~ transects across the slope of the Eurasian Basin in the longitude range of 31–159°E measured in the years 2002–2015 within the framework of NABOS project (in total 39 transects). The data are freely available at the site http://nabos.iarc.uaf.edu. In addition, a CTD transect across the entire Eurasian Basin and over the Lomonosov Ridge starting at 92°E at the slope from R/V *Polarstern* in 1996 (hereafter PS96) was also included. The locations of the CTD transects are shown in Fig. 1. ~~It can be seen from the map in Fig. 1 that m~~ Most of the ~~CTD~~ transects are aligned cross-slope and grouped at longitudes of 31, 60, 90, 92, 94, 96, 98, 103, 126, 142, and 159°E. Four of the 40 transects crossed zonally the St. Anna Trough (at the latitude of 81, 81.33, 81.42, and 82°N) through which the BSBW enters the Eurasian Basin. Most of the CTD casts covered the upper layer from the sea surface to either 1000 m depth or to the bottom (if the total depth ~~of the sea~~ was shallower ~~less than 1000 m); some of the CTD casts (a. A~~pproximately every third or fourth~~) covered the depths from the sea surface~~ cast was down to the sea bottom even if the sea depth exceeded 1000 m.

To estimate the ~~strength of the FSBW or the BSBW or both branches~~ volume transport of the Atlantic Water, we applied standard dynamical method. The no-motion level (the depth of zero velocity ~~depth~~) was determined from the following consideration. If the baroclinic current occupies the upper layer or/and some intermediate layer, the no-motion level can be chosen in a calm deep layer (where the horizontal density gradient is relatively small). On the contrary, in case of a near-bottom gravity flow, the no motion level can be reasonably chosen ~~somewhere~~ well above the near-bottom flow. We adopted for the level of no-motion ~~level~~ either 1000 m depth or the sea bottom depth if the latter was smaller than 1000 m for the FSBW, and ~~some level in the vicinity of~~ approximately 50 m ~~depth~~, where density contours were more or less flat, for the observations of BSBW in the St. Anna Trough (see also below).

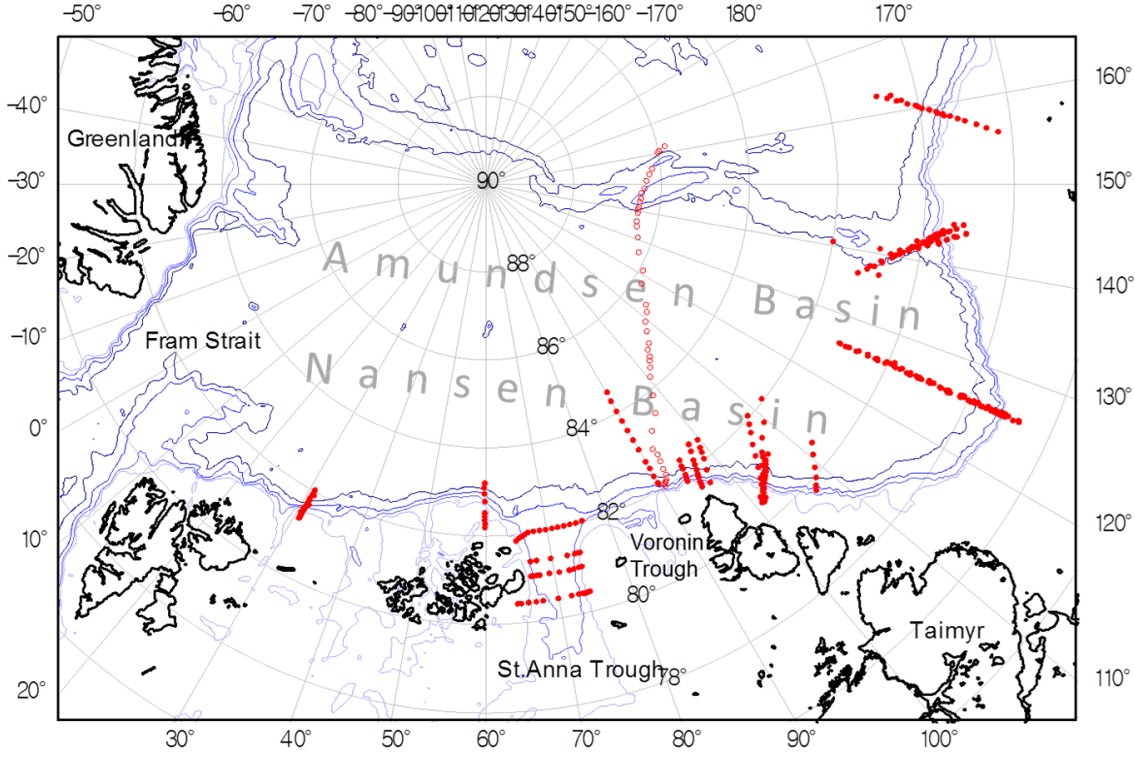

Fig. 1. Bathymetric map of the Eurasian Basin with 300, 500, 1000, and 2000 m contours shown. The red filled and blank circles are the locations of CTD stations on the NABOS and PS96 transects, respectively.

Since the FSBW brings saline and warm water to the Eurasian Basin, the geostrophic ~~estimates of the volume flow rate were~~ transport was found by integration over the depth range with positive temperature, $\theta > 0$ °C, and relatively high salinity, $S > 34.5$ (the salinity is given in the practical salinity scale), that is, ~~some areas in the~~ near-surface layers with warm and fresh water (which cannot be attributed to AW) were excluded. For the observations of BSBW in the St. Anna Trough the geostrophic ~~estimates of the volume flow rate were found~~ transport was calculated by integration over a depth range with ~~the non-averaged~~ temperature below 0 °C and ~~the~~ salinity above 34.5. If both AW branches ~~of AW~~ were present on the transect, the integration was performed over the entire depth range ~~except~~ but excluding the cold near-surface layer ($\theta < 0$ °C) and the ~~areas in the near-surface layer with~~ warm ($\theta > 0$ °C) and relatively fresh ($S < 34.5$) ~~water~~ near-surface layer. The zero velocity depth in this case was chosen ~~in accordance to~~ after inspection of the observed pattern of density contours, i.e. ~~its resemblance with~~ suggesting either the near-surface flow pattern or the near-bottom flow pattern (see Section 3). The details and limitation of the geostrophic velocity calculations are discussed in Zhurbas (2019).

## 3. Results

### 3.1 Variability of the thermohaline pattern on the AW pathway along the slope of Eurasian Basin

#### 3.1.1 CTD transects analysis

The transformation of thermohaline signatures (i.e. patterns of salinity $S$, potential temperature $\theta$, and potential density anomaly $\sigma_\theta$, versus cross-slope distance and depth) of the AW flow on its pathway along the slope of the Eurasian Basin are presented in Fig.2. The $\sigma_\theta$ contours on transects at 31°E diverge towards the continental slope margin (to the south), shallowing above the warm/saline core of the AW and sloping down beneath it associated with a

eastward subsurface flow. Such ~~a structural feature of the~~ distribution of isopycnic surfaces was observed on all NABOS transects taken across available continental slope at 31°E. According to Fig. 2 the warm/saline core of the Fram Strait Branch of the AW with the maximum temperature $\theta max$ of 4.88°C at the depth $Z_{\theta max}$=102 m and the maximum salinity $Smax$ of 35.11 at the depth $Z_{Smax}$=176 m is found on the slope at about 1000 m isobath.

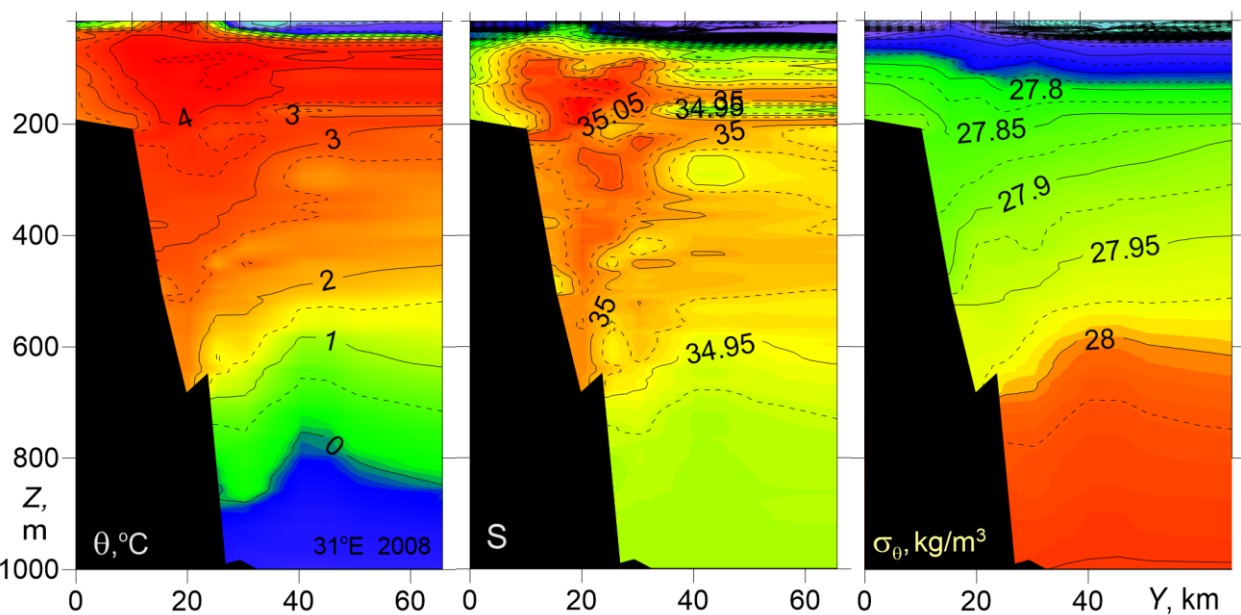

Fig. 2. Temperature $\theta$, salinity $S$, and potential density anomaly $\sigma_\theta$ versus cross-slope distance and depth for the NABOS-2008 transect across the Eurasian Basin slope at 31°E.

       Figure 3 presents temperature, salinity, and potential density ~~versus distance and depth~~ for two zonal transects across the St. Anna Trough at latitudes of 81 and 82°N. A stable pool of cold

($\theta$ <0°C) and dense ($\sigma_\theta > 28$ kg/m³) water in the bottom layer is seen adjacent to the eastern slope of the trough. The transfer of the densest water pool to the eastern slope corresponds to a geostrophically balanced near-bottom gravity flow to the North. ~~Note, that the gravity bottom currents are a typical feature of ocean dynamics and can develop in the narrows and troughs of various ocean basins (Arneborg et al., 2007; Zhurbas et al., 2012).~~ This near-bottom gravity

current carries also waters of Atlantic origin, which are strongly cooled due to mixing with ~~shallow waters of the Arctic shelf seas~~ shelf waters in (the Barents and Kara seas~~)~~. Above the near-bottom gravity flow of the BSBW one can observe two-core structure of warm FSBW with temperature up to 2.5 °C that enters the St. Anna Trough from the north-west at the western side of the trough and leaves it for the north-east at the eastern side of the trough. At 82°N, the

BSBW overflows a ridge-like elevation east of the St. Anna Trough (top panels in Fig. 3). ~~Results of s~~ Studies of the currents ~~velocities and thermohaline characteristics of the waters masses~~ and hydrography in the St. Anna Trough can be found in (Schauer et al., 2002a, b; Rudels et al., ~~2014~~ 2015; Dmitrenko et al., 2015).

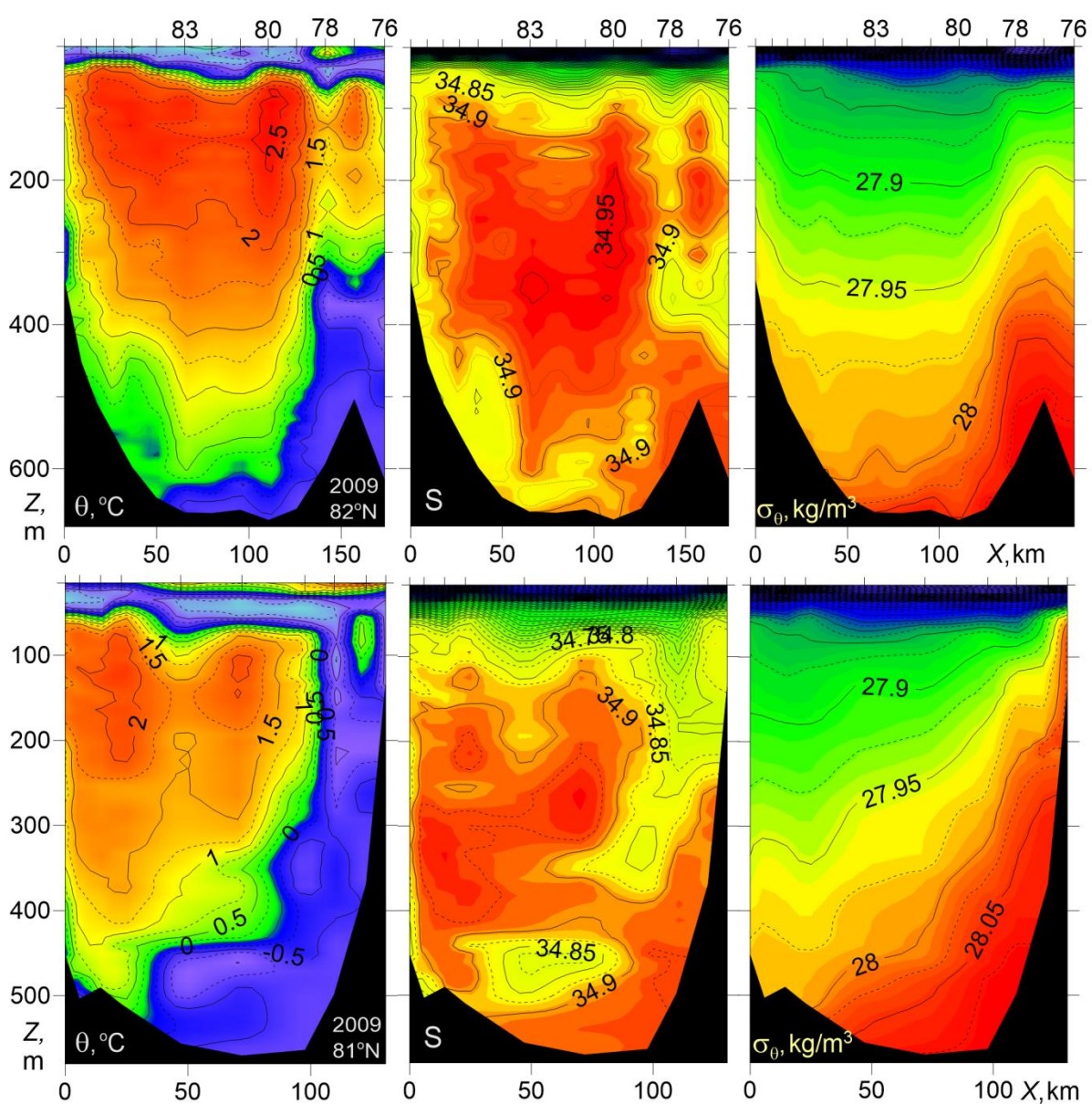

Fig. 3. Temperature θ, salinity S, and potential density anomaly $\sigma_\theta$ versus distance and depth for zonal transects across the St. Anna Trough at latitudes of 81°N (bottom, NABOS-2009), and 82°N (top, NABOS-2009). The X-axis is directed to the east.

In order to understand the effect of the FSBW and the BSBW transformation on geostrophic volume flow rate, it is necessary to identify water masses of different origin. For that purpose the following criterion is often used (Walsh et al., 2007; Pfirman et al., 1994): the water masses of the FSBW are characterized by $\theta > 0$ °C, and the BSBW can be identified by ~~the following expressions:~~ $-2$ °C $< \theta < 0$ °C, $34.75 < S < 34.95$ and $27.8$ kg/m$^3$ $< \sigma_\theta < 28.0$ kg/m$^3$. Other approaches to define BSBW are given in Schauer et al. (1997; 2002a, b) and Dmitrenko et al. (2015). According to Schauer et al. (1997; 2002a, b) the BSBW includes all waters that enter the Nansen Basin from the St. Anna and Voronin troughs. The temperature of these waters, however, can reach ~1 °C. The justification for this approach was based on $\theta$-$S$ analysis of the waters of the north-eastern part of the Barents Sea and the St. Anna and Voronin troughs. According to Dmitrenko et al. (2015), the BSBW consists of two water masses, and the temperature of the warmer water mass can only slightly exceed 0 °C (for more details see section 3.1.2). Here we will rely on the definitions of the FSBW and BSBW proposed by Dmitrenko et al. (2015).

In Fig. 4 the CTD transect at 92°E carried out in the *Polarstern*-1996 expedition just east of the entrance point of the BSBW to the Eurasian Basin from the St. Anna Trough and Voronin Trough is presented. It can be assumed that a part of the BSBW extends deep into the Basin, mixing with the FSBW, while another part of the BSBW ~~moves~~ flows eastward along the slope according to the general cyclonic circulation observed in the Eurasian Basin. On the presented transect the BSBW is observed in the depth range below 600 m as a narrow, about 10 km wide strip of cold water near the slope (see also Subsection 3.1.2) adjacent to a 300 km wide zone occupied by the warm FSBW. The ~~pattern of the~~ potential density distribution of FSBW on this transect is similar to transects at 31°E. Namely, despite of the masking effect of vertical undulations of $\sigma_\theta$ contours caused by internal waves and mesoscale eddies (one of subsurface, intra-pycnocline eddies is probably identified at the distance of $Y$=510 km), ~~one cannot miss the tendency of shallowing/sloping down the $\sigma_\theta$ contours~~ isopycnals tend to shoal/deepen above/below the FSBW core towards the continental slope margin (to the south) which, in terms of geostrophic balance implies the eastward flow of FSBW. The FSBW core on the 92°E transect is found at 40 km distance from the slope, with the maximum temperature $\theta max$=2.79°C at $Z_{\theta max}$=271 m and salinity $Smax$=34.97 at $Z_{Smax}$=329 m. Therefore, the FSBW on its pathway along the slope of the Eurasian Basin from 31°E to 92°E has cooled, desalinated, sank and become denser by ~~approx.~~ about 2 °C, 0.1, 150 m, and 0.1 kg/m$^3$, respectively. Another ~~significant~~ distinct feature ~~seen~~ in the PS96 transect is ~~an~~ a layer with increased temperature ~~pool~~

in the layer of ~~between~~ 180 ~~and~~ 300 m ~~at the distance of~~ depth at $Y$=600–750 km in the vicinity of the Lomonosov Ridge, which can be attributed to the geostrophically-balanced FSBW return flow cyclonically circulating around the Eurasian Basin (Rudels et al., 1994; Swift et al., 1997).

According to Schauer et al. (2002b) ~~where the thermohaline structure along~~ who studied the PS-96 section ~~was studied in detail~~, the horizontal and vertical scales of the BSBW were taken at 30 km and 800 m, respectively. This differs from our interpretation based on the definition of BSBW with temperature less than 0 °C.

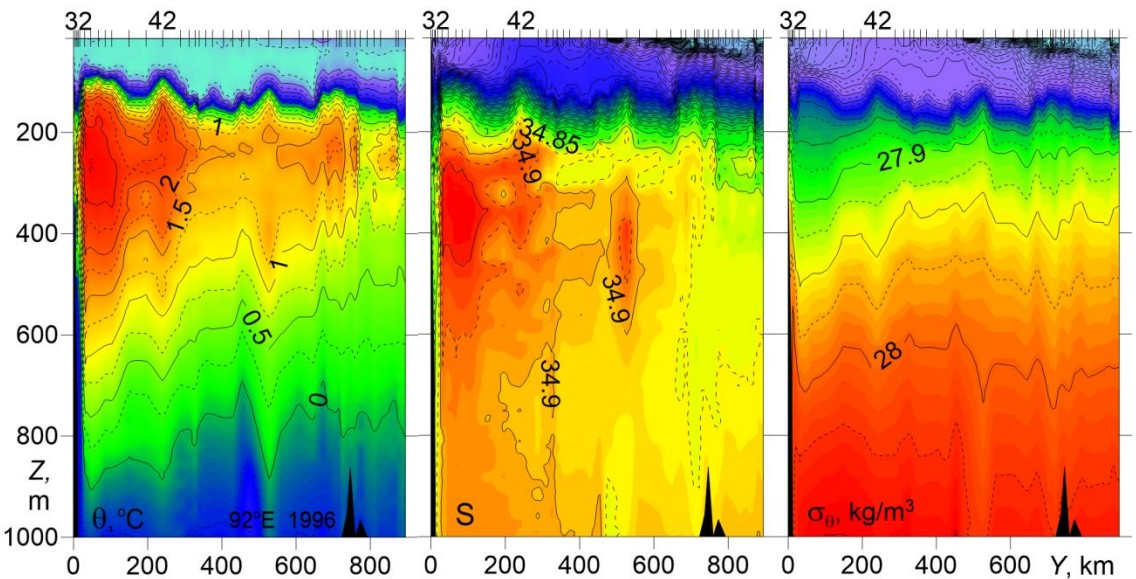

Fig. 4. Temperature $\theta$, salinity $S$, and potential density anomaly $\sigma_\theta$ versus distance and depth for cross-shelf transects at 92°E (PS-1996).

Further east, in the longitude range of 94–107 °E (NABOS-09), the denser part of BSBW ~~dives~~ under the FSBW~~,~~ is characterized by an eastward geostrophic current with isopycnals sloping towards the North in a 150 km wide zone adjacent to the slope (see Fig. 5, top panel). Less saline water at the slope is the less dense ~~Barents Sea Branch water~~ BSBW that has entered the Nansen Basin when the slope narrows north of Severnaya Zemlya (Schauer et al., 1997).

The vertical location of the FSBW layer ~~has not changed much relative~~ is similar to the 92°E in the section PS-96 but the maximum temperature has further decreased: in the transect in Fig. 5, the top panel, $\theta max$=1.98 °C at $Z_{\theta max}$=245 m and $Smax$=34.95 at $Z_{Smax}$=365 m. The bottom panel of Fig. 5 presents the ~~data from~~ transect at 142°E (NABOS-09) which is located on the Lomonosov Ridge, between the Amundsen and Makarov Basins. The comparison of the two transects obtained in the same year shows that the vertical scale of the ~~especially~~ warm FSBW water ($\theta$>1.5 °C) has significantly decreased. Nevertheless, the FSBW waters are also observed at this longitude and affect the slopes of isopycnic surfaces in a layer up to 300 m. The cold waters with $\theta$<0 °C, which can be associated with the BSBW, are observed only at two stations

in the depth range close to 1000 m, and are ~~practically~~ absent at ~~the~~ depths above 950 m. The isopycnic surfaces in the bottom panel of Fig. 5 are relatively flat, indicating weak geostrophic flow (see Section 3.2). ~~Note that, the water with absolutely~~ The "absolutely" stable" thermohaline stratification ~~is well visualized~~ below the temperature maximum with temperature decreasing and salinity increasing with depth (Fig. 5, bottom panel)~~: the temperature decreases and salinity increases with depth. This feature of the mean thermohaline stratification~~ is common to the Upper Polar Deep Water (UPDW) layer (Rudels et al., 1999).

In Fig. 6 three transects are presented, ~~two of which were made~~ at 126°E and 142°E (NABOS-2005) and ~~the third one was made~~ in the Makarov Basin at 159° E (NABOS-2007). On the transect along 126°E large slopes of isopycnic surfaces are observed, which corresponds to a fairly ~~intensive~~ strong geostrophic flow (see Section 3.2), confined to the depth range of 200−400 m, that is, to the area occupied by the FSBW. At the 142°E transect ~~which is located~~ on the Lomonosov Ridge, and at the 159°E transect in the Makarov Basin, the FSBW can be still identified as a warm layer ~~within a depth range of~~ between 200 and 400 m, where the maximum temperature is reduced to 1.49 °C and 1.42 °C, respectively (Fig. 6). The 142°E transect implies some eastward geostrophic transport, whereas at the 159° E transect, and in the area of cold waters (~~the depth range~~ below 800 m) in the sections shown in Fig. 6, the baroclinic flow is weak or absent.

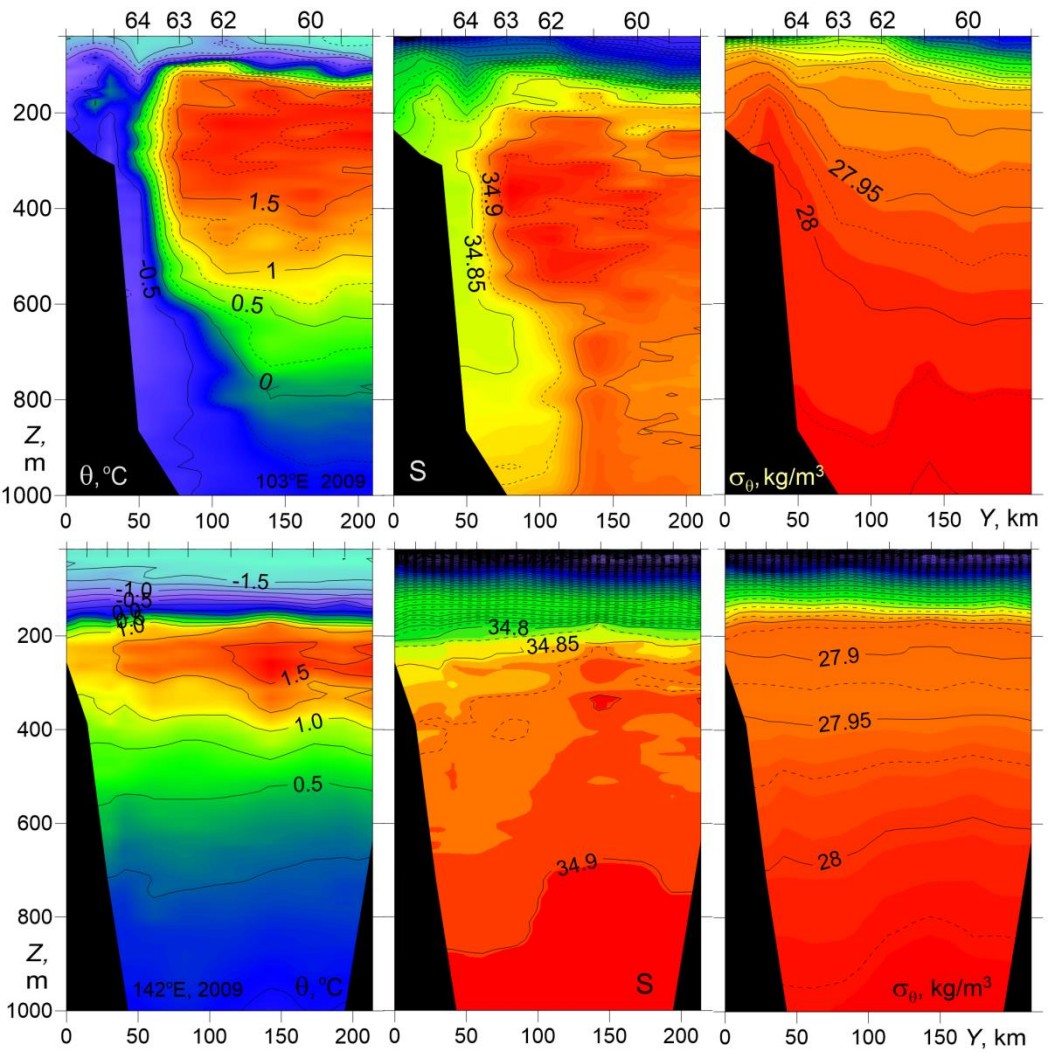

Fig. 5. Temperature $\theta$, salinity $S$, and potential density anomaly $\sigma_\theta$ versus distance and depth for cross-shelf transects at 103°E (upper) and 142°E (lower) (NABOS-09).

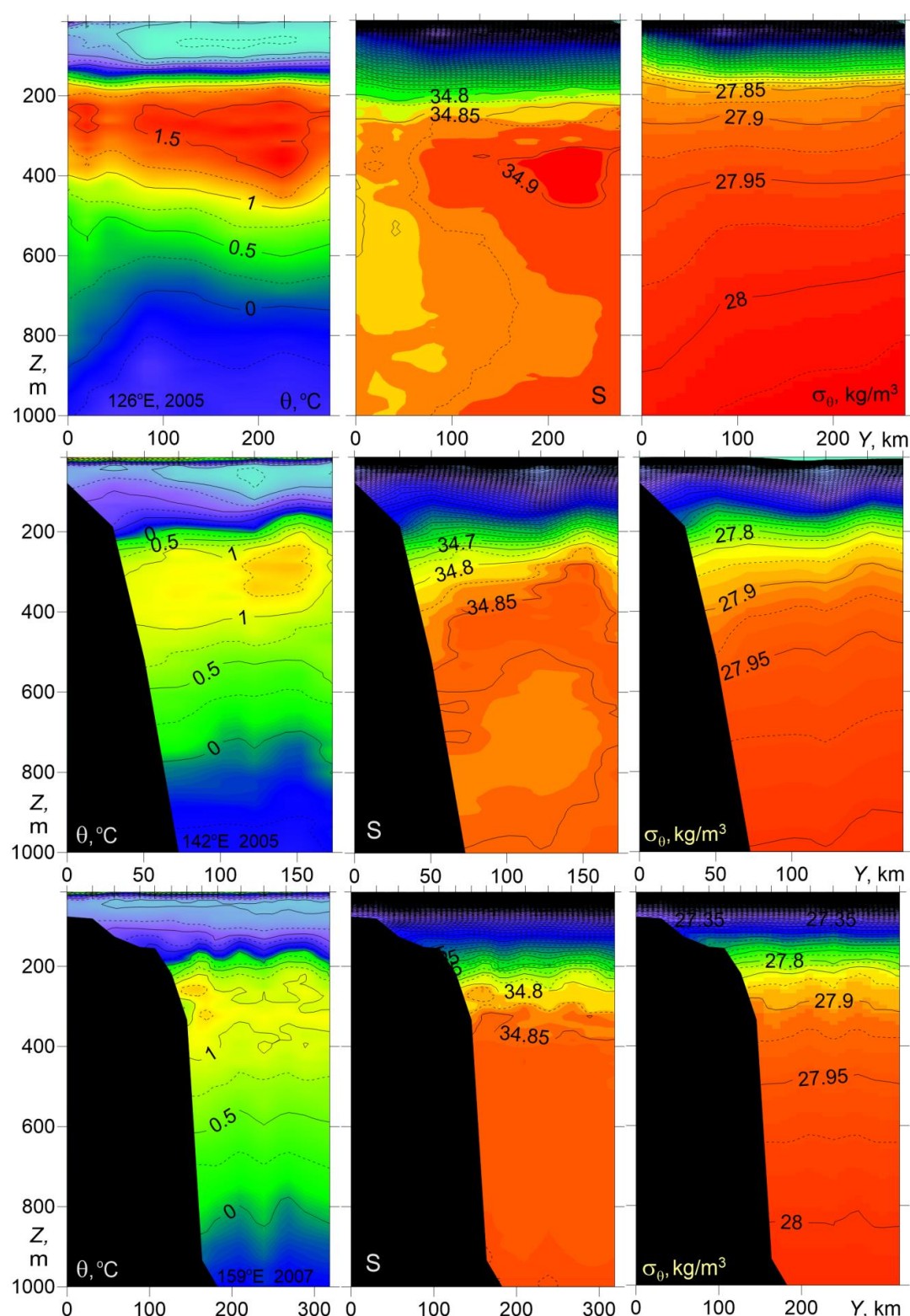

Fig. 6. Temperature $\theta$, salinity $S$, and potential density anomaly $\sigma_\theta$ versus distance and depth for cross-shelf transects at 126°E, 142°E (top and middle, NABOS-2005) and 159°E (bottom, NABOS-2007).

In summary, ~~the~~ a combined FSBW-BSBW structure with isopycnals sloping down to the north (from the slope), is typical for the longitude range 94–107°E. ~~On~~ In the transects ~~made~~ along 126°E, 142°E, and 159°E, ~~the slopes of isopycnic surfaces~~ sloping isopycnals ~~indicating~~

~~the baroclinic flow,~~ were observed generally in the depth range of 200–400 m, that is in the area occupied by the FSBW. As the FSBW moved along the continental slope of the Eurasian Basin, ~~a significant decrease of~~ its core temperature ~~was observed in the FSBW core~~ decreased, but could be identified at all transects, including the two transects in the Makarov Basin (159°E). The cold waters ~~on~~ in the transects along 126°E, 142°E and 159°E, which can be associated with

the BSBW, had a minimum temperature   above −0.5 °C, were ~~observed in the depth range~~ located below 800 m, and had ~~a little effect on the spatial structure of~~ relatively flat isopycnic surfaces ~~and horizontal gradient of density~~.

### 3.1.2 $\theta$-S analysis

The difficulty in identifying the BSBW in the eastern part of the Nansen Basin is related to
the overlapping ranges of temperature and salinity inherent to the BSBW and the UPDW: − 0.5 °C<θ<0 °C, and the salinity is close to 34.9 (Rudels et al., 1994; Walsh et al., 2007). It is also important to note that the BSBW in the St. Anna Trough mixes with the FSBW. Therefore, not only the cold Atlantic Waters, which are transported by the bottom gravity current, but also mixed warmer waters can enter the Nansen Basin through the trough (see Fig. 3). A detailed $\theta$-S
analysis of different CTD sections can provide useful information on the transport and transformation of FSBW and BSBW. ~~Note that a pronounced~~ A distinct $\theta$-S ~~signal~~ signature ~~clearly~~ indicates that the water mass has entered the area of observation. The absence of ~~a signal~~ a signature on the theta-S space indicates ~~one of the following: a)~~ either the water mass did not enter the area of observation~~; b) it entered the area of observation being highly~~ or was
transformed~~, namely, mixed~~ after mixing with other waters.

The differences in the behavior of the $\theta$-S values are observed in the upper and deep layers of the Eurasian Basin and the St. Anna Trough (Fig.7). On the other hand, one cannot miss a similarity in the shape of the $\theta$-S curves in the salinity range of 34.5−35.0. The similarity is obviously caused by the presence of FSBW. Fig. 7 demonstrates the transformation of the FSBW
and BSBW moving along the continental slope of the Eurasian Basin. More detailed information on the BSBW transformation can be extracted from $\theta$-S diagrams presented in Fig. 8.

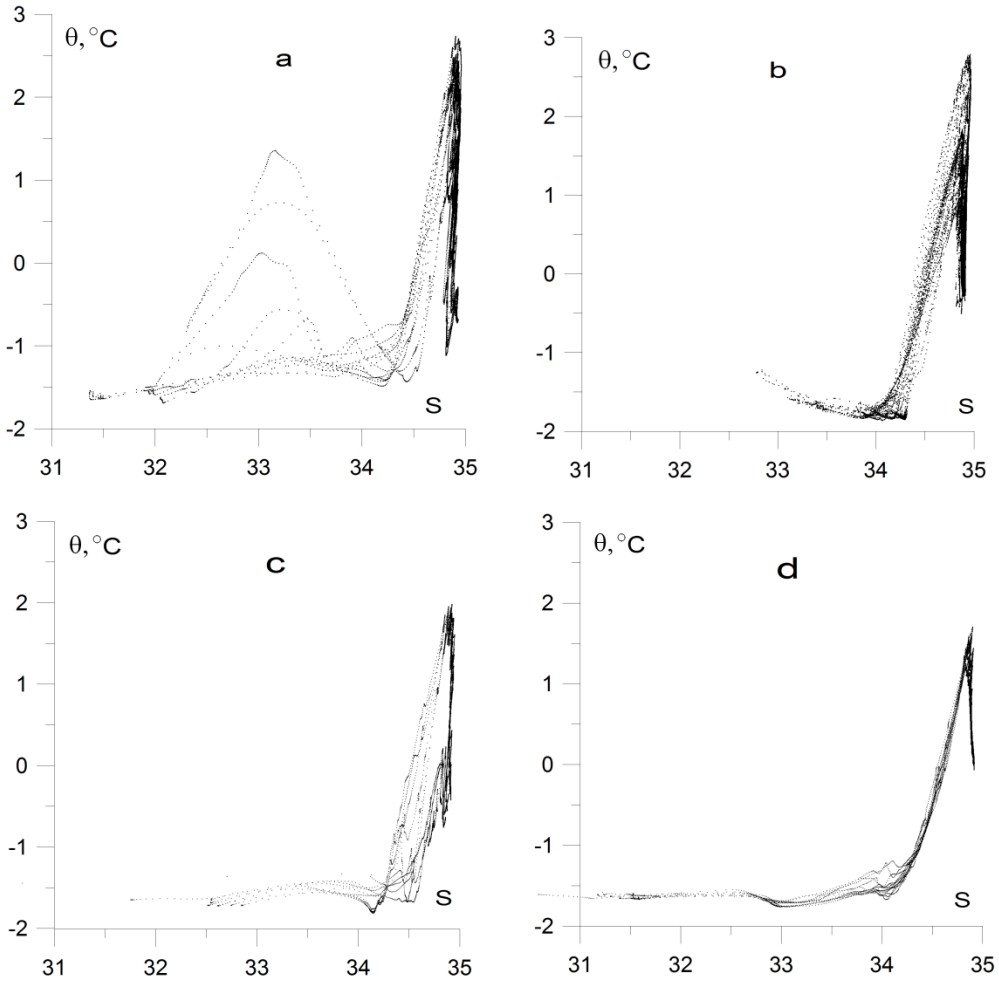

Fig. 7. *θ-S* diagrams based on the CTD profiling in (a) the St. Anna Trough (NABOS-09, 82° N), (b) the PS-96 section at 92°E, and the NABOS-09 sections at 103°E (c) and 142°E (d). For convenience of presentation, the points of the *θ-S* curves with salinity below 30 were ~~dropped~~ excluded.

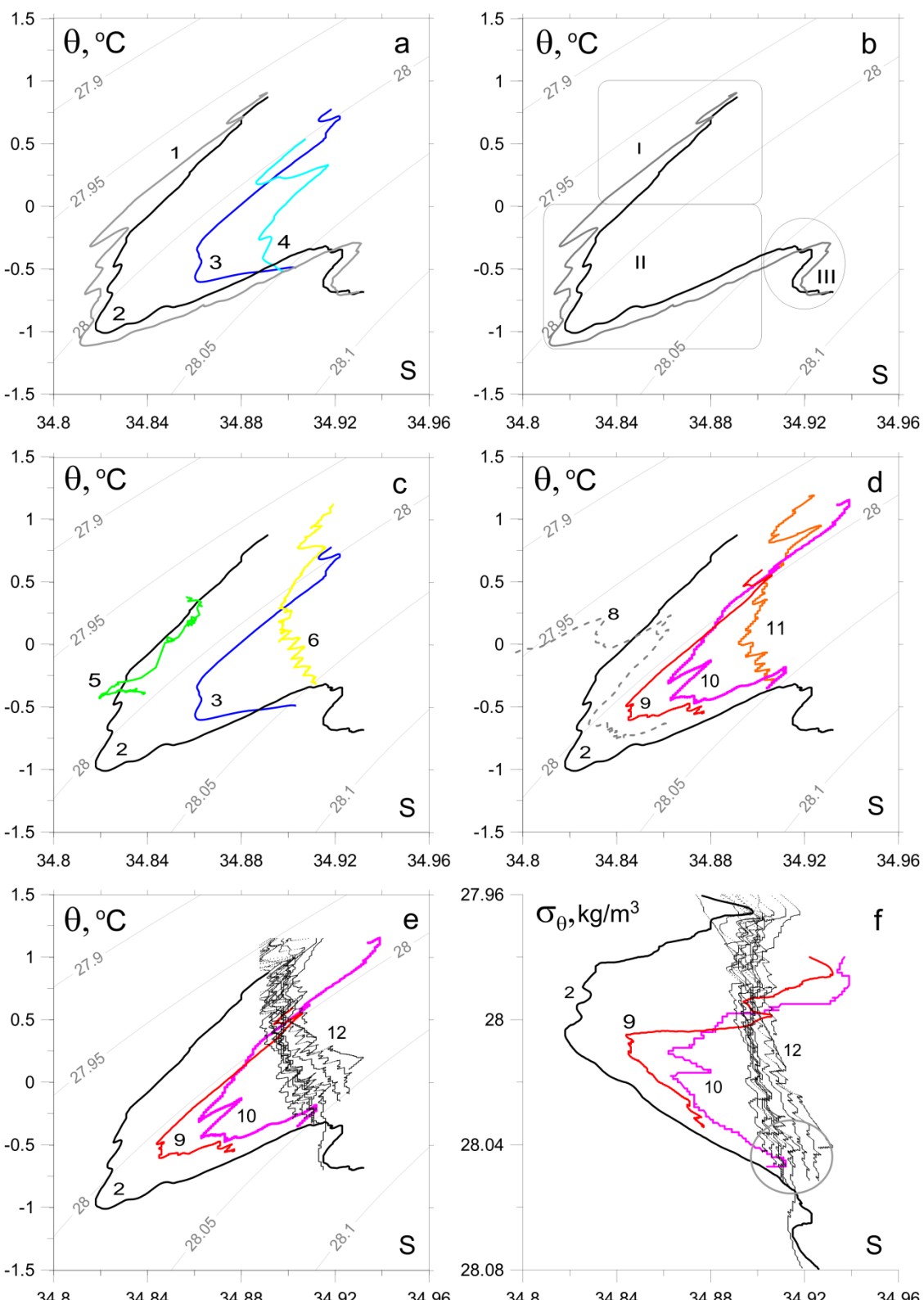

Fig. 8. Thermohaline values of the BSBW and FSBW: a) based upon the CTD profiles, obtained in the St. Anna Trough (NABOS-09, section 82°N), curves 1−4 correspond to the stations (st.) 76, 78, 83 and 80, respectively; b) the same as "a" but only curves 1 and 2 are presented; regions I, II, III illustrate three different water masses in accordance with (Dmitrenko et al., 2015); for explanation see the text; c) based upon the section of PS-96, curves 5 and 6 corresponding to st. 32 and 42, respectively (depth range 600−1000 m), curves 2 and 3 are shown for the reference;

d) for CTD profiles at the 103°E section, NABOS-09, curve 8 (st. 64), curve 9 (st. 63), curve 10 (st. 62), curve 11 (st. 60), and curve 2 for the reference (see Fig. 5 for the location of the stations); e) based upon the CTD profiles in the depth range 500−1200 m measured at the 126°E (section of NABOS-09), curves 12; curves 2, 9 and 10 are shown for the reference; f) the same as "e" but presented in coordinates $\sigma_\Theta$, $S$.

The $\theta$-$S$ curves marked as 1 and 2 in Fig.8a correspond to stations 76 and 78, respectively, which were located at the eastern slope of the St. Anna Trough just in the near-bottom gravity current carrying the BSBW, while the curves marked as 3 and 4 correspond to stations 83 and 80 located near the mid-point (thalweg) of the trough in the western periphery of the gravity current (the location of the stations is shown in Fig. 3). To visualize ~~better~~ the BSBW transformation

better, the points of $\theta$-$S$ curves in the temperature and salinity ranges of $\theta > 1.2$ °C and $S < 34.76$, respectively, were omitted. Similar $\theta$-$S$ curves in the St. Anna Trough were observed within NABOS Program in other years (NABOS-13, NABOS-15).

    The curves 1 and 2 in Fig. 8a have similar knee-like shape (Dmitrenko et al., 2015) formed by (i) the upper warm and saline water layer of the FSBW ($\theta \gg 0$ °C), (ii) the intermediate

colder and fresher water layer of BSBW ($\theta < 0$ °C) underlying the FSBW, and (iii) the denser, warmer and saltier "true" mode of the BSBW ($\theta \approx 0$ °C), see Fig. 8b: FSBW (region I), BSBW (region II), "true" mode BSBW (region III). ~~The difference between t~~The BSBW ~~and~~ differs from the "true" mode BSBW ~~is in that the former~~, and is more diluted with the colder and fresher Barents Sea water (for more details see ~~paper by~~ Dmitrenko et al., 2015). We will be

interested in the transformation of the main part of the knee (region II), namely the transformation of BSBW.

    In Fig. 8c the comparison of typical $\theta$-$S$ curves related to the St. Anna Trough (they are also shown in the other panels of Fig. 8 for reference) with that of the 92°E section of PS-96 is given: the curves 5 and 6 correspond to st. 32 and st. 42 (depth range 600−1000 m) of the PS-96

section, respectively. St. 32 was located next to the slope, while st. 42 was located about 250 km apart from the slope. The coincidence of curve 5 with a part of curve 2 ~~evidences for the~~ implies a BSBW ~~moving~~ flow along the slope of Nansen Basin (see Fig. 4 and its legend 1). Curve 6 corresponds to the UPDW. The $\theta$-$S$ diagrams for CTD profiles at the section 103°E are presented by curves 8-11 (see Fig. 5 for the locations of stations). Curves 8, 9, and 10 are similar to curve

2, and indicate the BSBW. Curve 11, ~~being~~ similar to curve 6 in Fig. 8c, corresponds to ~~the $\theta$-$S$~~ ~~values of~~ the UPDW. However, the BSBW is not observed ~~in the section~~ at 126°E: see Fig. 8e, where a collection of $\theta$-$S$ curves (collectively referred as 12) presents all CTD profiles in the depth range 500−1800 m measured at ~~the section~~ 126°E of NABOS-09. Also we do not observe

the BSBW further to the east on the ~~section~~ 142°E section of NABOS-09 (not shown) ~~as well as~~ or in the Makarov Basin.

~~To estimate the potential density of deep waters at the sections 103°E and 126°E~~ $\sigma_\theta$-$S$ ~~diagrams are shown in Fig. 8f: curves 2, 9 and 10 correspond to~~ $\theta$-$S$ ~~curves 2, 9 and 10 presented in Fig. 8d, curves 12 correspond to curves 12 in Fig. 8e. As one can see, t~~The BSBW at 103°E and 126°E is also characterized by a knee-shape ~~diagram also in coordinates~~ in $\sigma_\theta$, $S$ coordinates (Fig. 8f, numbers correspond to those in other panels) . However the knee-shape diagram is not observed along 126°E (curve 12) in these coordinates. The dense and cold deep waters in the section 126°E have $\sigma_\theta$, $\theta$, $S$ values typical for the "true" BSBW mode (Dmitrenko et al., 2015). Nevertheless, ~~it is hardly correct to consider~~ these waters (see $\sigma_\theta$, $S$ values inside the circle; Fig. 8f) ~~as the "true" BSBW mode, since~~ $\sigma_\theta$, $\theta$, $S$ ~~values of these waters satisfactorily~~ also correspond to ~~$\sigma_\theta$, $\theta$, $S$ values of~~ the UPDW characteristics hence cannot be distinguished as the "true" BSBW mode. To evaluate the transformation of the "true" mode of BSBW an additional analysis is required, which is beyond the scope of this paper.

~~The results presented in Fig. 8 show that t~~The BSBW ~~signal~~ which is characterized by the knee-shape diagram in coordinates $\theta$-$S$ and $\sigma_\theta$-$S$, is not visible at 126°E (Fig. 8). This is consistent with the conclusion formulated in Subsection 3.1.1 that by 126°E the BSBW is not accompanied by any noticeable ~~perturbations~~ tilt of isopycnals. Moreover, given the characteristic feature of the $\theta$-$S$ structure of BSBW in the St. Anna Trough (curves 1−4 in Fig. 8a) was observed in other years, we carried out a similar analysis using all available CTD data and found that the BSBW ~~signal is either strongly weakened or not visible~~ is not distinct at this longitude (see Fig.9). The only exception was 2002, when the BSBW ~~signal~~ was still observed at 126ºE. It suggests that the BSBW and FSBW begin to mix intensively immediately after 103ºE. ~~However~~ On the other hand, the FSBW ~~signal~~ is well identified at 126ºE and further along the slope of the Eurasian Basin (and even in the Makarov Basin), while we cannot say the same about the BSBW ~~signal~~. Thus, one may assume that east of 126ºE the geostrophic volume flow rate of the AW is mainly provided by the FSBW.

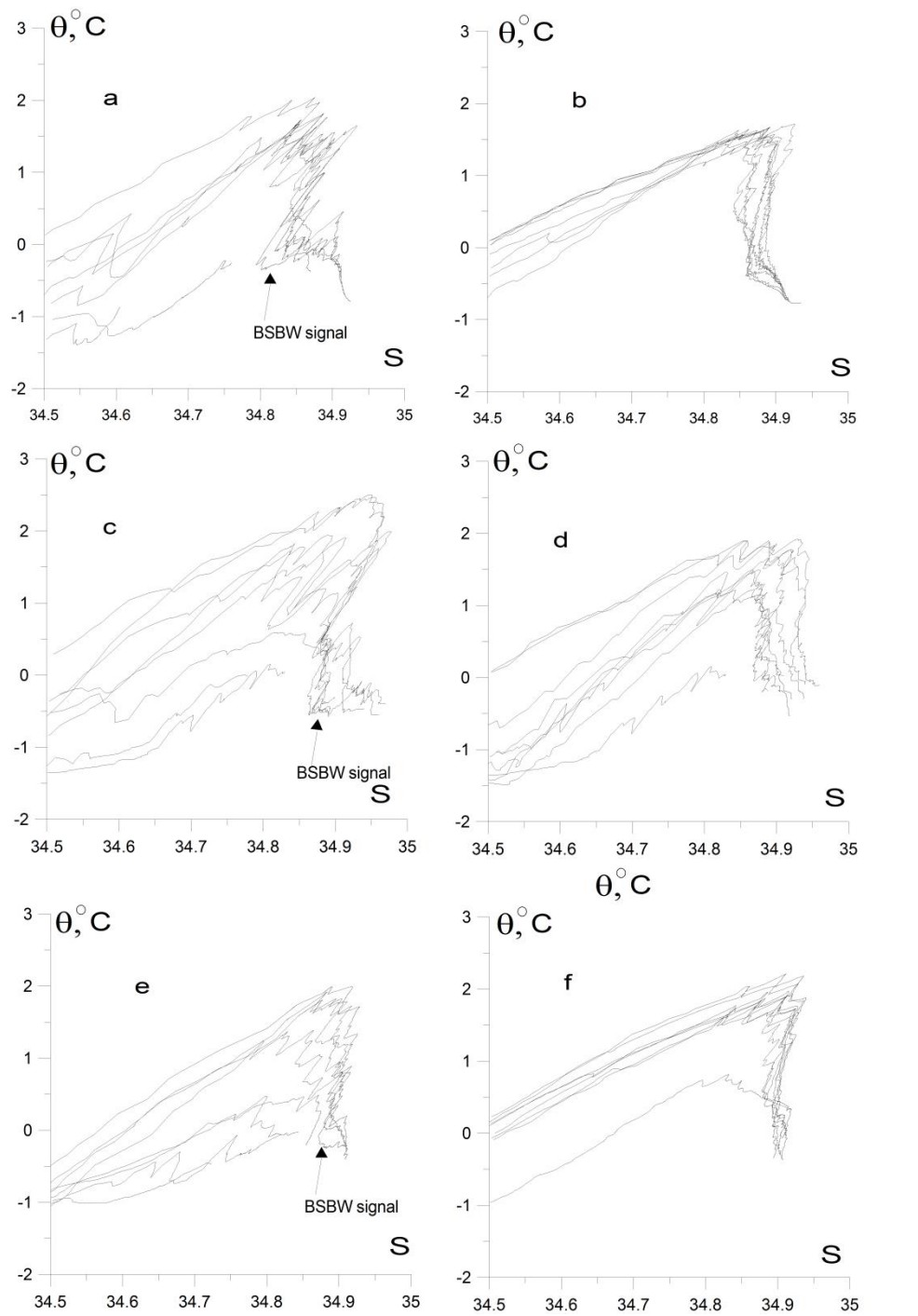

Fig. 9. θ-S diagrams based on the CTD profiling : NABOS-05: (a) and (b), 103°E (a), 126°E (b); NABOS-06: (c) and (d), 103°E (c), 126°E (d); NABOS-08: (e) and (f), 103°E (e), 126°E (f).

## 3.2 Characteristics of the Atlantic Water flow and geostrophic estimates of the volume flow rate

The estimates of the geostrophic volume flow rate ~~V, as well as estimates of~~ and the hydrological parameters describing the AW flow in the Eurasian and Makarov Basins, are presented in Table 1. The geostrophic estimates of the near-bottom ~~gravity~~ volume flow rate of the BSBW in zonal transects across the St. Anna Trough are presented in Table 2. The only

exception is the transect at 82°N, where the near-bottom gravity current with a considerable eastward component due to overflow across a sufficiently deep ridge (approx. 500 m deep) east of the St. Anna Trough (Fig. 3, top panels) makes the estimate of AW transport northward questionable. Note also that ~~prior to the BSBW entering the area of the Eurasian Basin~~ to the west of the St. Anna Trough our estimates refer to the FSBW; to east of this region BSBW enters the Eurasian Basin and our estimates should be attributed to the joint contribution of the two branches ~~– the FSBW and BSBW – to the transfer of the AW~~ (FSBW and BSBW).

The hydrological parameters shown in Table 1 can be interpreted as follows. The maximum water temperature of the AW may exceed 5 °C in cases when the AW inflow to the Eurasian Basin consists of especially warm water masses. Typical changes in the ~~maxima~~ temperature and salinity maxima of the AW moving along the slope over a distance of about 1000 km are approximately 1–2 °C and 0.1, respectively. ~~Such values of the maximum temperature of the AW~~ These changes lead to a slight increase in potential density and therefore a deviation of the AW from the isopycnic distribution ~~should~~ can be expected. ~~This effect is~~ These changes are most likely associated with the exchange of heat, salt, and mass with the surrounding waters ~~due to the formation of~~ through intrusive layering and ~~the influence of~~ double diffusion (~~on the observation and study of intrusions in the Arctic Basin~~ see, e.g., ~~Rudels et al., 1999;~~ Kuzmina et al., 2011; Polyakov et al., 2012; Kuzmina et al., 2018) and ~~also with the AW core transformation by~~ sea ice melting and cooling (Rudels, 1998). The intrusions, in particular, can also contribute to the reduction of the AW heat and salt content and the volume flow rate. The differences in the AW heat and salt content and the volume flow rate can be clearly seen from the PS-96 section when comparing data from stations near the continental slope of the Eurasian Basin at 92°E and from the vicinity of the Lomonosov Ridge at 140°E.

It is worth noting that the maximum value of the AW temperature ($\theta_{max}$) ~~according to the presented~~ in this data ~~set~~ is always observed in the upper layer of the Eurasian Basin at ~~the~~ depths below the ~~density jump layer~~ pycnocline but not exceeding 350 m, while the maximum salinity ($S_{max}$) at sections in the eastern part of the Basin can be observed at depths greater than 1000 m.

$X_{\theta max}$ in Table 1 is the distance of the AW core (which can be associated with $\theta_{max}$) from the slope/shelf boundary. The highest value and the maximum variation of this parameter is observed near 126°E and 142°E, where a two-core structure of AW is often observed (Pnyushkov et al., 2015).

The noticeable increase of $\theta_{max}$ in 2006 at 31°E and 103°E and the intensive warming of the AW were first reported in (Polyakov et. al., 2011). The present results show that the increase

of the temperature of the AW in 2006 was also accompanied by an increase of volume transport (see Table 1, the section along 103°E and reasonings below). This can be caused not only by the warming of the AW, but also by an increased inflow of the AW to the Eurasian Basin.

The ~~evaluations of~~ geostrophic ~~current~~ transport in the range of 31–159°E ~~are~~ is characterized by a high variability (Table 1). This may be due to ~~the following reasons: a) the deviation of some sections from the normal~~ a) a section orientation oblique to the current; b) the difference in the horizontal scales of the sections; c) ~~some~~ uncertainty in the choice of the reference level for geostrophic calculations; d) meandering of the flow; e) the effect of synoptic quasi-geostrophic eddies on the flow volume rate. ~~All of these reasons contribute some noise to the resulting volume flow rate estimates.~~ In order to find statistically consistent estimates of the variability of geostrophic volume flow rate along the slope of the basin based on a limited ~~material~~ data set, the following was done. The volume flow rates obtained for all sections within the range 31°−92 °E for different years were used to calculate the mean volume flow rate (region I; the number of volume flow rate values ~~to be~~ averaged is $N = 6$). Similarly, the average volume flow rate was calculated for the region 94°−107°E (region II; $N = 9$). The remaining average estimates of geostrophic volume flow rate were calculated for sections 126°E (region III; $N = 9$), 142°E (region IV; $N = 10$) and 159°E (region V; $N = 2$). Then the ~~confidence intervals with a probability of~~ 95% ~~(typical confidence interval)~~ and 80% ~~(acceptable confidence interval for working with a limited statistical material)~~ confidence intervals were determined using the Student t-distribution. All estimates of average volume flow rates and confidence intervals are presented in Tables 1 and 2.

~~The above mean estimates allow us to conclude that~~ On the average, the volume flow rate increases from region I to region II, then decreases to region III and ~~after that decreases to~~ region IV, followed by a sharp decrease in region V. However, ~~the 95% confidence intervals validate~~ only the difference between the volume flow rate in region II and the values in regions IV and V are significant at 95% confidence. Transport values bounded by the confidence intervals for regions II, IV and V are  (0.46; 1.72), (0.12; 0.44) and (-0.37; 0.43), respectively. These intervals indicate that the mean volume flow rate in region II exceeds the value of the same parameter in regions IV and V with a high probability of 95%. The 80% confidence intervals overlap only for regions III and IV, (0.25; 0.53) and (0.18; 0.38), respectively. In this regard, ~~we can declare that the above described~~ the change in the volume flow rate along the slope is ~~reliable~~ significant with a probability of 80%, except for changes in volume flow rate from region III to region IV.

The above values of the mean volume flow rate and confidence intervals also suggest that the increase in volume flow rate in 2006 ~~was caused by the climate impact~~ is significant, and not

caused by the "noise" in the data. Indeed, the volume flow rates in regions II, III, and IV in 2006 exceeded the upper limits of the corresponding 95% confidence intervals. From statistical point of view such a significant increase in volume flow rates at the same time in three regions is a very rare event that can hardly be explained by random "noise" in the data caused, for example, by the influence of synoptic eddies.

Let us turn our attention to the following features of the volume flow rate estimates: high volume flow rate estimates at 96°E, 103°E, 107°E, a negative volume flow rate estimate at 126°E in 2013 and low volume flow rate estimates at 31°E, 98°E in 2009 (Table 1). Indeed, the AW volume flow rate in the BSBW area of entry into the Eurasian Basin in 2013 was almost equal to the maximum volume flow rate in 2006 (103°E) and was quite high up to the longitude 430    107°E. This phenomenon as well as the intense warming in 2006 can be associated with the ~~impact of climate~~ recent changing conditions in the Arctic. ~~The~~We hypothesize that the negative volume flow rate at 126°E was~~, according to the authors, due to~~ because of the influence of local return flows which can be observed near the slope (Pnyushkov et al., 2015). Low FSBW volume flow rate estimates in 2009 are probably associated with a strong deviation of the flow from the 435    slope, which may ~~have been resulted in an underestimation of~~ underestimate the AW volume ~~flow rates~~ transport due to the small length of the ~~cuts~~ transects to the north (see also Section 4 ~~below~~). ~~Another reason may be a sharp decrease in the intensity of the flow of the AW through the Fram Strait that most likely took place that year.~~

~~It is important to analyze average values of volume flow rate $V_{mean}$ in region I and in the St.~~ 440    ~~Anna Trough.~~ The mean value of the FSBW volume flow rate in region I is $V_{mean} = 0.5$ Sv. This estimate of volume flow rate is about half the estimate of the BSBW mean volume flow rate, $V_{mean} = 0.79$ Sv ($N = 3$, Table 2). (The difference is significant at 80% confidence interval~~s do not overlap indicating that the BSBW volume flow rate does exceed the FSBW volume flow rate~~). The BSBW mean volume flow rate exceeding nearly twice the FSBW mean volume flow 445    rate results in a dominance of the BSBW pattern of potential density contours in the longitude range of 94–107°E (region II), where both branches of the AW are present. Moreover, the sum of the mean values of the FSBW and the BSBW volume flow rate geostrophic estimates~~,~~ $V_{mean} = 0.5 + 0.79 = 1.29$ Sv, corresponds well ~~to the mean geostrophic estimate of volume flow rate for~~ with the combined FSBW and BSBW flow within the region II: $V_{mean} = 1.09$ Sv. Thus, the 450    increase in geostrophic ~~volume flow rate~~ transport in region II is mainly due to the influence of the BSBW. ~~It should be noted that, according to sections 3.1.1 and 3.1.2, t~~The decrease in geostrophic volume flow rate in region III can also be associated primarily with the BSBW,

namely, with the decrease in the BSBW ~~signal~~ transport in the 126ºE section and further along the slope (see sect. 3.1.1 and 3.1.2).

Finally, at the ~~section~~ 159°E ~~located~~ section in the Makarov Basin, the geostrophic estimate of the along-slope volume flow rate of mixed waters of the FSBW and the BSBW has further greatly reduced down to $V_{mean} = 0.03$ Sv ($N = 2$), which is of more than one order of magnitude smaller than that in the Nansen and Amundsen Basins. Despite the low statistical significance of the latter estimate (due to small value of $N = 2$) one may conclude that the major part of the AW entering the Arctic Ocean circulates cyclonically within the Nansen and Amundsen Basins, and only its small part flows to the Makarov Basin (Rudels et al., 2015; Rudels, 2015). However, additional studies ~~using more CTD data~~ are required to confirm this result.

Table 1. Characteristics of the Atlantic Water flow in the course of its propagation along continental slope of the Eurasian Basin of the Arctic Ocean. *Dist* is the along-slope distance from the Fram Strait; $\theta_{max}$ is the maximum temperature; $\sigma_\theta (Z_{\theta max})$, $S(Z_{\theta max})$, $Z_{\theta max}$, and $X_{\theta max}$ are the values of potential density, salinity, depth, and lateral displacement from the slope for the point $\theta_{max}$; $S_{max}$ and $Z_{Smax}$ are the ~~same as $\theta_{max}$ and $Z_{\theta max}$ but for the salinity~~ maximum salinity and depth of $S_{max}$ ; $V$ is the geostrophic estimate of the volume flow rate. The mean values and 95% / 80% confidence intervals of the volume rate, $V_{mean}$, calculated separately for CTD transects at 31−92°E, 94−107°E, 126°E, 142°E and 159°E, are ~~presented too~~ also shown. The last row in the Table presents the characteristics of the return flow of the AW by the Lomonosov Rigde at the longitude 140°E and latitude 86.5°N (PS96, see Fig. 1). Year is given in the first column (e.g. NABOS06 corresponds to 2006).

| *Exp* | *Lon* [°E] | *Dist* [km] | $\theta max$ [°C] | $\sigma_\Theta(Z_{\theta max})$ [kg/m³] | $S(Z_{\theta max})$ | $Z_{\theta max}$ [m] | $X_{\theta max}$ [km] | $S_{max}$ | $Z_{Smax}$ [m] | $V$ [Sv] |
|---|---|---|---|---|---|---|---|---|---|---|
| NABOS06 | 31 | 404 | 5.670 | 27.579 | 34.980 | 42 | -11 | 35.099 | 72 | 0.57 |
| NABOS08 | 31 | 404 | 4.883 | 27.771 | 35.103 | 101 | 0 | 35.105 | 176 | 0.80 |
| NABOS09 | 31 | 404 | 3.691 | 27.818 | 34.999 | 89 | 0 | 35.002 | 91 | 0.10 |
| NABOS09 | 60 | 856 | 2.503 | 27.891 | 34.951 | 175 | 10 | 34.981 | 363 | 0.47 |
| NABOS13 | 90 | 1290 | 2.600 | 27.903 | 34.975 | 250 | 41 | 34.996 | 333 | 0.46 |
| PS96 | 92 | 1322 | 2.786 | 27.875 | 34.960 | 271 | 33 | 34.968 | 329 | 0.58 |
| | | | | | | | | $V_{mean} = 0.50 \pm0.24$ / $\pm0.14$ Sv | | |
| NABOS15 | 94 | 1355 | 2.445 | 27.946 | 35.012 | 331 | 33 | 35.015 | 365 | 0.47 |
| NABOS13 | 96 | 1388 | 2.548 | 27.902 | 34.969 | 207 | 70 | 34.978 | 264 | 2.06 |
| NABOS09 | 98 | 1421 | 2.300 | 27.906 | 34.948 | 220 | 79 | 34.971 | 345 | 0.09 |
| NABOS05 | 103 | 1561 | 2.029 | 27.870 | 34.876 | 179 | 39 | 34.934 | 309 | 0.32 |
| NABOS06 | 103 | 1561 | 2.528 | 27.888 | 34.950 | 220 | 50 | 34,978 | 260 | 2.23 |
| NABOS08 | 103 | 1561 | 1.980 | 27.886 | 34.891 | 201 | 60 | 34.929 | 325 | 0.42 |
| NABOS09 | 103 | 1561 | 1.984 | 27.913 | 34.925 | 244 | 50 | 34.951 | 365 | 0.87 |
| NABOS13 | 103 | 1561 | 2.278 | 27.904 | 34.942 | 215 | 80 | 34.956 | 419 | 1.59 |
| NABOS13 | 107 | 1695 | 1.903 | 27.937 | 34.945 | 359 | 120 | 34.948 | 404 | 1.77 |
| | | | | | | | | $V_{mean} = 1.09 \pm0.63$ / $\pm0.38$ Sv | | |
| NABOS02 | 126 | 2104 | 1.406 | 27.938 | 34.902 | 324 | 243 | 34.932 | 2061 | 0.05 |
| NABOS03 | 126 | 2102 | 1.341 | 27.941 | 34.899 | 336 | 342 | 34.921 | 1886 | 0.41 |
| NABOS04 | 126 | 2102 | 1.770 | 27.906 | 34.896 | 271 | 87 | 34.925 | 2431 | 0.61 |

| | | | | | | | | | | |
|---|---|---|---|---|---|---|---|---|---|---|
| NABOS05 | 126 | 2102 | 1.695 | 27.936 | 34.926 | 359 | 227 | 34.935 | 2841 | 0.75 |
| NABOS06 | 126 | 2102 | 1.905 | 27.923 | 34.930 | 284 | 193 | 34.960 | 968 | 0.77 |
| NABOS07 | 126 | 2102 | 2.085 | 27.907 | 34.928 | 266 | 242 | 34.942 | 340 | 0.60 |
| NABOS08 | 126 | 2102 | 2.195 | 27.885 | 34.911 | 206 | 235 | 34.939 | 365 | 0.31 |
| NABOS09 | 126 | 2102 | 1.907 | 27.909 | 34.913 | 316 | 33 | 34.932 | 1018 | 0.40 |
| NABOS13 | 126 | 2102 | 1.946 | 27.937 | 34.949 | 346 | 228 | 34.951 | 428 | -0.21 |
| NABOS15 | 126 | 2102 | 1.653 | 27.918 | 34.898 | 246 | 400 | 34.942 | 3816 | 0.22 |
| | | | | | | | | $V_{mean}$ = 0.39 ±0.22 / ±0.14 Sv | | |
| NABOS03 | 142 | 2456 | 1.089 | 27.912 | 34.841 | 269 | 41 | 34.862 | 1000 | 0.06 |
| NABOS04 | 142 | 2456 | 1.401 | 27.909 | 34.865 | 281 | 0 | 34.907 | 1608 | 0.21 |
| NABOS05 | 142 | 2456 | 1.492 | 27.906 | 34.870 | 284 | 100 | 34.906 | 1550 | 0.26 |
| NABOS06 | 142 | 2456 | 1.981 | 27.874 | 34.876 | 234 | 111 | 34.960 | 1016 | 0.60 |
| NABOS07 | 142 | 2456 | 1.855 | 27.879 | 34.870 | 231 | 0 | 34.920 | 2064 | 0.09 |
| NABOS08 | 142 | 2456 | 1.599 | 27.915 | 34.890 | 260 | 200 | 34.908 | 347 | 0.23 |
| NABOS09 | 142 | 2456 | 1.704 | 27.915 | 34.900 | 253 | 101 | 34.917 | 1082 | 0.22 |
| NABOS13 | 142 | 2456 | 1.475 | 27.940 | 34.909 | 331 | 115 | 34.926 | 1150 | 0.18 |
| NABOS15 | 142 | 2456 | 1.353 | 27.936 | 34.892 | 326 | 106 | 34.913 | 1372 | 0.63 |
| | | | | | | | | $V_{mean}$ = 0.28 ±0.16 / ±0.10 Sv | | |
| NABOS07 | 159 | 2783 | 1.424 | 27.887 | 34.839 | 255 | 0 | 34.880 | 1075 | -0.01 |
| NABOS08 | 159 | 2783 | 1.383 | 27.893 | 34.843 | 245 | 0 | 34.889 | 1266 | 0.06 |
| | | | | | | | | $V_{mean}$ = 0.03 ±0.40 / ±0.10 Sv | | |
| PS96back | 140E 86.5N | 3178 | 1.812 | 27.890 | 34.880 | 219 | ≈ 700 | 34.902 | 472 | -0.09 |

Table 2. Geostrophic estimates of the volume flow rate for near-bottom gravity flow of the Barents Sea Branch of Atlantic Water (BSBW) on zonal transects across the St. Anna Trough. The uncertainty estimates are 95% and 80% confidence intervals.

| *Exp* | NABOS09 | NABOS13 | NABOS15 | |
|---|---|---|---|---|
| *Lat* [°N] | 81.00 | 81.33 | 81.41 | $V_{mean}$ |
| *V* [Sv] | 0.89 | 0.73 | 0.76 | 0.79 ±0.22 / ±0.10 |

### 3.3 Interannual variability of the AW temperature-salinity values and the volume flow rate

Within the NABOS project, ~~in accordance with Table 1,~~ the cross-slope CTD transects at 103°E, 126°E, and 142°E were repeatedly performed for a number of annual campaigns (Table 1): 2005, 2006, 2008 and 2013 (103°E), 2002–2009, 2013 and 2015 (126°E), 2003–2009, 2013, and 2015 (142°E). ~~T~~ We use the repeated transects ~~may contain some information on~~ to describe the inter-annual variability of the AW~~, and we attempted to explore such a possibility~~.

Time series of the ~~maximum temperature of the~~ AW temperature maximum , $\theta_{max}$ , and the related values of salinity $S(\theta_{max})$ and potential density anomaly $\sigma_\theta(\theta_{max})$ (Fig. 10) show that the period of 2006 to 2008 was characterized by ~~not only~~ an increased temperature of the AW in the eastern part of the Eurasian Basin, ~~but~~ an increased salinity and density reduction. The temperature excess during this period was as large as 0.6–1.0 °C relative to 2002–2003 and 0.3–0.6 °C relative to 2013–2015. ~~The time series of corresponding values of salinity $S(\theta_{max})$~~

displayed in 2006 local maxima at the transects 126°E and 142°E, and the absolute maximum at the transect 103°E; the salinity excess for the maxima largely decreased with the longitude from approximately 0.06 at 103°E to less than 0.01 at 142°E. In accordance with our analysis the time series of $\theta_{max}$ had a maximum in 2013 but only at 103°E (see Table 1 and Fig.10). The time series of $S(\theta_{max})$ display a trend an of an increase of AW salinity in 2006 2008 and 2013 also over time, that can be referred to as a AW salinization in early 2000s. The change of salinity of AW at 142°E in time also draws attention to the following aspect: the salinity increases almost monotonously in the period from 2003 to 2013. How can such behavior of salinity be explained The mechanism behind this salinity evolution is not clear. It is also worth noting that the maxima of $\theta_{max}$ and $S(\theta_{max})$ in 2006 and 2013 (at 103°E) were accompanied by the volume flow rate highs maxima in transport.

## 4 Discussion

Here we discuss the following issues: a) differences in the identification of the BSBW; b) a comparison of the geostrophic volume flow rate estimates with other studies; c) the weakening of the BSBW signal at 126 ºE and further east.

a) Advection and interaction of waters with different $\theta$-$S$ characteristics in the Arctic Basin, as well as the impact of climate change that has been observed over the past decade (Polyakov et al., 2017) complicate an accurate identification of water masses. However, a robust approach to the determination of the FSBW and BSBW, which was proposed in Dmitrenko et al. (2015), is effective for distinguishing the water masses of these AW the FSBW and BSBW branches. As an exception, this approach does not take into account some cases, namely fails when the FSBW temperature is below 0 °C (see Fig. 2 in Dmitrenko et al., 2015), and/or the BSBW temperature is close to 1 °C (see Fig. 6 in Schauer et al., 2002a). If such cases are rare, then either of the two approaches can be used to identify the BSBW and FSBW. Indeed, the identification of the BSBW on the PS-96 section in our case (we used the approach proposed by Dmitrenko et al., 2015; see paragraph 3.1.1) does not differ much from that proposed by Schauer et al. (2002b). However, these discrepancies can lead to almost an order of magnitude difference in estimates of the volume flow rate of the BSBW only due to the differences in the BSBW cross-sectional area.

b) Based on the velocity measurements with moored instruments (1997−2010) in the area of the West Spitsbergen Current (WSC) near the Fram Strait (zonal transect at ~78º50ˊ N), it was found that approximately 3 Sv of the AW flows into the Nansen Basin (Beszczynska-Möller et. al., 2012). The long-term mean volume transport confined to the WSC core branch (or Svalbard

branch in accordance with Schauer et al., 2004) included 1.3±0.1 Sv of the AW warmer than 2ºC. The offshore WSC branch (or Yermak branch) carried on average 1.7±0.1 Sv of the AW. ~~Investigation of water transport in and north of the Fraim Strait based upon CTD measurements on zonal and meridional sections have been done by Marnela et al. (2013).~~ The variability range

of ~~the estimates of~~ the AW geostrophic transport of the Svalbard branch ~~was calculated~~ for meridional sections ~~made in~~ from 1997, 2001, and 2003 (summer/fall)~~, and~~ was between 0.06 Sv and 0.7 Sv (Marnela et al., 2013). In Kolås and Fer (2018) observations of the oceanic current and thermohaline field (in summer 2015) in the three sections were used to characterize the evolution of the WSC along 170 km downstream distance. ~~G~~Absolute geostrophic transports of

AW ~~were calculated on the basis of absolute geostrophic velocities and it was shown that~~ ranged from 0.6 Sv to 1.3 Sv ~~of the AW is carried by~~ in the Svalbard branch. In accordance with earlier studies of the currents in ~~the~~ Fram Strait, recirculation of the AW can be significant, and the volume flow rate of the AW entering the Arctic Ocean ~~can be equal only 1 Sv (Rudels, 1987), or it~~ ranges from 0.6 Sv to 1.5 Sv (Rudels, 1987; Aagaard and Carmack, 1989).

Our estimate of the mean volume flow rate $V_{mean}$ in region I (31º−92 ºE) is in the range of ~~variation in~~ the above estimates. However, the upper confidence limit of our estimate does not reach 1 Sv. Moreover, we used ~~the inequality~~ T> 0ºC to identify the AW while in Beszczynska-Möller et al. (2012) the volume flow rates of the AW entering the Eurasian Basin through ~~the~~ Fram Strait were determined ~~from the~~ for waters with T > 2 °C ~~condition. In this regard, we can~~

~~admit that our assessment is somewhat underestimated. Probably, this may be due to the fact that~~ Comparatively smaller transport in region I may be because the sections along ~~the longitudes~~ 31°E (see Fig. 1) are less than 100 km~~. Actually, at the sections along this longitude~~ wide and do not cover the full extent of the FSBW (Fig. 2, upper panel) ~~only a part of the FSBW is observed~~. Given ~~that the volume flow rate estimate is sensitive~~ the sensitivity to the definition of AW and

the resulting ~~to the accepted value of~~ cross-sectional area ~~of the AW~~ (see ~~issue~~ point "a" above), the volume transport may be underestimated. ~~One cannot also ignore the fact that horizontal density gradients of the geostrophic flow can be strengthened or weakened during~~ It is possible that the formation and passage of synoptic eddies~~, the influence of which on the average density field cannot be filtered out~~ leads to variability in transport rates. According to Perez-Hernandez

et al. (2017) north of Svalbard (between 21 and 33°E) in September, 2013, a large difference was found in the estimates of geostrophic volume flow rate (from 0.53 Sv to 3.39 Sv) due to the passage of eddies and meandering of the current. Våge et al. (2016) based on geostophic velocities at two CTD sections across the boundary current near 30º E (September, 2012) evaluated a net AW volume flow rate of 1.6±0.3 Sv. ~~Authors of this paper~~ They found evidence

of a large eddy affecting the mean volume transport calculations.The barotropic velocity component, which is not taken into account in our estimates, can also ~~affect the values of the volume flow rates~~ contribute to larger transports. However, ~~if the ice cover in the Eurasian Basin is high, the barotropic addition to the flow velocity in a stratified ocean hardly can play a decisive role~~ in conditions with high ice concentration in the Eurasian Basin, we might expect a

reduced barotropic contribution from the sea level changes induced by wind forcing. In ~~accordance to~~ cruise reports, the NABOS CTD sections were characterized by ~~the~~ ice concentrations of 50−100% (see https://uaf-iarc.org/nabos-cruises/). Exceptions occurred in the near-slope areas of the Laptev Sea, that is, in the sections along ~ 126ºE, where the ice concentration varied from 0 to 100%, having a maximum value in the northern part of the

sections. In such areas, the contribution of the barotropic component to the flow velocity can be ~~very significant~~ large. For example, using long-term measurements (1995 − 1996) from a mooring in the near-slope area of the Laptev Sea, Woodgate et al. (2001) showed that the contribution of the barotropic component to the velocity of the Arctic Ocean ~~B~~boundary ~~C~~current (AOBC) was equal to the contribution of the first three baroclinic modes. ~~To estimate~~

~~the volume flow rate they assumed that the~~ Assuming an average velocity based on the measurements in the upper 1200 m layer ~~was~~ of 4.5 cm/s and ~~the horizontal extension of the flow was~~ a width of 50 ~~to~~ 84 km. ~~At such values of the velocity and cross section of the flow~~ the volume flow rate was estimated at 5 ± 1 Sv. This ~~estimate differs from~~ is larger than our average estimate of the AW volume flow rate along 126 ºE (0.39±0.22, Table 1) by an order of

magnitude. Such a difference can be explained not only by the absence of a barotropic contribution in our case, but also by the fact that we took into account the volume transport of AW only (i.e. the cold, low-salinity surface layer was excluded) and considered certain season (August and September). Indeed, according to long-term measurements at 6 moorings on a section along 126 ºE, the AOBC volume flow rate varied from 0.3 Sv to 9 Sv (Pnyushkov et al.,

2018 b). Such a wide range in volume flow rate estimates is probably due to a combined effect of seasonal variability and mesoscale eddies (Pnyushkov et al., 2018 a).

The fact that seasonal variations can in some cases significantly affect the AW volume flow rates (see also the discussion ~~of different estimates of the AW volume flow rate~~ in Pnyushkov et al., 2018 b) is confirmed by a number of observations (Schauer et al., 2002a;

Beszczynska-Möller et al., 2012; Pnyushkov et al., 2018 b). For example, the volume flow rate of the AW in the northwestern part of the Barents Sea was 0.6 Sv ~~according to velocity measurements in summer~~ (Schauer et al., 2002a). This ~~estimate~~ agrees well with our estimate of the AW transport ~~of AW~~ in the St. Anna Trough, 0.79 ± 0.22 Sv (Table 2). However, the analysis of current velocity measurements in the winter season at the same section in the

northwestern part of the Barents Sea ~~gives~~ gave a completely different estimate of ~ 2.6 Sv (Schauer et al., 2002a).

       c) According to Dmitrenko et al. (2009), the BSBW ~~signal is~~ can be satisfactorily identified at 142°E. However, a "pattern" in the $\theta$-$S$ diagram far from the place of the BSBW entry into the Eurasian Basin can be regarded as the BSBW signal, if it maintains the similarity

with the "pattern" of the BSBW at the exit from the St. Anna Trough, that is, with the so-called "knee" (Dmitrenko et al., 2015). Our analysis showed that the "knee" is regularly observed at 103°E, while at 126°E it is ~~either~~ absent ~~or weakens strongly and~~ , weak or distorted. ~~Apparently this is quite natural,~~ This may be expected since the flow velocity is small, and the BSBW covers a distance from 103°E to 126°E for 1−2 years. However, despite of such a long travel time, Fram

Strait branch is well identified not only at 126°E, but also further along the slope. ~~It seems acceptable to associate this situation with characteristic features of~~ This suggests stronger transformation and mixing of, primarily, the BSBW. The BSBW transformation can be due to various reasons, including mixing with the FSBW caused by thermohaline intrusive layering at absolutely stable stratification (Merryfield, 2002; ~~Kuzmina et al., 2013;~~ Kuzmina et al., 2014;

Kuzmina, 2016~~, Zhurbas N., 2018; Kuzmina et al., 2018, 2019~~). ~~Indeed, the intrusive layering in the ocean influences the processes of exchange and mixing of various water masses (see, e.g., Stern, 1967; Fedorov, 1976; Joyce, 1980; Zhurbas et al., 1993; Rudels et al., 1999; Kuzmina, 2000; Walsh and Carmack, 2003). Other reasons for the BSBW signal disappearance may be:~~ , the influence of the slope topography, the impact of local counterflows near the slope (see, for

example, Pnyushkov et al., 2015), lateral convection (Ivanov and Shapiro, 2005; Ivanov and Golovin, 2007; Walsh et al., 2007), the impact of the Arctic Shelf Break Water (Aksenov et al., 2011; Ivanov and Aksenov, 2013) and  mixing due to eddies (Schauer et al., 2002; Dmitrenko et al., 2008; Aagaard et al., ~~2012~~ 2008; Pnyushkov et al., 2018a ). The understanding of the processes of transformation and mixing of the BSBW and FSBW is necessary to verify an

important concept proposed by Rudels, et al. (2015) that the BSBW supplies the major part of the AW to the Amundsen, Makarov and Canadian Basins, while the FSBW remains almost fully in the Nansen Basin.

## 5 Summary

      The $\theta$-$S$ properties and the volume flow rate estimates of the current carrying the AW in
the Eurasian Basin and St. Anna Trough were obtained based on the analysis of CTD data collected within the NABOS program in 2002–2015; additionally CTD transect PS-96 was considered. ~~All estimates are given in tabular form.~~

FSBW was present at all transects, including the two transects in the Makarov Basin (159°E), while the cold waters at the transects along longitudes 126°E, 142°E and 159°E, which can be associated with the influence of the BSBW, were observed in the depth range below 800 m and had little effect on the spatial structure of isopycnic surfaces and horizontal gradient of density. It is shown using $\theta$-$S$ analysis that the BSBW signal, which is characterized by the knee-shape feature in coordinates $\theta$, $S$ and $\sigma_{\Theta}$, $S$ (see Fig.8), is either strongly weakened or not visible at the longitude 126°E (excluding the observations in 2002 at 126 °E), while the FSBW signal is well identified at 126ºE and further along the slope of the Eurasian Basin. Based on the revealed features of the temperature, salinity and density fields, it is suggested that east of 126ºE the geostrophic volume transport of AW is mainly provided by the FSBW.

~~In order to assess spatial variability of the AW geostrophic volume flow rate, standard statistical analysis was used. It is shown with a 80% probability that t~~ The geostrophic volume flow rate of AW increases (with 80% confidence) from the region of 31ºE−92ºE (0.5 ± 0.14 Sv) to the region of 94ºE−107ºE (1.09 ± 0.38 Sv), then decreases to the region of 126ºE (0.39 ± 0.14 Sv) and becomes small (0.03 ± 0.1 Sv) in the Makarov Basin (159ºE).

The temporal variability of hydrological parameters and of the AW volume flow rate is summarized as follows. The time series of $\theta_{max}$ had an absolute maximum in 2006–2008 that can be interpreted as a result of heat pulse in the early 2000s (Polyakov et al., 2011). In accordance with our analysis the time series of $\theta_{max}$ had a maximum in 2013 but only at the longitude 103°E (Table 1 and Fig.10). The time series of $S(\theta_{max})$ display a trend of ~~an~~ increase of AW salinity ~~in 2006 2008 and 2013 also~~ over time, that can be referred to as a AW salinization in early 2000s. Moreover the salinity increases almost monotonously in the period from 2003 to 2013 at 142ºE . It is important to underline also that the maxima of $\theta_{max}$ and $S(\theta_{max})$ in 2006 and 2013 (103°E) are accompanied by the volume flow rate highs. A significant increase in geostrophic volume flow rate identified in 2006 is shown to be caused by climate impact.

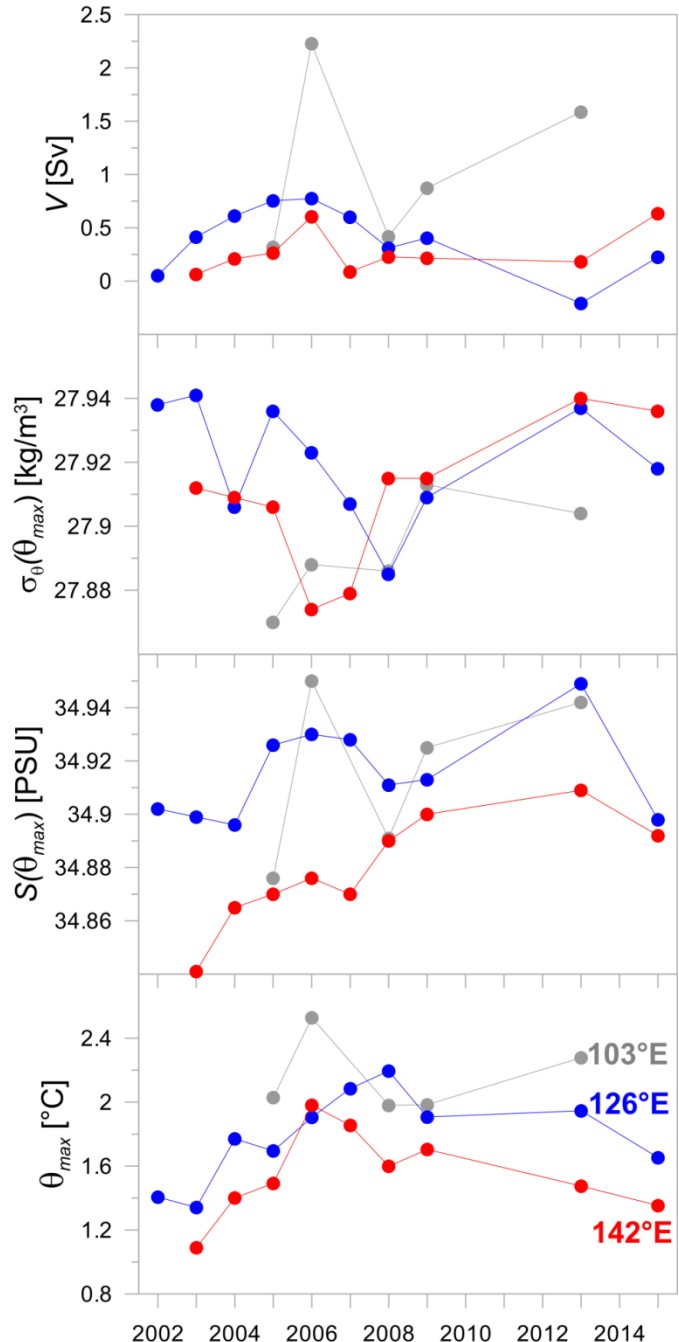

Fig. 10. Interannual variability of the maximum temperature $\theta_{max}$ and the related values of salinity $S(\theta_{max})$, potential density anomaly $\sigma_\theta(\theta_{max})$ and volume flow rate $V$ on the cross-slope transects at 103°E, 126°E and 142°E.

*Acknowledgments*. This research, including the approach development, data processing and interpretation, performed by Nataliya Zhurbas, was funded by Russian Science Foundation, project no. 17-77-10080. Natalia Kuzmina ($\theta$-$S$ analysis, statistical analysis, participation in discussion) was supported by the state assignment of the Shirshov Institute of Oceanology RAS (theme no. 0149-2019-0003).

The authors are very grateful to the NABOS group for providing the opportunity to use the CTD-data.

The authors are very grateful to the editor for evaluating the article and help in the work on the text and anonymous reviewers for useful comments.

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
