# Peer review of "Variability of the thermohaline structure and transport of Atlantic water in the Arctic Ocean based on NABOS hydrography data"

_Ocean Science, 2019_

## Referee Comment (RC1) · Anonymous Referee #1 · 28 Jun 2019

Review of the manuscript "Assessment of variability of the thermohaline sturcture and transport of Atlantic water in the Arctic Ocean based on NABOS CTD data" by N. Zhurbas and N. Kuzmina (OS-2019-54)

The manuscript present an interesting analysis of CTD data collected by NABOS and Polarstern expeditions in the Nansen, Amundsen, and Makarov basins. Data are underutilized and a new analysis of them should be welcomed. Careful analysis is done for evaluation the patterns and variability of temperature and salinity along these cross-sections. The authors used standard dynamic method to evaluate geostrophic water transports using cross-slope sections at several locations of the Arctic Ocean. Sev-

eral interesting conclusions were made. Specifically, a maximum of geostrohic water transport was found in 2006-2008, which coirresponds well to temperature and salinity variability in the eastern Arctic Ocean. However, I have several major comments and quite a few minor comments to the text. This may require substantial work, Rhus, I recomment major revision before the manuscript is published.

Major comments:

1. All findings of the paper should be placed in the context of the existing literature. I provided some references in Minor comments.

2. Care should be taken to separate spatial and temporal variability. See my minor comments for specifics.

Minor comments:

1. Abstract, lines 11-12. Sentences like "Estimates. . ." in the current for do not carry any useful information and should be modified or skipped completely.

2. Line 15: Instead of "on" one can use "along"

3. Abstract, lines 24-26: Same as in comment 1.

4. Line 34: 2000s, not 2000-ies.

5. Lines 35-38: Same as comment 1.

6. Intro: More recent papers related to analysis of NABOS data can be useful for the analysis and should be mentioned in the Intro: Pnyushkov, AV, et al. Heat, salt, and volume transports in the eastern Eurasian Basin of the Arctic Ocean from 2 years of mooring observations, Ocean Science, 2018 Dmitrenko, IA, et al. Atlantic water flow into the Arctic Ocean through the St. Anna Trough in the northern Kara Sea Journal of Geophysical Research: Oceans 120 (7), 5158-5178 Pnyushkov, AV, et al. Structure and variability of the boundary current in the Eurasian Basin of the Arctic Ocean, Deep Sea Research Part I: Oceanographic Research Papers 101, 80-97

7. Fig 1. One can use different colors to show different years.

8. Line 111: "original", not "primary".

9. Lines 134-148: The paper by Pnyushkov et al. 2018, Structure and dynamics of mesoscale eddies over the Laptev Sea continental slope in the Arctic Ocean, Ocean Science 14 (5), 1329-1347 can be useful in this context.

10. Lines 191-210: Materials of this paragraph should be viewed in the context of the recent paper by IA Dmitrenko et al. (2015) cited by the authors.

11. Lines 217-231: These estimates of water mass parameters may be obsolete considering strong changes which occurred over the recent decade or so (e.g. Polyakov et al. Greater role for Atlantic inflows on sea-ice loss in the Eurasian Basin of the Arctic Ocean, Science 356 (6335), 285-291)

12. Line 238: "Strip", not "stripe".

13. Fig. 8. Please use colors to separate profiles.

14. Lines 382-396. Somehow the authors should take into consideration temporal change of water masses at the selected locations vs. spatial changes. They can do that by analyzing repeated NABOS CTD sections and compare temporal and spatial changes.

15. Line 411: Please repeat your conclusion here.

16. Fig. 9: Please provide profiles in color.

17. Lines 432-439: Method of defining the area is not well described.

18. Table 1: Please include year as the third column for each line.

19. Lines 476-478: Please place these estimates in the context of paper by Pnyushkov et al. 2015.

20. Line 493: What authors are referred here? Why the distance of the AW core from

the slope is the key parameter for the flow dynamics?

21. Lines 511-522: Please place these results in the context of recent Pnyushkov et al 2018 paper.

22. Discussion of volume transports should incorporate sensitivity of these results to northward extension of sections which varied in time and space.

23. Lines 537-547: This paragraph should be placed in the context of existing mooring-based estimates of water transports across the Barents Sea and Fram Strait.

24. Lines 549-554: This paragraph should be placed in the context of the mooring-based estimates by Woodgate et al. 2001. Note, that my comments 23-24 will encourage the authors to discuss why their estimates are much lower than those coming from mooring observations e.g. Pnyushkov et al., Woodgate et al., Dmitrenko et al. Paragraph, lines 563-587 , gives some hint, but discussion is far from complete. Particularly, I am not happy with the authors' attempt to explain their low values of water transports by limited area of sections.

25. Lines 602-625: Please place these results in the context of recent Pnyushkov et al 2018 paper.

26. Line 618: "Pulse", not "impulse".

27. Summary: Please try to avoid sentences like lines 627-628, 632-633. Summary includes materials which were essentially a brief overview of the previous sections. Thus, I do not repeat my comments related to the previous sections here for Summary, but the authors should go through their summary and check whether my criticism in in the previous comments is applicable here.

---

## Referee Comment (RC2) · Anonymous Referee #2 · 1 Jul 2019

Comments on: "Assessment of variability of the thermohaline structure and transport of Atlantic water in the Arctic Ocean bases on NABOS CTD data" by Zhurbas & Kuzmina

General: This work uses CTD data from mainly the NABOS experiment to compute geostrophic velocities and transports of Atlantic water along the continental slope of the Arctic Ocean (the Eurasian Basin). A level of no motion is chosen either at 1000m or, in the case of St. Anna Trough, close to the surface to catch the expected bottom intensified flow. The variability of the Atlantic water properties and transport both along the path of the Atlantic water and between different years is presented. The inflows through Fram Strait (the Fram Strait Branch) and down the St. Anna Trough (the

[Figure]

Barents Sea Branch) are identified and discussed separately.

The manuscript is long and introduces many details, but some crucial discussions are missing. Only CTD data are used in the transport calculations, but no comparisons with other existing estimates are made, neither with current measurements at the slope, nor with transport estimates from the gateways, Fram Strait and the Barents Sea opening. The discussion about the variability of the properties of the Atlantic inflow branches is different. Here many comparisons with existing descriptions about the Atlantic water circulation are made, and most of the conclusions agree with previously published results. In spite of analyzing an extensive data set, both in time and space, few new results are presented.

Below some specific points are listed that should be looked at. They are given in order of appearance not significance. The authors way of writing is different from mine and I have not tried to make any specific, editorial changes. However, the manuscript would benefit from a tighter and more focused writing.

Specific points: line 12: Polarstern

line 45: change "increased" to higher.

line 47: The BSB is also cooled directly by the atmosphere.

line 52: change "observationally derived velocities" to direct current observations. There are also several estimates of the geostrophic transports through Fram Strait which could be cited.

line 55: I would remove "density driven" most readers know the meaning of geostrophic flow

line 57: Internal waves disturb the density surfaces and will therefore affect the geostrophic calculation.

line 62: "array"?
line 64: Interannual is temporal.

lines 66 to 70: are not needed, remove.

line 75: "famous" Perhaps an overstatement.

line 77: "locations. . . . . .are".

line 92: Equation 1 shows the thermal wind relations. The geostrophic velocities are given by -rhofv = -dp/dx, rhofu = –dp/dy

lines 102-104: If you go directly for the transport between the stations, it is just to integrate the specific volume anomalies of two neighbouring stations from the reference level to the surface.

lines 105-120: I think that this part is too detailed and more obscures than clarifies what is done. Moreover, the main problem with the geostrophic computations in the Arctic Ocean, especially near the continental slope, is not where the level of no motion is located but rather: Is there any level of no motion at all? The geostrophic estimates made of e.g. the transport of the West Spitsbergen Current in Fram Strait are always lower than estimates based on direct current measurements. There is a barotropic velocity component that is not captured by the geostrophic calculations.

lines 124-137: This long paragraph, indicating what should be, but is not, done, could be shortened considerably (or removed). I do not know what "masking" means in this context, but if you have a dense enough station net to resolve the eddies (or waves) the transports generated by the eddy will cancel. If it is not dense enough, there will be an error.

lines 148-149: The in situ density is what you use in geostrophic computations. It is not something that can be adopted or not at will.

lines 158-160: Here you claim that the potential density anomaly, not the in situ anomaly, determines the transport. Which is it? (see comment above) I think you

could shorten (or remove) lines 155-161.

lines 177-199: This should be shortened. Furthermore, I miss here references to Hanzlick and Aagaard (1980), Schauer et al. (2002a) and Dmitrenko et al. (2015) who discuss the recirculation of the Atlantic water in the trough and the outflow of dense water from the Barents Sea.

I also do not see why the Voronin Through is mentioned here. If the section on the top panel is the one shown on Figure 1, it does not extend to the Voronin Trough, and the bathymetric feature seen on the right is not a ridge separating the St. Anna Trough from the Voronin Trough.

lines 200-213: The properties of the inflows to the Arctic Ocean vary and I think that it is of little use to try to fit the water masses into strict definitions. The FSBW is in its upper part generally warmer and more saline than the BSBW and also in the deeper layers the BSBW shows up as a salinity minimum. However, some denser BSBW contributions can show up as high salinity contributions, but these inputs would sink to deeper levels below 1000m (Dmitrenko et al., 2105).

lines 213-237: The description of this section is difficult to understand because the sections shown in Figure 4 are too small to reveal any of the details discussed in the text. If I had not known this section from Schauer et al. (2002b) (not referred to here) I would not have been able to follow the details.

In general, the figures and the comparison between figures would be improved, if the horizontal scale is the same on all figures. As it now is, a section across the Eurasian Basin occupies the same width as a short section just down the slope.

lines 241-243: It is not appropriate to talk about a gravity current here. The outflow from the shelf, either from St. Anna Trough or farther east, has here reached its neutral density level and moves horizontally with the boundary current. After all, we do not refer to the North Atlantic Deep Water moving south along the North and South American

continental slopes as a gravity current.

lines 290-419: TS analysis. This part is too meandering, and it is difficult to find out what the authors want to communicate. Showing isopycnals (preferably sigma-1) in figure 8 might help. Also 8f should not have sigma-0 but sigma-1 as vertical axis. One question addressed here appears to be the contributions from the Barents Sea that is lost beyond the section at 103E. Are the NABOS stations from 126E shown in figures 8e & 8f taken to the bottom or only to 1000m? The "true" BSBW referred to by Dmitrenko et al. (2015) should at 126E be found below 1000m and if the stations do not extend deeper than 1000m it is no wonder if this contribution is not observed. In the upper part the TS shape of the NABOS stations looks like less saline Barents Sea Branch Water being intrusively mixed into the more saline Nansen Basin water column.

This section should be shortened and made clearer. It was nice to see the references to Dmitrenko et al., (2015) and Schauer et al. (2002b) referred to here, but they should be referred to earlier as should Schauer et al. (2002a) (not referred to).

lines 427-428: The stations at 82N in the St. Anna Trough shown on Figure 1 do not extend to the Voronin Trough. Are there other stations not shown on the map?

lines 420-545: This again is a long section and could be shortened. The most significant result here is the small transports and comparisons with direct current observations should be made. Woodgate et al. (2001) find a flow in the boundary current at the Laptev Sea slope of 5 Sv, which then splits with 2.5 Sv moving in the Amundsen Basin along the Lomonosov Ridge and 2.5 Sv crossing the ridge and flowing along the East Siberian Sea slope. Why are there so large differences in transports? The possibility of the presence of a barotropic component of the transport and also the effects of the sloping bottom should be discussed. There are also no comparisons with the measured and computed inflows through Fram Strait and over the Barents Sea.

The deduced recirculation in the Eurasian Basin is not new, but was suggested by Rudels et al. (1994) and for the warm Atlantic inflow in the 1990s a return flow along

the Lomonosov Ridge was observed in 1994 by Swift et al. (1997).

Summary: I think this manuscript could be published but not without a major rewrite. It should be shortened, especially the more general descriptions of the circulation and of the methods used, and the focus should be on the new findings.

---

## Editor Comment (EC1) · Ilker Fer (Editor) · 8 Jul 2019

Dear Nataliya,

Thank you for your efforts in analysing an extensive and useful data set. The synthesis will be of interest to the Ocean Science readers, but major revisions are needed.

The write-up needs substantial improvement. Please make an effort to improve the narrative. I attach a marked-up version with some comments you could consider (material to avoid, move, restructure etc.).

Regarding the analysis, I find it crucial to include a careful error estimate for the

geostrophic volume transport analysis, and a better discussion on the barotropic contribution.

Thanks, Ilker

Please also note the supplement to this comment:
https://www.ocean-sci-discuss.net/os-2019-54/os-2019-54-EC1-supplement.pdf

**Supplement:**

[revised manuscript text omitted]

---

## Author Comment (AC1) · 8 Jul 2019

Dear Reviewer #1,

We are very grateful for attention to the manuscript and useful remarks.
We will certainly take into account all comments and all papers which recommended while manuscript revision.
We will also present an extended discussion of the obtained values of geostrophic volume flow rate of the AW, including a comparison with estimates of volume flow rate that are given in the other works.

Now we would like to clarify some comments: numbers 14, 15, 17 and 18.

Comment # 14.
*14. Lines 382-396. Somehow the authors should take into consideration temporal change of water masses at the selected locations vs. spatial changes. They can do that by analyzing repeated NABOS CTD sections and compare temporal and spatial changes.*

This comment is not clear to us. In Fig. 8 and 9 are given *T, S*-diagrams and vertical profiles, which are obtained on the basis of measurements NABOS09 ( 2009 year). The only exception is the *T, S*-diagram in Fig. 8c, based upon the section of PS-96; this case is specifically mentioned – lines 355–358. Using the sections (NABOS09), spaced apart, we analyzed changes in the characteristic features of BSBW (namely, the change in the amplitude of the "knee" (Dmitrenko et al., 2015)) moving away from St. Anna Trough. Such an analysis is similar to the analysis of the variability of the thermohaline structure of intrathermoclinic eddies as they move away from the region of their origin. It should be also noted that the "knee" (Dmitrenko et al., 2015), observed on *T, S*-diagrams for BSBW, is a typical feature of BSBW. Therefore, such a "signal" on the *T, S*-curves may indicate the presence of BSBW in a particular observation area.
Will You be so kind to explain, what additional analysis You mean in this comment?

Comment # 15
*15. Line 411: Please repeat your conclusion here.*

Unfortunately, this comment is not clear to us. The text of the manuscript on line 411 and near it states that intrusions in ocean can be one of the main mechanisms for the exchange and mixing of water masses.

Comment # 17
*17. Lines 432-439: Method of defining the area is not well described*

Will You be so kind to explain this comment in more detail.

Comment # 18
*18. Table 1: Please include year as the third column for each line.*

The first column of the Table 1 contains the year of measurements: for example, NABOS06 means that measurements were made in 2006. Can we explain this in the caption of Table 1, and not add the year as a separate column?

---

## Author Comment (AC2) · 12 Jul 2019

Dear Prof. Fer,

We are very grateful for Your attention to the work, comments and help with the text of the manuscript. It is very important for us.

We will definitely present a discussion on the barotropic contribution. Before starting to work on this comment, we would like in terms of discussion to write a preliminary answer to the Referee #2. Clarifying some issues will help us, in particular, to consider in more detail the effect of barotropic additive on the volume flow rate of the AW. This

answer is currently under preparation.

We will also be grateful to You for an explanation of what do You mean by the estimation of the error of geostrophic volume transport. The fact is that under the NABOS program periodically (every year) in a certain area the only one (not two or three) section is made. In the work (Zhurbas, 2019, included in the list of references; now accepted for publication), the error associated with the determination of the reference horizon was analyzed in detail for some of the NABOS sections. This error did not exceed 14%. But apparently You mean an another assessment, namely, volume transport standard deviation. Therefore, we can obtain, for example, the average volume transport in a certain area over several years and estimate the deviation from this average value. Do You mean such an assessment?

Kind regards, Nataliya Zhurbas

---

## Author Comment (AC3) · 24 Jul 2019

Dear Referee #2,

We are very grateful for Your attention to work and comments. Of course, we will shorten the manuscript in accordance with Your comments, but we will try to keep a description of the method of estimating geostrophic volume flow rate, since we believe that this section is important for readers who make similar assessments in other areas of the ocean. We will also discuss in detail the effect of the barotropic velocity component on volume flow rate and take into account all the works recommended by You.

[Figure]

In this preliminary answer to Your comments we would like to discuss some questions. Here we want to focus on two comments.

1. You write: Lines 105-120: "I think that this part is too detailed and more obscures than clarifies what is done. Moreover, the main problem with the geostrophic computations in the Arctic Ocean, especially near the continental slope, is not where the level of no motion is located but rather: Is there any level of no motion at all? The geostrophic estimates made of e.g. the transport of the West Spitsbergen Current in Fram Strait are always lower than estimates based on direct current measurements. There is a barotropic velocity component that is not captured by the geostrophic calculations."

From our point of view, the choice of a reference horizon for estimating geostrophic volume flow rate is very important. Yes, we agree that there are the cases when the horizon, where the velocity equals to zero, does not exist (due to the barotropic velocity component). Therefore, it is important to find a horizon where the velocity is low. Such a horizon is located in a layer in which the horizontal density gradients are minimal. It is highlighted in the manuscript.

Regarding the barotropic velocity component: the effect of barotropic additives on volume flow rate is described below in the manuscript (lines $118-124$). According to our analysis, the CTD sections were made in the areas of the Arctic Ocean covered with ice, and, therefore, the wind-driven elevations of the free surface could not cause a large barotropic velocity component. Exceptions could occur in the eastern part of the Eurasian Basin, where the ice cover could be discontinuous. We will discuss these cases in detail in the article. We will also re-analyze the available information on the ice conditions during the measurement period under the NABOS program. The question is: do You suppose that even when the sea surface is covered with ice, a barotropic additive can significantly affect the volume flow rate near the slope?

2. You write: Lines 290-419: "TS analysis. This part is too meandering, and it is difficult to find out what the authors want to communicate. Showing isopycnals (preferably
sigma-1) in figure 8 might help. Also 8f should not have sigma-0 but sigma-1 as vertical axis. One question addressed here appears to be the contributions from the Barents Sea that is lost beyond the section at 103E. Are the NABOS stations from 126E shown in figures 8e & 8f taken to the bottom or only to 1000m? The "true" BSBW referred to by Dmitrenko et al. (2015) should at 126E be found below 1000m and if the stations do not extend deeper than 1000m it is no wonder if this contribution is not observed. In the upper part the TS shape of the NABOS stations looks like less saline Barents Sea Branch Water being intrusively mixed into the more saline Nansen Basin water column.

This section should be shortened and made clearer. It was nice to see the references to Dmitrenko et al., (2015) and Schauer et al. (2002b) referred to here, but they should be referred to earlier as should Schauer et al. (2002a) (not referred to)."

This paragraph is devoted to the analysis of the "signals" of the presence of different water masses in the various zones of the Basin and is necessary to clarify some results obtained in paragraph 3.1. It is important to note here that a pronounced signal clearly indicates that the water mass has entered the area of observation. However, the absence of a signal indicates one of the following: a) the water mass did not enter the area of observation; b) it entered the area of observation being highly transformed, namely, mixed with other waters. We will insert a brief explanation in the text.

If by the term sigma-1 You mean a potential density calculated from the 1000 m horizon, then all figures 8a-e should be redrawn, since the potential temperature was calculated from the zero horizon. Moreover, each figure of Fig.8 contains a T-S diagram, based upon the CTD profiles, obtained in the St. Anna Trough. However, the depth of the Trough, where these CTD profiles were obtained, was only 600 m (see Fig. 9), and the knee-like feature was observed in the depth range of 300−600 m (see Fig. 9). It should be also taken into account that in paragraph 3.1a all transects were drawn using the sigma-theta from the zero horizon, and it is logical to observe a certain correspondence in the presentation of data. It is also important that such a presentation of our results makes it convenient to compare them, for example, with the results of the

work (Dmitrenko et al., 2015), in which the zero reference horizon was used to estimate potential density.

Now we think that the most optimal solution is to draw lines of equal density calculated from the zero horizon in Fig. 8a-e, and leave Fig. 8f unchanged.

In the same comment You write: "One question addressed here appears to be the contributions from the Barents Sea that is lost beyond the section at 103E."

There is no such statement in our article! Possibly we could not accurately express our reasoning.

Based on the analyzed data, we obtained (to be precise) that the BSBW signal (the main part of the "knee") either weakens strongly and distorted towards 126E (similarity is lost with a "perfect" knee signal; see Fig. 1 in the attachment), or is not observed at all at this longitude (Fig. 8 of the manuscript). We assume that such a situation is typical. It suggests that the BSBW and FSBW begin to mix intensively immediately after 103E. However, the FSBW signal is well identified at 126E and further along the slope of the Eurasian Basin (and even in the Makarov Basin), while we cannot say the same about the BSBW signal. Naturally, we also observed rare cases of deviation from the typical situation. For example, the observations were exceptional in 2002, when a rather intense BSBW signal was observed at 126E (see Fig. 2 in the attachment to the answer). The analysis of rare atypical cases is beyond the scope of this work. We will clarify our conclusions on the results obtained in the manuscript. We also do not exclude that additional studies are required based on new CTD data to verify our conclusion about the typical BSBW signal pattern at 126E and further along the slope of the Eurasian Basin.

Further You write: "Are the NABOS stations from 126E shown in figures 8e & 8f taken to the bottom or only to 1000m? The "true" BSBW referred to by Dmitrenko et al. (2015) should at 126E be found below 1000m and if the stations do not extend deeper than 1000m it is no wonder if this contribution is not observed."

NABOS-09 CTD data at 126E in 5 cases reached a depth of 1100 m, and in one case 1800 m. In the caption to Fig. 8 it is signed that the depth range for the curves "12" is 500−1200 m. In this figure we have presented the T-S diagram of curves "12" up to the maximum depth of measurements for each profile. It should also be noted that some data of the NABOS CTD profiling, on the basis of which our conclusions were made, reached the depths of more than 1500 m.

We would like to draw Your attention to the fact that in the paper we consider only the main part of the "knee" (BSBW, region II in Fig.8 e). We did not analyze of the transformation of the "true" mode of the BSBW (see lines 378−380).

Further You write: "In the upper part the TS shape of the NABOS stations looks like less saline Barents Sea Branch Water being intrusively mixed into the more saline Nansen Basin water column."

Fig. 8 provides useful information on the BSBW signal and its transformation as it moves away from St. Anna Trough. From our point of view the intrusive layering can play an important role in the transformation of the BSBW waters (see, for example, (Kuzmina et al., 2011)). To explain the mechanism of this process, targeted research is needed.

Kind regards, Nataliya Zhurbas and Natalia Kuzmina

[Figure]

Fig. 1.

θ,°c

NABOS02, 126° E
st. 03-06

strong knee signal

S

**Fig. 2.**

[Figure]

---

## Author Comment (AC4) · 24 Jul 2019

Dear Referee #1,

We would like to make a small addition and answer Your comment #14 more correctly.

You write: 14. Lines 382-396. "Somehow the authors should take into consideration temporal change of water masses at the selected locations vs. spatial changes. They can do that by analyzing repeated NABOS CTD sections and compare temporal and spatial changes."

This paragraph is devoted to the analysis of the "signals" of the presence of different

water masses in the various zones of the Basin. It is important to note here that a pronounced signal clearly indicates that the water mass has entered the area of observation. However, the absence of a signal indicates one of the following: a) the water mass did not enter the area of observation; b) it entered the area of observation being highly transformed, namely, mixed with other waters. We will insert a brief explanation in the text.

Focusing on the BSBW signal in St. Anna Trough (it can be assumed that the signal of 2009 is typical of BSBW), we are looking for a similar (based on T-S curves) signal in other areas of the Arctic Basin. And we analyze the data obtained in different years.

Briefly, our analysis comes down to the following results:

Based on the analyzed data, we obtained (to be precise) that the BSBW signal (the main part of the "knee") either weakens strongly and distorted towards 126E (similarity is lost with a "perfect" knee signal; see Fig. 1 in the attachment), or is not observed at all at this longitude (Fig. 8 of the manuscript). We assume that such a situation is typical. It suggests that the BSBW and FSBW begin to mix intensively immediately after 103E. However, the FSBW signal is well identified at 126E and further along the slope of the Eurasian Basin (and even in the Makarov Basin), while we cannot say the same about the BSBW signal. Naturally, we also observed rare cases of deviation from the typical situation. For example, the observations were exceptional in 2002, when a rather intense BSBW signal was observed at 126E (see Fig. 2 in the attachment to the answer). The analysis of rare atypical cases is beyond the scope of this work. We will clarify our conclusions on the results obtained in the manuscript.

A detailed analysis of the spatial and temporal variability of the BSBW signal in the Arctic Basin (the presentation of results for all sections in different years, the criteria for recognition of the "knee" structure, etc.) is beyond the scope of this paper. Paragraph 3.1b is necessary to understand some of the results obtained in paragraph 3.1a.

Kind regards, Nataliya Zhurbas and Natalia Kuzmina

$\theta,\ ^{\circ}c$

NABOS03, 126 $^{\circ}$E
st.18-20

week knee signal

S

**Fig. 1.**

θ,°c

NABOS02, 126° E
st. 03-06

strong knee signal

S

Fig. 2.

---

## Author Response (AR1)

**The Editor**

Dear Dr. Fer,

Many thanks for your kind attitude to our work. All your comments have been accounted for as follows.

*The write-up needs substantial improvement. Please make an effort to improve the narrative. I attach a marked-up version with some comments you could consider (material to avoid, move, restructure etc.).*
All your comments (the sticky notes) in the attached marked-up version were accepted.

*Regarding the analysis, I find it crucial to include a careful error estimate for the geostrophic volume transport analysis, and a better discussion on the barotropic contribution.*
We added to Table 1 and 2 the mean values of the geostrophic volume transport estimates along with the 95% confidence intervals. The barotropic contribution was thoroughly considered in the Discussion chapter, lines 603-622.

**Reviewer 1**

Dear Reviewer 1,

Thank you very much for your comprehensive review. All your comments have been accounted for as follows.

*1. All findings of the paper should be placed in the context of the existing literature. I provided some references in Minor comments.*

*2. Care should be taken to separate spatial and temporal variability. See my minor comments for specifics.*

*Minor comments:*

*1. Abstract, lines 11-12. Sentences like "Estimates..." in the current for do not carry any useful information and should be modified or skipped completely.*
The sentence was dropped.

*2. Line 15: Instead of "on" one can use "along"*
"On" has been changed for "along" is several similar cases.

*3. Abstract, lines 24-26: Same as in comment 1.*
We left it lines as is.

*4. Line 34: 2000s, not 2000-ies.*
Corrected.

*5. Lines 35-38: Same as comment 1.*
We included to here the mean values of the transport estimates.

*6. Intro: More recent papers related to analysis of NABOS data can be useful for the analysis and should be mentioned in the Intro: Pnyushkov, AV, et al. Heat, salt, and volume transports in the eastern Eurasian Basin of the Arctic Ocean from 2 years of mooring observations, Ocean Science, 2018 Dmitrenko, IA, et al. Atlantic water flow into the Arctic Ocean through the St. Anna Trough in the northern Kara Sea Journal of Geophysical Research: Oceans 120 (7), 5158-5178 Pnyushkov, AV, et al. Structure and variability of the boundary current in the Eurasian Basin*

*of the Arctic Ocean, Deep Sea Research Part I: Oceanographic Research Papers 101, 80-97*
We included the recommended references to the Introduction.

*7. Fig 1. One can use different colors to show different years*
This would not be convenient because e.g. the 126°E section was repeated in 10 different years, so the colors would merge.

*8. Line 111: "original", not "primary".*
"Primary" was dropped.

9. Lines 134-148: The paper by Pnyushkov et al. 2018, Structure and dynamics of mesoscale eddies over the Laptev Sea continental slope in the Arctic Ocean, Ocean Science 14 (5), 1329-1347 can be useful in this context.
We included the reference to the Discussion chapter.

*10. Lines 191-210: Materials of this paragraph should be viewed in the context of the recent paper by IA Dmitrenko et al. (2015) cited by the authors.*
This paragraph was re-worked accordingly.

*11. Lines 217-231: These estimates of water mass parameters may be obsolete considering strong changes which occurred over the recent decade or so (e.g. Polyakov et al. Greater role for Atlantic inflows on sea-ice loss in the Eurasian Basin of the Arctic Ocean, Science 356 (6335), 285-291)*
The difficulties in definitions of water mass parameters due to climate change are addressed in Discussion chapter, lines 553-563. Reference to Polyakov et al. (2017) was included to there.

*12. Line 238: "Strip", not "stripe".*
Corrected

*13. Fig. 8. Please use colors to separate profiles.*
Done.

*14. Lines 382-396. Somehow the authors should take into consideration temporal change of water masses at the selected locations vs. spatial changes. They can do that by analyzing repeated NABOS CTD sections and compare temporal and spatial changes.*
The temporal change of water masses and transport has been discussed in Chapter 3.3. The spatial change was addressed in Abstract (lines 29-37), Chapter 3.1 (lines 137-406) and Discussion (lines 630-660). Also please see our previous reply to your comments during the interactive discussion.

*15. Line 411: Please repeat your conclusion here.*
Done (see lines 369-371).

*16. Fig. 9: Please provide profiles in color.*
Done.

*17. Lines 432-439: Method of defining the area is not well described.*
Now the method is well defined (see lines 399-400).

*18. Table 1: Please include year as the third column for each line.*

Year is given in the first column (see the Table 1 caption).

*19. Lines 476-478: Please place these estimates in the context of paper by Pnyushkov et al. 2015.*

The reference to Pnyushkov et al. (2015) was added.

*20. Line 493: What authors are referred here? Why the distance of the AW core from the slope is the key parameter for the flow dynamics?*
AW is not a passive mixture, it can affect the basin's dynamics. The distance of the AW core to the slope controls the flow instability. We added an explanation and a reference (lines 440-441).

*21. Lines 511-522: Please place these results in the context of recent Pnyushkov et al. 2018 paper.*
The reference to Pnyushkov et al. (2018) was added.

*22. Discussion of volume transports should incorporate sensitivity of these results to northward extension of sections which varied in time and space.*
Done (see lines 583-585 in Discussion chapter).

*23. Lines 537-547: This paragraph should be placed in the context of existing mooring-based estimates of water transports across the Barents Sea and Fram Strait.*
Done (see lines 567-629 in Discussion chapter).

*24. Lines 549-554: This paragraph should be placed in the context of the mooring- based estimates by Woodgate et al. 2001. Note, that my comments 23-24 will encourage the authors to discuss why their estimates are much lower than those coming from mooring observations e.g. Pnyushkov et al., Woodgate et al., Dmitrenko et al. Paragraph, lines 563-587, gives some hint, but discussion is far from complete. Particularly, I am not happy with the authors' attempt to explain their low values of water transports by limited area of sections.*
Done (see lines 567-629 of the Discussion Chapter where a thorough comparison of the geostrophic volume flow rate estimates obtained in this work with the other studies is considered.

*25. Lines 602-625: Please place these results in the context of recent Pnyushkov et al2018 paper.*
Done (see Discussion chapter).

*26. Line 618: "Pulse", not "impulse".*
Corrected.

*27. Summary: Please try to avoid sentences like lines 627-628, 632-633. Summary includes materials which were essentially a brief overview of the previous sections. Thus, I do not repeat my comments related to the previous sections here for Summary, but the authors should go through their summary and check whether my criticism in in the previous comments is applicable here.*
We dropped those sentences.

**Reviewer 2**

Dear Reviewer 2,

Thank you very much for your comprehensive review. All your comments have been accounted for as follows.

*Specific points:*
*line 12: Polarstern*
*line 45: change "increased" to higher.*
Corrected.

*line 47: The BSB is also cooled directly by the atmosphere.*
We pointed out it in the revised MS

*line 52: change "observationally derived velocities" to direct current observations.*
Corrected.

*There are also several estimates of the geostrophic transports through Fram Strait which could be cited.*
Some citations of the kind are given in Discussion chapter. If you mean a specific article, please let as know about its title and authors.

*line 55: I would remove "density driven" most readers know the meaning of geostrophic flow.*
Removed.

*line 57: Internal waves disturb the density surfaces and will therefore affect the geostrophic calculation.*
Internal waves are ageostrophic disturbances. For correct calculation of geostrophic velocities, the internal waves' disturbances should be removed. We focused on geostrophic estimates of the volume flow rates because they are less affected by internal waves.

line 62: "array"?
Corrected.

*line 64: Interannual is temporal*
That's right, 'temporal' was dropped.

*lines 66 to 70: are not needed, remove.*
We would prefer to keep these paragraph to formulate clearly the main goal of the paper and approach we used.

*line 75: "famous" Perhaps an overstatement.*
"famous" was removed.

Corrected.

*line 92: Equation 1 shows the thermal wind relations. The geostrophic velocities are given by -rhofv = -dp/dx, rhofu = –dp/dy*
Equation 1 was dropped (the Editor's advice).

*lines 102-104: If you go directly for the transport between the stations, it is just to integrate the specific volume anomalies of two neighbouring stations from the reference level to the surface.*
Why to the surface? Cold and low saline surface layer has nothing common with the AW. These lines were dropped (the Editor's advice).

*lines 105-120: I think that this part is too detailed and more obscures than clarifies what is done. Moreover, the main problem with the geostrophic computations in the Arctic Ocean, especially near the continental slope, is not where the level of no motion is located but rather: Is there any level of no motion at all? The geostrophic estimates made of e.g. the transport of the West Spitsbergen Current in Fram Strait are always lower than estimates based on direct current measurements. There is a barotropic velocity component that is not captured by the geostrophic calculations.*
To our mind, the choice of the no level motion level is always important because e.g. if we took it at the bottom in the St. Anna Trough, the geostrophic transport of BSBW would change sign. This paragraph was considerably shortened, and a discussion on the barotropic velocity

component was added to the Discussion chapter.

*lines 124-137: This long paragraph, indicating what should be, but is not, done, could be shortened considerably (or removed). I do not know what "masking" means in this context, but if you have a dense enough station net to resolve the eddies (or waves) the transports generated by the eddy will cancel. If it is not dense enough, there will be an error.*
This long paragraph was considerably shortened; now it does not contain mentioning of "masking".

*lines 148-149: The in situ density is what you use in geostrophic computations. It is not something that can be adopted or not at will.*
This sentence was dropped.

*Comment: lines 158-160: Here you claim that the potential density anomaly, not the in situ anomaly, determines the transport. Which is it? (see comment above) I think you could shorten (or remove) lines 155-161.*
This sentence was removed.

*lines 177-199: This should be shortened. Furthermore, I miss here references to Hanzlick and Aagaard (1980), Schauer et al. (2002a) and Dmitrenko et al. (2015) who discuss the recirculation of the Atlantic water in the trough and the outflow of dense water from the Barents Sea.*
This paragraph was shortened, and the references were added.

*I also do not see why the Voronin Through is mentioned here. If the section on the top panel is the one shown on Figure 1, it does not extend to the Voronin Trough, and the bathymetric feature seen on the right is not a ridge separating the St. Anna Trough from the Voronin Trough.*
The section on the top panel does not extend to the Voronin Trough, but it does show that the BSBW overflows a ridge-like elevation (i.e. a bathymetric feature) seen on the right and therefore one could easily imagine that the BSBW would overflow a ridge separating the St. Anna Trough from the Voronin Trough. We modified this peace of text accordingly.

*lines 200-213: The properties of the inflows to the Arctic Ocean vary and I think that it is of little use to try to fit the water masses into strict definitions. The FSBW is in its upper part generally warmer and more saline than the BSBW and also in the deeper layers the BSBW shows up as a salinity minimum. However, some denser BSBW contributions can show up as high salinity contributions, but these inputs would sink to deeper levels below 1000m (Dmitrenko et al., 2105).*
You are right. However, to calculate the volume flow rate of a specific water mass, one needs to have a strict definition of it. We added a discussion on the issue to the Discussion chapter.

*lines 213-237: The description of this section is difficult to understand because the sections shown in Figure 4 are too small to reveal any of the details discussed in the text. If I had not known this section from Schauer et al. (2002b) (not referred to here) I would not have been able to follow the details.*
We agree that the horizontal scale in Fig. 4 provides a poor resolution of the BSBW near the slope. We referred Schauer et al. (2002b) to here and described their results here and also in Discussion chapter.

*In general, the figures and the comparison between figures would be improved, if the horizontal scale is the same on all figures. As it now is, a section across the Eurasian Basin occupies the same width as a short section just down the slope.*

Given that the NABOS CTD section at 31°E, 2008 (Fig. 2), and the PS-96 section (Fig. 4) are 60 and 900 km long, respectively, the same horizontal scale on all figures will result in 15 times difference in horizontal size of the figures. We could not imagine it…

*lines 241-243: It is not appropriate to talk about a gravity current here. The outflow from the shelf, either from St. Anna Trough or farther east, has here reached its neutral density level and moves horizontally with the boundary current. After all, we do not refer to the North Atlantic Deep Water moving south along the North and South American continental slopes as a gravity current.*
We dropped mentioning of gravity flows everywhere in the MS except for the St. Anna Trough.

*lines 290-419: TS analysis. This part is too meandering, and it is difficult to find out what the authors want to communicate. Showing isopycnals (preferably sigma-1) in figure 8 might help. Also 8f should not have sigma-0 but sigma-1 as vertical axis. One question addressed here appears to be the contributions from the Barents Sea that is lost beyond the section at 103E. Are the NABOS stations from 126E shown in figures 8e & 8f taken to the bottom or only to 1000m? The "true" BSBW referred to by Dmitrenko et al. (2015) should at 126E be found below 1000m and if the stations do not extend deeper than 1000m it is no wonder if this contribution is not observed. In the upper part the TS shape of the NABOS stations looks like less saline Barents Sea Branch Water being intrusively mixed into the more saline Nansen Basin water column.*
*This section should be shortened and made clearer. It was nice to see the references to Dmitrenko et al., (2015) and Schauer et al. (2002b) referred to here, but they should be referred to earlier as should Schauer et al. (2002a) (not referred to).*
This part was thoroughly re-worked. The sigma-0 isopycnals were added to Fig. 8. To our mind, the sigma-0 is more appropriate to here than sigma-1 because most of TS curves in Fig. 8 corresponds to the upper 1000m layer and it is better for comparison with Figs. 2-6 where the sigma-0 contours are presented. Some NABOS stations from 126E shown in Figs 8e and 8f were taken to 1800m. The shape of the curves in Fig. 8f will not change principally if sigma-1 is taken instead of sigma-0 for the vertical axis. All the references imaginable were added. Some results of the TS analysis were discussed in the Discussion chapter. Also please see our previous reply to your comments during the interactive discussion.

*Comment: lines 427-428: The stations at 82N in the St. Anna Trough shown on Figure 1 do not extend to the Voronin Trough. Are there other stations not shown on the map?*
Mentioning of the Voronin Trough to here was dropped.

*lines 420-545: This again is a long section and could be shortened. The most significant result here is the small transports and comparisons with direct current observations should be made. Woodgate et al. (2001) find a flow in the boundary current at the Laptev Sea slope of 5 Sv, which then splits with 2.5 Sv moving in the Amundsen Basin along the Lomonosov Ridge and 2.5 Sv crossing the ridge and flowing along the East Siberian Sea slope. Why are there so large differences in transports? The possibility of the presence of a barotropic component of the transport and also the effects of the sloping bottom should be discussed. There are also no comparisons with the measured and computed inflows through Fram Strait and over the Barents Sea.*
Please see the Discussion chapter where the issue was thoroughly discussed. It's not clear to us what do you mean by the term "sloping bottom effects". Please explain. If you believe that there is a need to include more references to the Discussion chapter or elsewhere in the manuscript, please provide information about the titles and authors.

*The deduced recirculation in the Eurasian Basin is not new, but was suggested by Rudels et al. (1994) and for the warm Atlantic inflow in the 1990s a return flow along the Lomonosov Ridge was observed in 1994 by Swift et al. (1997).*

We know that this recirculation has been known for a long time. We estimated the transport and explained what this transport should be attributed to. A reference to Rudels et al. (1994) and Swift et al. (1997) was included (see line 224).

---

## Author Response (AR2)

Dear Dr. Fer,

We did a lot of work on the manuscript.

We took into account all comments of the reviewers, reduced repetitions in the manuscript, explained the results of a statistical analysis of the volume flow rate changes over time and along the slope of the Eurasian basin, added the necessary scientific articles to the list of literature and compared previous results with our results.

We also tried to convince reviewer #1 of the need for analysis of temperature and salinity on CTD sections.

We are sending clean and marked-up versions of the article and detailed answers to the reviewers.

In the marked-up version the pieces of text excluded from the manuscript (crossed out) are marked also in green for convenience (to be clearly visible). Additions to the manuscript are marked in yellow.

Kind regards,

Nataliya Zhurbas and Natalia Kuzmina

**Reply to Referee #1.**

**1.**We were very upset and surprised by Your second review of the revised manuscript.

We tried to take into account all your comments. In section 1 we referred to the papers You recommended. In particular, Dmitrenko et al. (2015) is mentioned 15 times in our manuscript! Pnyushkov et al. (2015, 2018 a, b) are mentioned 9 times in total! Analyzing the results we obtained, we made a comparison with the results of other authors (see, for example, sections 3.1, 3.2, 3.3 and Discussion). Actually, this is the fulfillment of your requirement: "All findings of the paper should be placed in the context of the existing literature".

**Remark**

Pnyushkov et al. (2015) is devoted to measurements of the coastal current velocity in the Eurasian Basin based on moored instruments. Dual-core structure of the AW is also analyzed in this work by sections of the temperature field. Our work is devoted to other studies. First of all, we were interested in the estimation of velocities by the density field based on the dynamic method; such estimations were presented in Pnyushkov et al. (2015) for three NABOS sections (2006). However, unfortunately, this paper does not provide estimates of the volume flow rates (transport) to which our manuscript is devoted (see also below).

The work of Pnyushkov et al. (2018) was also very useful for us. We referred several times to this paper in the Discussion.

In this version of the manuscript, according to Your recommendations, we have changed the

Introduction.

**2.**We were also surprised by the following. In the first review You wrote:

«The manuscript present an interesting analysis of CTD data collected by NABOS and Polarstern expeditions in the Nansen, Amundsen, and Makarov basins… Careful analysis is done for evaluation the patterns and variability of temperature and salinity along these cross- sections…»

But in the second review You write:

«Remove from the text discussion of temperature, salinity and (mostly) density changes.»

But is it really possible to estimate the volume flow rates of water masses, discuss the obtained estimates and compare them with the estimates of other authors without determining the water masses?

Here is what Pnyushkov et al. (2018), section 6.3, say about this:

…"The estimates of AW transports are also sensitive to the temperature and salinity ranges used for the identification of this water."

Note also that the addition of information on the definition of the BSBW and FSBW in the manuscript was necessary. In particular, we had to compare our interpretation of the PS-96 section and the interpretation given by Schauer et al. (2002).

Now we have explained in the Introduction why it is important to analyze the temperature and salinity of water masses, citing Pnyushkov et al. (2018).

**3**.Regarding the separation of spatial and temporal variability.

Firstly, we would like to receive a more specific remark: it is not clear to us what exactly You mean.

Secondly, we asked You a question (during the Discussion period): what kind of analysis would You like to see? And we did not receive an answer. We revised section 3.1.2 and clarified the conclusions of it. In support of our conclusions, a figure (new Fig.9) is now presented in the manuscript, which shows the differences in T-S values at different sections at different times.

Thirdly, in accordance with Your remarks and the remarks of the Scientific Editor of the manuscript, in the manuscript version 2 we have calculated the average volume flow rates for the year according to the available data and presented confidence intervals (Table 1).

It is also worth mentioning that CTD sections obtained by the NABOS program allow estimating spatial and temporal variability: sections are made in the same zones of the Arctic Basin annually in August−September.

**Consider Your comments in the second review.**

**You write**:

1.«My previous comment #1 required the authors to place their research in the context of what has been done in this research area in the past. This is essential requirement because it would

identify new and exciting results coming from the authors' research. E.g. after reading the entire Section 3.1 I left wondering what new is in this section. The same is about the part of Section 3.2 devoted to temperature, salinity, and density changes. This is also true for Section 3.3 which, despite the title, does not discuss water transport changes. All these three sections include multiple repetitions.»

**Answer:** Section 3.1.1 presents CTD sections which make it possible to judge the change in the T, S, and σ characteristics of the flow along the slope. Such a sequence of sections was not previously published. In conclusion of section 3.1.1, we noted:

«On the transects made along 126°E, 142°E, and 159°E, the slopes of isopycnic surfaces indicating the baroclinic flow, were observed generally in the depth range of 200–400 m, that is in the area occupied by the FSBW… The cold waters on the transects along 126°E, 142°E and 159°E, which can be associated with the BSBW, had a minimum temperature above -0.5 °C, were observed in the depth range below 800 m and had a little effect on the spatial structure of isopycnic surfaces.»

Could You tell us in what papers this conclusion was presented earlier?

Section 3.1.2 says that

«The results presented in Fig. 8 show that the BSBW signal which is characterized by the knee-shape diagram in coordinates $\theta$-S and $\sigma_\theta$-S, is not visible at 126°E. This is consistent with the conclusion formulated in Subsection 3.1.1 that by 126°E the BSBW is not accompanied by any noticeable perturbations of isopycnals. Moreover, given the characteristic feature of the $\theta$-S structure of BSBW in the St. Anna Trough (curves 1−4 in Fig. 8a) was observed in other years, we carried out a similar analysis using all available CTD data and found that the BSBW signal is either strongly weakened or not visible at this longitude.»

Would You, please, give a reference to the article where this conclusion was obtained earlier?

Section 3.2 presents a table in which not only the volume flow rates, but the hydrological parameters are given. Without such a table, we would not be able to prepare Fig. 10, which contains new information. Table 1 is unlikely to become better if the hydrological parameters are removed from it. For example, from Table 1 (see also Fig. 10) it follows that in the period under consideration the salinity in the core AW in the 142°E section increased with time. Would You, please, give a reference to a paper where such result was presented earlier?

Now, in the new version of the manuscript (Subsection 3.2), we discuss in detail the causes of "noise" in the flow estimates of AW transport and, based on the standard statistical approach, determine the change in AW transport along the slope.
It is shown also (Subsection 3.2) that the significant increase in geostrophic transport in 2006 most likely to associate with climate impact (see also below ).
Also, based on your comment about the parameter $X_{\theta max}$ in the first review we made an edit to the manuscript .

Section 3.3 briefly describes the graphs of Fig. 10, which is common in the scientific papers.

Our estimates of the volume flow rates are discussed in section 4. There is also a comparison of our estimates of transport with estimates of other authors.

In accordance with Your remark about repetitions in different sections of the manuscript, we have shorten the text and excluded repetitions.

**Further You write**:

2) «I do not think that the part of "Results" devoted to temperature, salinity and density changes carries any new information. One of the major emphases is the attempt to separate the Barents and Fram Strait branches of the Atlantic Water. First, there is an extensive literature on that subject which is not presented in the manuscript. Secondly, I doubt that using just temperature and salinity data would allow successful separation of the water masses because of their strong mixing along the Eurasian slope which makes the boundary between these water masses extremely eluded. Chemical data analysis is the key for the successful separation.»

**Answer**: We have answered this remark above. In addition we can say that we have not searched for a method for the exact separation of the BSBW and FSBW. We have only showed that FSBW and BSBW are being mixed, and this process can lead to a decrease in the BSBW signal in the density field, and, consequently, to a decrease of geostrophic flow. To determine BSBW, we relied on latest papers in peer-reviewed scientific journals.

**Further You write:**

3) «Analysis of water transports based on geostrophic estimates is the golden nugget of the manuscript. However, now it is buried under unnecessary (to my view) discussions of temperature and salinity.»

**Answer**: We appreciate Your positive assessment of this part of our work. However, we consider our paper as an integral study, which consists of two interconnected parts.

**Further You write**:

4) Still a lot should be done to demonstrate that the estimates of water transports presented in the paper are not affected by spatial variability

**Answer**: This remark is not clear to us. Please explain specifically what You mean by the term "spatial variability". We mean by this term the change in average flow rate AW along the slope.

According to the analysis of the data available, we have shown that the geostrophic volume flow rate of the AW varies along the slope. Unfortunately, the amount of data is not so large, so in some cases we had to calculate average values of transport based on data from several neighboring sections, for example, along 31°E and 90°E, that is, we worked with limited statistical material. We determined confidence intervals with a probability of 95%. The decrease in the volume flow rate after 94°E−107°E is evidenced by the fact that the confidence intervals of the volume flow rate for 94°E−107°E and 142°E do not overlap: 0.46 Sv≤V(94°E−107°E) ≤1.72 Sv and 0.12 Sv≤V(142°E) ≤0.44 Sv.

Now, in the present version of the manuscript, we have given the necessary explanation (Subsection 3.2). Additionally, in the present version of the manuscript, we have estimated confidence intervals with a probability of 80%. In this case it was found that the average geostrophic volume flow rate increases in the confluence zone of the two AW branches, and then decreases as the AW moves along the slope (see Subsection 3.2).

If You think that it is necessary to provide additional evidence that the increase of the volume flow rates in 2006 was due to climate impact, and not the influence of eddies, then we can say the following. According to the Table 1, the volume flow rates in 2006 for sections along

94°E−103°E, 126°E, 142°E significantly exceeded the corresponding upper limits of confidence intervals. This indicates that: a) such events (a strong deviation from the average value) are quite rare; b) it is unlikely that synoptic eddies simultaneously increased the volume flow rates in three sections far from one another. We noted now this in the manuscript (Subsection 3.2).

**At the end of the second review You write:**

«My recommendations would be:
Focusing the manuscript on analysis of water transports based on geostrophic estimates. Remove from the text discussion of temperature, salinity and (mostly) density changes.»

**Answer**. Firstly, the work consists of two connected parts: from our point of view, mixing of the two AW branches can cause a decrease in baroclinicity, and, consequently, a decrease in geostrophic volume flow rate. Secondly, the authors still have the right to an independent vision of the presentation of their scientific work.

**You also write:**

« Provide solid evidence that the estimates carry signal and not just synoptic noise.»

**Answer**: We work with a limited statistical material, and all estimates are probabilistic, therefore the term "solid evidence" is not applicable to such estimates (see above confidence intervals and Subsection 3.2 in the manuscript). The estimated average values of the volume flow rate and confidence intervals show that the variability of the average geostrophic volume flow rate along the slope and its strong increase in 2006 indicate a signal, not noise in the data (see above). More accurate estimates can be obtained on the basis of longer time series or such an experiment on the sea, according to the results of which it will be possible to describe a three-dimensional structure of the current, that is, to distinguish the mean flow and synoptic eddies.

**Remark**. Synoptic eddies are generally satisfactorily described in the quasi-geostrophic approximation and, therefore, the horizontal velocities of the eddies are determined from geostrophic relations as well as the mean current velocity. If we are working with a section, it is impossible to exclude quasi-geostrophic large scale disturbances by averaging the primary CTD data; at best only ageostrophic disturbances (internal waves or sub-mesoscale eddies) can be filtered out.

**You also write:**

Compare these estimates with the existing ones (e.g. recent Pnyushkov et al. studies). I would like to see cross-sections of these inferred currents.

**Answer**:In Discussion we compared our volume flow rate estimates with those obtained in other studies, including Pnyushkov et al. (2018b).

 Now, in the present version of the manuscript, we have made additions: we compared our estimates with the estimates obtained by different authors in the Fram Strait (see Section 4).

Here we present for You velocity calculations based on the dynamic method along section 31°E (2006, NABOS) using the original data (see below **Appendix)**. These velocities can be compared with Fig. 8. Pnyushkov et al. (2015). There is no explanation of which method of averaging the primary data was used in this paper.  We have also estimated the volume flow rate at ~30°E based on Fig. 8, upper fragment ( Pnyushkov et al., 2015). Using different methods of

color recognition and color matching with scale in the Figure 8, we obtained the following transport evaluations: 0.15 Sv-0.4 Sv. It would be very interesting to estimate the transport based on digital data.

**You also write:**

Do analysis in depth of spatial changes versus temporal changes.

**Answer:** The manuscript contains now all estimates indicating the spatial and temporal variability of the volume flow rates. We used standard methods for separating spatial and temporal variability based on available data (see Subsections 3.1.2. and 3.2). If You have specific suggestions for clarifying the analysis, we are ready to discuss them.

**APENDIX**

To demonstrate the calculated geostrophic velocities and explain why we prefer to deal with geostrophic transport rather than geostrophic velocities, let's address e.g. the NABOS-2006 section at 31°E (see Figure 1).

[Figure]

Figure 1. Temperature (a), potential density (b) and geostrophic velocity relative to 800m reference level (c, d) vs distance and depth for NABOS-2006 transect across the slope at 31°E; (e) – geostrophic velocity at 200m depth vs distance. The geostrophic velocities were calculated using finite differences between neighbor stations without any averaging/interpolation/gridding of density field and were not averaged afterwards: (c) – all stations were used for geostrophic calculations; (d) – one station was dropped (marked by a red cross).

   Figure 1 (e) demonstrates clearly that dropping of a one station results in a decrease of the maximum geostrophic velocity on the transect from $u_{max} = 0.16$ m/s to $u_{max} = 0.10$ m/s while the geostrophic transport remains unchanged at $V = 0.57$ Sv.

   It seems exemplarily to compare our geostrophic transport/velocity estimates for the NABOS-2006_31°E transect (Figure 1) with the estimates by Pnyushkov et al. (2015) carried out for the same transect relative the 0m reference level (Figure 2). Pnyushkov et al. (2015) did not present results of the geostrophic transport calculations, but from Figure 2 it can be roughly assessed at $0 < V < 0.5$ Sv (more accurate estimates see, please, above) which, despite of the difference in the reference level used, fits more or less our estimate $V = 0.57$ Sv. On the other hand, Pnyushkov et al. (2015) got $u_{max} = 0.05 - 0.06$ m/s (see Figure 2) which is considerably smaller than ours $u_{max} = 0.10 - 0.16$ m/s. That is, estimates of geostrophic velocity are sensitive to a subjective choice of averaging/interpolation/gridding procedures applied to the density field while the geostrophic transport estimates are practically insensitive to the choice.

[Figure]

Figure 2. Temperature (contours, °C) and geostrophic velocity (colours, cm/s) relative to 0m reverence level vs latitude and depth for the same NABOS-2006 transect at 30°E taken from Pnyushkov et al. (2015).

**Reply to Referee #2**

comments - a direct font is used, answers - italics are used

Line 10: Correct this sentence.
*Done.*

Line 23: Can the geostrophic current be computed in another way?
*We corrected the text.*

Lines 39 ff: Perhaps too many references.
*We have reduced the number of refarensec here.*

Lines: 49-50: How do we know that the shallow shelf water is cold and why?
*We have included references to related articles.*

Line 55: There are a few geostrophic transport estimates for the exchanges through Fram Strait: e.g. Rudels (1987), Schlitzholz and Houssais (2002), Rudels et al. (2008), Marnela et al., (2013). Some of these discuss water mass transformations and also the effects of the sloping bottom (based on Jakobsen and Jensen, 1926).
*We have included references to some of the articles You have proposed.*

Line: 60: Internal waves and eddies might disturb the density field and therefore also make geostrophic transport estimates unreliable.
*We have excluded from the text a sentence that seems arguable to You. However, we want to emphasize that in geostrophic assessments ageostrophic disturbances (internal waves and sub-mesoscale eddies) should be filtered out. To calculate geostrophic velocities, it is necessary to carry out an appropriate averaging of the data of the density field in the section. When working with the original data, it is better to rely on estimates of the volume flow rate: with such estimates the ageostrophic effects are almost completely filtered out. This thesis can be proved mathematically (Zhurbas, 2019). But the large-scale synoptic disturbances, which are described in terms of quasi-geostrophic equations, cannot be completely filtered out (for more details see the response to the RC1 review).*

Line 65: Is array the correct word to use here?
*We corrected the text.*

Lines 65-75: This part can be shortened.
*Done.*

Lines 116-118: What does this sentence mean?
*We have corrected this sentence too.*

Lines 148-151: Although the statement that S-max should always be below the T-max is correct, these lines do not make it clearer. A value at a point does not say anything about the stratification. Knowledge of the equation of state is required. I suggest that this is dropped.
*Done*

Lines 170 ff: According to figure 1 all stations in the section are in St. Anna Trough and there are no stations in the Voronin Trough. I suggest that the discussion about the importance of the Voronin Trough is removed or modified. The outflow farther to the east hangs on the slope and contains fairly warm (>0C) Atlantic water as well as colder low salinity Barents Sea branch water. This branch mixes with the Fram Strait branch north of Severnaya Zemlya.
*The discussion about the importance of the Voronin Trough is removed from the text*

Lines around 240, Figure 5: I am surprised that the cold, less saline water at the slope is not commented upon here. This is water of the less dense Barents Sea branch that has entered the Nansen Basin when the slope narrows north of Severnaya Zemlya.
*We have inserted the commentary on less saline water at the slope in the text.*

Lines 277-287: There is not only a decrease in the Fram Strait branch Atlantic water temperature but also in its salinity. These subsurface changes suggest mixing with a less dense part of the Barents Sea branch.
*Here we wanted to pay attention to sharp changes in temperature in the core AW. Detailed changes in temperature and salinity are given in Table 1.*

Line 310, Figures 7a and 8: Which stations are from the Voronin Trough?
*Sorry. We forgot to exclude   "Voronin Trough" from the text here. Now it is done.*

Figure 8e and 8f: Looking at the TS diagram I would expect the cold Barents Sea outflow to enter the Nansen Basin water column at the level where the deep small-scale interleaving is found. Looking at figure 8f, it is not dense enough to reach that deep. However, colder water is more compressible, and if its density is referred to the depth where the interleaving is found, its density might correspond to that of the interleaving.
*Yes, You are right, for the analysis of quasi-isopicnic exchange it is necessary to calculate the potential density from the horizon of the location of the layer with a interleaving structure. We use this in our other works (Kuzmina et al., 2011). However, this article is not devoted to the analysis of intrusive layering and quasi-isopycnic exchange (!).Figures 8e and 8f are needed to demonstrate the transformation of the BSBW signal. We already discussed this issue during the Discussion period: for comparison with the work Dmitrenko et al. (2015), as well as for comparison with transects in Section 3.1.1, we calculated the potential density from the zero horizon. As far as we understood, You agreed with this.*

Line 341: Since there are many bends and "knees" in these TS curve and just for clarity, I would drop the "knee" and only refer to the salinity and temperature minima, (region II).
*We relied on research Dmitrenko et al. (2015), and it is hardly correct to refuse the term introduced in this work. If You write "knees", so You mean many minima by this. Moreover, the term "the salinity and temperature minima" also describes the upper layer, not just BSBW. Therefore, we ask You to agree with the using of the term "knee" in our manuscript: the text explains in detail what feature of TS values this term denotes.*

Lines 368-380: If the Barents Sea branch water mixes isopycnally with the Fram Strait branch water we would not see any perturbations in the density field, but a cooling and freshening of the Fram Strait branch, which is observed. With a broad definition of the Fram Strait branch and a narrow definition of the Barents Sea branch, it is to be expected that the Barents Sea branch becomes absorbed in the Fram Strait branch. The uPDW was originally introduce to explain the stable-stable stratification found below the temperature maximum in the Amerasian Basin and was explained by a vertical redistribution of heat from the Atlantic water to deeper layers by saline and dense entraining plumes.
*The text (lines 368-380) does not talk about the physical mechanisms of the BSBW transformation. Here, based on empirical analysis, the variability of the BSBW signal is described. The physical mechanisms of the BSBW transformation are discussed in Section 4 (Discussion) as recommended by the Scientific Editor of the manuscript.*
*You are talking about an interesting mechanism of the BSBW signal transformation. We can include Your comment in the Discussion section if You provide an appropriate link.*

Lines 390-405: I am not convinced by this discussion without also looking at salinity and density profiles. There are two major outflow of Barents Sea branch water from the Kara Sea. The denser part sinks down the St. Anna Trough and enters below 1000m. The less dense outflow occurs west of Severnaya Zemlya and enters into and below the Fram Strait branch Atlantic core. To separate these two inflow by looking at observation in St. Anna Trough and northeast of Severnaya Zemlya is difficult and perhaps not worth the effort.
*According to Your remark we have shortened the text of the manuscript by deleting Fig. 9.*

*In accordance with the recommendations of another reviewer, instead of the previous Fig. 9, we presented a figure which demonstrated the transformation of "knee" .*

Lines 413-414: How do we know from the section at 82N that there is an eastward (geostrophic?) flow parallel to the section. Are data from the entire section array in the St. Anna Trough considered?

*Of course, without velocity measurements we do not know about the existence of an eastward (geostrophic) flow. But analyzing Fig. 3 (the upper fragment), we can assume the existence of a flow of water to the right through a slight elevation of the bottom. In our calculations we excluded this case in order to avoid errors in estimating the volume flow rate exactly along the St. Anna Trough.*

*We considered all data of CTD transects freely available at http://nabos.iarc.uaf.edu.*

Lines 424-425: There is also the possibility that the uppermost layer, containing the temperature maximum, is transformed by sea ice melting and cooling into a low salinity upper layer and that the temperature maximum observed farther to the east derives from deeper layers.

*We have added this to the text and given the references to the relevant works.*

Line 428: What surrounding waters?

*We think that "surrounding water" is the water of the UPDW. However, in such a context it is not at all necessary to specify the term "surrounding waters".*

Lines 440-450: Cannot the increased distance of the Atlantic water core from the slope at 126E be the beginning of the recirculation of the Fram Strait branch toward Fram Strait?

*We excluded this paragraph from the manuscript: the question of meandering (or recirculation) the flow needs a more thorough analysis*

Line 486: Change "two time smaller than" to "half"

*Done.*

Lines 569-580: The 3 Sv are entering the Nansen Basin. The three branches refer to the splitting of the Norwegian Atlantic Current further south. A total of 2Sv of Atlantic water entering the Nansen Basin is low compared to existing estimates (e.g. Tsubouchi et al., 2012).

*We made a correction: wrote 3Sv.*

Line 605: I cannot see why there cannot be a barotropic velocity component in the presence of sea ice.

*The barotropic additive to velocity arises due to local elevations of the free surface. Unlike the sea without ice cover, where the barotropic additive, as a rule, is created due to the convergence and divergence of the Ekman transport generated by the wind, this mechanism does not work in the sea covered with ice. There are other mechanisms which create a barotropic additive, for example, atmospheric pressure anomalies. However, if the sea is covered with solid ice, local elevations of the free surface are difficult or impossible. In addition, anomalies due to atmospheric pressure anomalies give a weak effect.*

*It is possible that we are not taking something into account, so it would be interesting to get Your opinion on this matter, namely: what reasons (or what forces) can cause a significant barotropic additive when the sea surface is covered with solid ice?*

Lines 660ff: If the flow rate decreases along the path of the flow and at the same time the characteristics becomes weaker due to mixing with surrounding waters, how does this fit with the continuity equation. Where does the water go? Is the boundary current losing mass to a return flow on the basin side of the boundary current?

*All is well with the continuity equation: the mass of water is stored. But the volume flow rate along the flow (especially the AW volume flow rate) can vary due to the absence of rigid impenetrable boundaries between the flow and the surrounding fluid (the volume flow rate does*

*not change in accordance with the continuity equation for a laminar fluid flow, for example, in a tube). In some cases, when the sections are located close to each other, the hypothesis of constancy of the AW volume flow rate can be accepted: for example, to estimate the flow rates during bifurcation of the flow, or, conversely, when two flow branches merge.*
*The reasons for the decrease in the geostrophic volume flow rate can be attributed to the relaxation of the thermohaline AW signal, and as a result of this, a decrease in horizontal density gradients. Local counterflows near the slope, separation of small eddies, turbulent involvement, etc. can also affect the decrease in the volume flow rate.*

Discussion and summary sections: I find these parts too long and too loose. They add questions but provide only more or less speculative answers. These sections should be shortened and more focused.
*We have shortened these sections and made them more focused.*

Summary: This work examines the circulation and transformations of the inflowing Atlantic water mainly in the Eurasian Basin of the Arctic Ocean. The transport estimates are obtained from geostrophic calculations and are generally significant lower than other published estimates. The presentation is overly pedagogical and repeating earlier statements. (I found that some misgivings and questions I had when reading were taken up several pages later. Since I wrote the comments while reading through the text, I found that some of my questions were addressed later in the manuscript. I think that it would shorten the text considerably if such questions are addressed directly. In any case, this manuscript can be considered a part in the ongoing discussion of the circulation of the Atlantic water in the Arctic Ocean and could be published with minor revisions and shortening.
*We have shortened the text of the manuscript according to Your comments, made the necessary corrections and tried to remove the pedagogical statements from the text. We have added to the text only explanations regarding the use of the statistical method to separate spatial and temporal variability (according to the recommendation of the Reviewer 1).*
*According to Your comment "I found that some of my questions were addressed later in the manuscript. I think that it would shorten the text considerably if such questions are addressed directly" – we would like to emphasize that some issues are discussed in the Discussion section. This section was formed after the first review round as recommended by the Scientific Editor. From our point of view, this recommendation allowed to improve the text of the manuscript and make it shorter.*

[revised manuscript text omitted]

---

## Author Response (AR3)

Dear Dr. Fer,

We are very grateful for your help in working on the manuscript.

We took into account all your comments. We also took into account the comments of the Referee #2 and tried to prove the validity of our point of view on the content of the manuscript for the Referee # 1.

We are sending you a clean version #5 of the revised manuscript, a marked-up version #5 of the manuscript and detailed answers to reviewers.

In the marked-up version of the manuscript the additions to the manuscript resulting from correction of the text, are marked in yellow.

Best Regards,

Nataliya Zhurbas

**Referee # 1 review and authors' responses**

R E V I E W
of the manuscript (os-2019-54) entitled
"Assessment of variability of the thermohaline structure and transport of Atlantic
water in the Arctic Ocean based on NABOS CTD data"
by Nataliya Zhurbas and Natalia Kuzmina

This is my third reading of the manuscript. Unfortunately, it was not improved after several loops of editing. My major points of criticism remained the same as they were formulated after reading the first and second versions of the manuscript:

The authors' analysis of temperature and salinity structure (including T-S analysis) does not carry any new information. It (entire Section 3.1 and parts of Section 3.2 devoted to analysis of temperature and salinity) can be completely eliminated from the text which I suggested in my previous review but the authors ignored my comment.

Section devoted to analysis of geostrophic currents includes some new results but it should be placed in the context of the previous similar studies (e.g. Pnyushkov et al. 2018). Besides care should be exercised separating spatial and temporal variability. Both points were stressed in my previous reviews but the authors did not address these points.

Analysis of geostrophic currents is not sufficient to understand the authors' findings. I need to see their computed sections of geostrophic currents, not just mean values presented in the table. I mentioned this point in my previous review but it was also ignored.

Based on that I recommend rejection.

ANSWER:

This review is practically no different from the previous one, despite the fact that the manuscript was deeply reworked after the previous review. In particular, a statistical analysis of volume flow rates was presented, which allows one to separate temporal and spatial variability from each other (see manuscript versions 3 and 4, as well as a detailed answer to the previous review).

The thesis "The authors' analysis of temperature and salinity structure (including T-S analysis) does not carry any new information" does not correspond to reality. The analysis presented in the

manuscript and the conclusions based on this analysis have not been published previously and are a new result.

The first recommendation of Referee#1 – "to exclude entire Section 3.1 and parts of Section 3.2 devoted to analysis of temperature and salinity" – violates the integrity of the manuscript. The exclusion of these sections deprives the authors of the ability to correctly assess, analyze and discuss the variability of geostrophic transport along the slope of the Eurasian Basin (see the answer to the previous review).

The second recommendation of Referee#1, to write a manuscript in the context of previous studies (e.g. Pnyushkov et al., 2018), is at least strange, although the paper (e.g. Pnyushkov et al., 2018) is very important for studying velocity variability in the Eurasian Basin. In this paper a detailed review is given on the estimates of the AW transport in the Eurasian Basin, which were obtained in various studies. However, this paper is devoted to long-term measurements of current velocities based on mooring observations along a section at 126° E. Our work is devoted to estimates of the volume flow rate of the AW based on various CTD sections obtained in different parts of the Eurasian Basin. Thus, our work is more likely to be close to studies in which geostrophic transport is estimated in different areas of the basin based on CTD sections (snapshot observations). Our manuscript provides references to many papers, including (Pnyushkov et al., 2018), and also compares the results obtained with the results of other works.

The third recommendation of Referee#1 (which appeared only in the second review) is to present the calculations of geostrophic currents. However, our work (and this follows from the title of the manuscript) is devoted to the analysis of the variability of the thermohaline field of the AW and its volume flow rate. In the response to the Referee's previous review, isotaches of velocities, estimated by the dynamic method, were presented for one of the sections (see the response to the previous review). It was also shown there that volume flow rate is a reliable estimate of the AW transfer. The representation of the velocity isotaches estimated by the dynamic method (geostrophic currents) for all 40 sections is beyond the scope of this manuscript.

New results of our work are listed in Section 5. From our point of view, they deserve attention.

Thus, the reasons for the rejection of the manuscript by Referee#1 are neither motivated nor clear.

Nevertheless, we are grateful to reviewer No. 1 for useful comments: the manuscript was supplemented by statistical analysis, which allows us to separate the temporal and spatial variability of AW transports.

**Referee # 2 review and authors' responses**

This is the third time I read through this manuscript and I still have difficulties. I think it is mainly due to the way the results are presented. My impression is that the authors underestimate their readers, and that they devote much time and space on explaining processes that every oceanographer knows (or should know) and on describing features in the Arctic Ocean known by anyone working in the field.

One example is the geostrophic boundary current. It is well known that there is an eastward moving geostrophic boundary current along the Eurasian continental slope, with or without a barotropic

reference velocity. The baroclinic part is distinguished by the spreading of the isopycnals towards the slope. This can be mentioned once, but it is not necessary to repeat this statement at every discussed section. Especially not since one of the main results of the study is the geostrophic transport in the boundary current. Should that transport at one section be westward, the observed distribution of the isopycnals could be discussed. No such instances occurred according to table 1.

In any case, I do not think it is fair of a reviewer to bring up something new this late in the game, but the above remarks might be of some use for the authors' next manuscript.

Below I have listed some minor points.

Abstract: I would have preferred present rather than past tense here, but that is a matter of taste.

Answer:
Done.

Line 43: I think that Basin should be written with b if it is not in a name (e.g. Eurasian Basin)

Answer:
Done.

Line 51: Why does a baroclinic flow follow the sea bed topography?

Answer:
We rewrote this sentence.

Line 61 Which Arctic Basin?

Answer:
We wrote "Arctic Ocean" now.

Line 75 Why does mixing with other waters reduce the flow rate? It should rather increase the flow rate, unless only the Atlantic water is meant and that the mixing removes Atlantic water out of its TS definition.

Answer:
We used the word "change" now.

But we want to say the following here.
Mixing and entrainment can indeed increase the volume flow rate by increasing the mass of moving water. However, this is not so obvious as it seems: the average flow velocity due to entrainment may decrease. What happens due to mixing with the volume flow rate at free boundaries can only be shown by simulation with the correct parametrization of the mixing process.

Line 85: Change "whole" to "entire"

Answer:
Done.

Line 118: What does "if we limit ourselves to geostrophic estimates of volume flow" means?

Answer:

This sentence is excluded from the text.

Line 163: "shallow shelf waters" Is here brine enriched cold shelf bottom water meant?

Answer:
We mean the shelf waters cooled due to atmospheric fluxes.

Line 223: "Barents sea Branch water".

Answer:
We corrected the sentence.

Line 232: What latitudes are meant here?

Answer:
Sorry! This is a typo. We corrected this sentence.

Line 295: No stations from the Voronin Trough.

Answer:
Corrected.

Line 323: remove "more"

Answer:
Done.

Line 336: The BSBW cannot be a flow.

Answer:
Corrected.

Line 337: Salinity at mid-depth (800-1000m) below 34.9 indicates input from the shelf and slope, most likely Barents Sea branch water. Salinities below 34.9 are not found at this depth range west of St. Anna Trough.

Answer:
On line 337 of version 4 of the manuscript, the following sentence is written:
«Curve 11, being similar to curve 6 in Fig. 8c, corresponds to the θ-S values of the UPDW.»

Note, that the curve 11 (st. 60), corresponds to the section along 103°E (see the caption under figure 8). Thus, everything is right here.

Line 350: Where in the western Nansen Basin is the UPDW observed? Give reference or station.

The UPDW in the Eurasian Basin is mainly formed by the intrusive mixing of the Barents Sea branch with the Fram Strait branch. In the Amerasian Basin the most characteristic UPDW is found (increasing salinity and decreasing temperature with depth). It is created by interactions between the Atlantic water and entraining dense and saline boundary plumes at the continental slope.

Answer:
Sorry! This is a typo. We rewrote this sentence. Many thanks!

Line 364: How can a flow rate be determined by a water mass? I assume what is meant is that the geostrophic transport of Atlantic water east of 126°E is mainly provided by the Fram Strait branch.

Answer:
We rewrote this sentence.

Lines 365-370: What does this paragraph mean?

Answer:
We excluded this paragraph from the manuscript

Lines 380-385: I do not understand the argument in this paragraph, especially not the meaning of the sentence "The probable presence of the eastward constituent of indefinite vale makes questionable the results of…" I would remove this paragraph.

If a discussion of a possible eastward flow at 82°N is really necessary, I would compute the transports through the 80°N section and the 82°N section, take the difference, and if the transport at 80°N is larger than that at 82°N, the difference could be an estimate of the transport moving eastward and not crossing the 82°N section.

Answer:
Here we cannot agree with You.
The volume flow rate estimate for the section along 82°N turned out to be significantly less (~0.46 Sv) than for other sections in the St. Anna Trough. To estimate the BSBW volume flow rate, that is, the volume flow rate of water entering the Nansen Basin, we cannot use this estimate. The decrease in the volume flow rate is associated, from our point of view, with the flow of these waters to the right. We tried to say it better.

Lines 405-410: The salinity maximum is removed, mainly by mixing with the Barents Sea branch inflow. The deep salinity maximum has no relation to the Atlantic water salinity maximum.

Answer:
Yes, of course, we know that the deep salinity maximum has no relation to the Atlantic water salinity maximum. We do not claim otherwise. Here we just drew the reader's attention to this feature. There is no inaccuracy here.

Line 583: I cannot follow this argument about the ice cover. In the Beaufort Gyre the wind drives the ice towards the center of the gyre increasing the sea level and leading to a dynamic topography that drives the circulation of the gyre. This topographic effect is seen both in the hydrography and by altimeter observations by satellites. I cannot see why a similar situation can arise in the Nansen Basin.

Answer:
We wrote this sentence more gently.

From our point of view, we can still assume that if the surface of the stratified ocean is covered with solid ice, then the barotropic velocity addition is unlikely to exceed the baroclinic component.

Thank You very much for careful reading of our manuscript. Your comments were very helpful to us.

[revised manuscript text omitted]

---

## Author Response (AR4)

Dear Dr. Fer,

We are very grateful for your help in working on the manuscript.
We corrected the text of the manuscript in accordance with your comments.

In the text of the manuscript, we would like to save the term "absolutely stable" stratification, which is often used to describe the UPDW layer. The description of the formation of intrusions in the UPDW layer differs from the description of intrusions due to double diffusion.

We are sending you a clean version #6 of the revised manuscript and a marked-up
version #6 of the revised manuscript.

In the marked-up version of the manuscript the additions to the manuscript resulting from correction of the text, are marked in yellow.

Best Regards,
Nataliya Zhurbas